# Gen-Oja: A Simple and Efficient Algorithm for Streaming Generalized Eigenvector Computation

**Kush Bhatia**[*]
University of California, Berkeley
kushbhatia@berkeley.edu

**Aldo Pacchiano**[*]
University of California, Berkeley
pacchiano@berkeley.edu

**Nicolas Flammarion**
University of California, Berkeley
flammarion@berkeley.edu

**Peter L. Bartlett**
University of California, Berkeley
peter@berkeley.edu

**Michael I. Jordan**
University of California, Berkeley
jordan@cs.berkeley.edu

## Abstract

In this paper, we study the problems of principal Generalized Eigenvector computation and Canonical Correlation Analysis in the stochastic setting. We propose a simple and efficient algorithm, Gen-Oja, for these problems. We prove the global convergence of our algorithm, borrowing ideas from the theory of fast-mixing Markov chains and two-time-scale stochastic approximation, showing that it achieves the optimal rate of convergence. In the process, we develop tools for understanding stochastic processes with Markovian noise which might be of independent interest.

## 1 Introduction

Cannonical Correlation Analysis (CCA) and the Generalized Eigenvalue Problem are two fundamental problems in machine learning and statistics, widely used for feature extraction in applications including regression [18], clustering [9] and classification [19].

Originally introduced by Hotelling in [16], CCA is a statistical tool for the analysis of multi-view data that can be viewed as a "correlation-aware" version of Principal Component Analysis (PCA). Given two multidimensional random variables, the objective in CCA is to obtain a pair of linear transformations that maximize the correlation between the transformed variables.

Given access to samples $\{(x_i, y_i)_{i=1}^n\}$ of zero mean random variables $X, Y \in \mathbb{R}^d$ with an unknown joint distribution $P_{XY}$, CCA can be used to discover features expressing similarity or dissimilarity between $X$ and $Y$. Formally, CCA aims to find a pair of vectors $u, v \in \mathbb{R}^d$ such that projections of $X$ onto $v$ and $Y$ onto $u$ are maximally correlated. In the population setting, the corresponding objective is given by:

$$\max v^\top \mathbb{E}[XY^\top] u \qquad \text{s.t.} \quad v^\top \mathbb{E}[XX^\top] v = 1 \text{ and } u^\top \mathbb{E}[YY^\top] u = 1. \qquad (1)$$

---

[*]Equal contribution.

In the context of covariance matrices, the objective of the generalized eigenvalue problem is to obtain the direction $u$ or $v \in \mathbb{R}^d$ maximizing discrepancy between $X$ and $Y$ and can be formulated as,

$$\arg\max_{v \neq 0} \frac{v^\top \mathbb{E}[XX^\top]v}{v^\top \mathbb{E}[YY^\top]v} \text{ and } \arg\max_{u \neq 0} \frac{u^\top \mathbb{E}[YY^\top]u}{u^\top \mathbb{E}[XX^\top]u}. \quad (2)$$

More generally, given symmetric matrices $A, B$, with $B$ positive definite, the objective of the principal generalized eigenvector problem is to obtain a unit norm vector $w$ such that $Aw = \lambda Bw$ for $\lambda$ maximal.

CCA and the generalized eigenvalue problem are intimately related. In fact, the CCA problem can be cast as a special case of the generalized eigenvalue problem by solving for $u$ and $v$ in the following objective:

$$\underbrace{\begin{pmatrix} 0 & \mathbb{E}[XY^\top] \\ \mathbb{E}[YX^\top] & 0 \end{pmatrix}}_{A} \begin{pmatrix} v \\ u \end{pmatrix} = \lambda \underbrace{\begin{pmatrix} \mathbb{E}[XX^\top] & 0 \\ 0 & \mathbb{E}[YY^\top] \end{pmatrix}}_{B} \begin{pmatrix} v \\ u \end{pmatrix}. \quad (3)$$

The optimization problems underlying both CCA and the generalized eigenvector problem are non-convex in general. While they admit closed-form solutions, even in the offline setting a direct computation requires $O(d^3)$ flops which is infeasible for large-scale datasets. Recently, there has been work on solving these problems by leveraging fast linear system solvers [14, 2] while requiring complete knowledge of the matrices $A$ and $B$.

In the stochastic setting, the difficulty increases because the objective is to maximize a ratio of expectations, in contrast to the standard setting of stochastic optimization [26], where the objective is the maximization of an expectation. There has been recent interest in understanding and developing efficient algorithms with provable convergence guarantees for such non-convex problems. [17] and [27] recently analyzed the convergence rate of Oja's algorithm [25], one of the most commonly used algorithm for streaming PCA.

In contrast, for the stochastic generalized eigenvalue problem and CCA problem, the focus has been to translate algorithms from the offline setting to the online one. For example, [12] proposes a streaming algorithm for the stochastic CCA problem which utilizes a streaming SVRG method to solve an online least-squares problem. Despite being streaming in nature, this algorithm requires a non-trivial initialization and, in contrast to the spirit of streaming algorithms, updates its eigenvector estimate only after every few samples. This raises the following challenging question:

> *Is it possible to obtain an efficient and provably convergent counterpart to Oja's Algorithm for computing the principal generalized eigenvector in the stochastic setting?*

In this paper, we propose a simple, globally convergent, *two-line* algorithm, Gen-Oja, for the stochastic principal generalized eigenvector problem and, as a consequence, we obtain a natural extension of Oja's algorithm for the streaming CCA problem. Gen-Oja is an iterative algorithm which works by updating two coupled sequences at every time step. In contrast with existing methods [17], at each time step the algorithm can be seen as performing a step of Oja's method, with a noise term which is neither *zero mean* nor *conditionally independent*, but instead is Markovian in nature. The analysis of the algorithm borrows tools from the theory of fast mixing of Markov chains [11] as well as two-time-scale stochastic approximation [6, 7, 8] to obtain an optimal (up to dimension dependence) fast convergence rate of $\tilde{O}(1/n)$.

**Notation**: We denote by $\lambda_i(M)$ and $\sigma_i(M)$ the $i^{th}$ largest eigenvalue and singular value of a square matrix $M$. For any positive semi-definite matrix $N$, we denote inner product in the $N$-norm by $\langle \cdot, \cdot \rangle_N$ and the corresponding norm by $\| \cdot \|_N$. We let $\kappa_N = \frac{\lambda_{\max}(N)}{\lambda_{\min}(N)}$ denote the condition number of $N$. We denote the eigenvalues of the matrix $B^{-1}A$ by $\lambda_1 > \lambda_2 \geq \ldots \geq \lambda_d$ with $(u_i)_{i=1}^d$ and $(\tilde{u}_i)_{i=1}^d$ denoting the corresponding right and left eigenvectors of $B^{-1}A$ whose existence is guaranteed by Lemma G.3 in Appendix G.3. We use $\Delta_\lambda$ to denote the eigengap $\lambda_1 - \lambda_2$.

## 2 Problem Statement

In this section, we focus on the problem of estimating principal generalized eigenvectors in a stochastic setting. The generalized eigenvector, $v_i$, corresponding to a system of matrices $(A, B)$,

where $A \in \mathbb{R}^{d \times d}$ is a symmetric matrix and $B \in \mathbb{R}^{d \times d}$ is a symmetric positive definite matrix, satisfies

$$Av_i = \lambda_i Bv_i. \qquad (4)$$

The principal generalized eigenvector $v_1$ corresponds to the vector with the largest value[2] of $\lambda_i$, or, equivalently, $v_1$ is the principal eigenvector of the non-symmetric matrix $B^{-1}A$. The vector $v_1$ also corresponds to the maximizer of the generalized Rayleigh quotient given by

$$v_1 = \arg\max_{v \in \mathbb{R}^d} \frac{v^\top Av}{v^\top Bv}. \qquad (5)$$

In the stochastic setting, we only have access to a sequence of matrices $A_1, \ldots, A_n \in \mathbb{R}^{d \times d}$ and $B_1, \ldots, B_n \in \mathbb{R}^{d \times d}$ assumed to be drawn i.i.d. from an unknown underlying distribution, such that $\mathbb{E}[A_i] = A$ and $\mathbb{E}[B_i] = B$ and the objective is to estimate $v_1$ given access to $O(d)$ memory.

In order to quantify the error between a vector and its estimate, we define the following generalization of the sine with respect to the $B$-norm as,

$$\sin_B^2(v, w) = 1 - \Big(\frac{v^\top Bw}{\|v\|_B \|w\|_B}\Big)^2. \qquad (6)$$

## 3   Related Work

**PCA.**   There is a vast literature dedicated to the development of computationally efficient algorithms for the PCA problem in the offline setting (see [23, 13] and references therein). In the stochastic setting, sharp convergence results were obtained recently by [17] and [27] for the principal eigenvector computation problem using Oja's algorithm and later extended to the streaming k-PCA setting by [1]. They are able to obtain a $O(1/n)$ convergence rate when the eigengap of the matrix is positive and a $O(1/\sqrt{n})$ rate is attained in the gap free setting.

**Offline CCA and generalized eigenvector.**   Computationally efficient optimization algorithms with finite convergence guarantees for CCA and the generalized eigenvector problem based on Empirical Risk Minimization (ERM) on a fixed dataset have recently been proposed in [14, 31, 2]. These approaches work by reducing the CCA and generalized eigenvector problem to that of solving a PCA problem on a modified matrix $M$ (e.g., for CCA, $M = B^{\frac{-1}{2}} A B^{\frac{-1}{2}}$). This reformulation is then solved by using an approximate version of the Power Method that relies on a linear system solver to obtain the approximate power method step. [14, 2] propose an algorithm for the generalized eigenvector computation problem and instantiate their results for the CCA problem. [20, 21, 31] focus on the CCA problem by optimizing a different objective:

$$\min \frac{1}{2} \hat{\mathbb{E}} |\phi^\top x_i - \psi^\top y_i|^2 + \lambda_x \|\phi\|_2^2 + \lambda_y \|\psi\|_2^2 \quad \text{s.t.} \quad \|\phi\|_{\hat{\mathbb{E}}[xx^\top]} = \|\psi\|_{\hat{\mathbb{E}}[yy^\top]} = 1,$$

where $\hat{\mathbb{E}}$ denotes the empirical expectation. The proposed methods utilize the knowledge of complete data in order to solve the ERM problem, and hence is unclear how to extend them to the stochastic setting.

**Stochastic CCA and generalized eigenvector.**   There has been a dearth of work for solving these problems in the stochastic setting owing to the difficulties mentioned in Section 1. Recently, [12] extend the algorithm of [31] from the offline to the streaming setting by utilizing a streaming version of the SVRG algorithm for the least squares system solver. Their algorithm, based on the shift and invert method, suffers from two drawbacks: a) contrary to the spirit of streaming algorithms, this method does not update its estimate at each iteration – it requires to use logarithmic samples for solving an online least squares problem, and, b) their algorithm critically relies on obtaining an estimate of $\lambda_1$ to a small accuracy for which it requires to burn a few samples in the process. In comparison, Gen-Oja takes a *single* stochastic gradient step for the inner least squares problem and updates its estimate of the eigenvector after each sample. Perhaps the closest to our approach is [4], who propose an online method by solving a convex relaxation of the CCA objective with an inexact stochastic mirror descent algorithm. Unfortunately, the computational complexity of their method is $O(d^2)$ which renders it infeasible for large-scale problems.

---
**Algorithm 1:** Gen-Oja for Streaming $Av = \lambda Bv$

---
**Input:** Time steps $T$, step size $\alpha_t$ (Least Squares), $\beta_t$ (Oja)
**Initialize:** $(w_0, v_0) \leftarrow$ sample uniformly from the unit sphere in $\mathbb{R}^d$, $\bar{v}_0 = v_0$
**for** $t = 1, \ldots, T$ **do**
    Draw sample $(A_t, B_t)$
    $w_t \leftarrow w_{t-1} - \alpha_t(B_t w_{t-1} - A_t v_{t-1})$
    $v'_t \leftarrow v_{t-1} + \beta_t w_t$
    $v_t \leftarrow \frac{v'_t}{\|v_t\|_2}$

**Output:** Estimate of Principal Generalized Eigenvector: $v_T$

---

## 4  Gen-Oja

In this section, we describe our proposed approach for the stochastic generalized eigenvector problem (see Section 2). Our algorithm Gen-Oja, described in Algorithm 1, is a natural extension of the popular Oja's algorithm used for solving the streaming PCA problem. The algorithm proceeds by iteratively updating two coupled sequences $(w_t, v_t)$ at the same time: $w_t$ is updated using one step of stochastic gradient descent with constant step-size to minimize $w^\top B w - 2 w^\top A v_t$ and $v_t$ is updated using a step of Oja's algorithm. Gen-Oja has its roots in the theory of two-time-scale stochastic approximation, by viewing the sequence $w_t$ as a fast mixing Markov chain and $v_t$ as a slowly evolving one. In the sequel, we describe the evolution of the Markov chains $(w_t)_{t \geq 0}, (v_t)_{t \geq 0}$, in the process outlining the intuition underlying Gen-Oja and understanding the key challenges which arise in the convergence analysis.

**Oja's algorithm.**  Gen-Oja is closely related to the Oja's algorithm [25] for the streaming PCA problem. Consider a special case of the problem, when each $B_t = I$. In the offline setting, this reduces the generalized eigenvector problem to that of computing the principal eigenvector of A. With the setting of step-size $\alpha_t = 1$, Gen-Oja recovers the Oja's algorithm given by

$$v_t = \frac{v_{t-1} + \beta_t A_t v_{t-1}}{\|v_{t-1} + \beta_t A_t v_{t-1}.\|}$$

This algorithm is exactly a projected stochastic gradient ascent on the Rayleigh quotient $v^\top A v$ (with a step size $\beta_t$). Alternatively, it can be interpreted as a randomized power method on the matrix $(I + \beta_t A)$[15].

**Two-time-scale approximation.**  The theory of two-time-scale approximation forms the underlying basis for Gen-Oja. It considers coupled iterative systems where one component changes much faster than the other [7, 8]. More precisely, its objective is to understand classical systems of the type:

$$x_t = x_{t-1} + \alpha_t \left[ h(x_{t-1}, y_{t-1}) + \xi_t^1 \right] \tag{7}$$
$$y_t = y_{t-1} + \beta_t \left[ g(x_{t-1}, y_{t-1}) + \xi_t^2 \right], \tag{8}$$

where $g$ and $h$ are the update functions and $(\xi_t^1, \xi_t^2)$ correspond to the noise vectors at step $t$ and typically assumed to be martingale difference sequences.

In the above model, whenever the two step sizes $\alpha_t$ and $\beta_t$ satisfy $\beta_t / \alpha_t \to 0$, the sequence $y_t$ moves on a slower timescale than $x_t$. For any fixed value of $y$ the dynamical system given by $x_t$,

$$x_t = x_{t-1} + \alpha_t [h(x_{t-1}, y) + \xi_t^1], \tag{9}$$

converges to to a solution $x^*(y)$. In the coupled system, since the state variables $x_t$ move at a much faster time scale, they can be seen as being close to $x^*(y_t)$, and thus, we can alternatively consider:

$$y_t = y_{t-1} + \beta_t \left[ g(x_*(y_{t-1}), y_{t-1}) + \xi_t^2 \right]. \tag{10}$$

If the process given by $y_t$ above were to converge to $y^*$, under certain conditions, we can argue that the coupled process $(x_t, y_t)$ converges to $(x^*(y^*), y^*)$. Intuitively, because $x_t$ and $y_t$ are evolving at different time-scales, $x_t$ views the process $y_t$ as quasi-constant while $y_t$ views $x_t$ as a process rapidly converging to $x^*(y_t)$.

Gen-Oja can be seen as a particular instance of the coupled iterative system given by Equations (7) and (8) where the sequence $v_t$ evolves with a step-size $\beta_t \approx \frac{1}{t}$, much slower than the sequence $w_t$, which has a step-size of $\alpha_t \approx \frac{1}{\log(t)}$. Proceeding as above, the sequence $v_t$ views $w_t$ as having converged to $B^{-1}Av_t + \xi_t$, where $\xi_t$ is a noise term, and the update step for $v_t$ in Gen-Oja can be viewed as a step of Oja's algorithm, albeit with Markovian noise.

While previous works on the stochastic CCA problem required to use logarithmic independent samples to solve the inner least-squares problem in order to perform an approximate power method (or Oja) step, the theory of two-time-scale stochastic approximation suggests that it is possible to obtain a similar effect by evolving the sequences $w_t$ and $v_t$ at two different time scales.

**Understanding the Markov Process** $\{w_t\}$. In order to understand the process described by the sequence $w_t$, we consider the homogeneous Markov chain $(w_t^v)$ defined by

$$w_t^v = w_{t-1}^v - \alpha(B_t w_{t-1}^v - A_t v), \tag{11}$$

for a constant vector $v$ and we denote its $t$-step kernel by $\pi_v^t$ [22]. This Markov process is an iterative linear model and has been extensively studied by [28, 10, 5]. It is known that for any step-size $\alpha \leq 2/R^2$, the Markov chain $(w_t^v)_{t\geq 0}$ admits a unique stationary distribution, denoted by $\nu_v$. In addition,

$$W_2^2(\pi_v^t(w_0, \cdot), \nu_v) \leq (1 - 2\mu\alpha(1 - \alpha R_B^2/2))^t \int_{\mathbb{R}^d} \|w_0 - w\|_2^2 d\nu_v(w), \tag{12}$$

where $W_2^2(\lambda, \nu)$ denotes the Wasserstein distance of order 2 between probability measures $\lambda$ and $\nu$ (see, e.g., [30] for more properties of $W_2$). Equation (12) implies that the iterative linear process described by (11) mixes exponentially fast to the stationary distribution. This forms a crucial ingredient in our convergence analysis where we use the fast mixing to obtain a bound on the expected norm of the Markovian noise (see Lemma 6.1).

Moreover, one can compute the mean $\bar{w}^v$ of the process $w_t$ under the stationary distribution by taking expectation under $\nu_v$ on both sides in equation (11). Doing so, we obtain, $\bar{w}^v = B^{-1}Av$. Thus, in our setting, since the $v_t$ process evolves slowly, we can expect that $w_t \approx B^{-1}Av_t$, allowing Gen-Oja to mimic Oja's algorithm.

# 5 Main Theorem

In this section, we present our main convergence guarantee for Gen-Oja when applied to the streaming generalized eigenvector problem. We begin by listing the key assumptions required by our analysis:

**(A1)** The matrices $(A_i)_{i\geq 0}$ satisfy $\mathbb{E}[A_i] = A$ for a symmetric matrix $A \in \mathbb{R}^{d\times d}$.

**(A2)** The matrices $(B_i)_{i\geq 0}$ are such that each $B_i \succcurlyeq 0$ is symmetric and satisfies $\mathbb{E}[B_i] = B$ for a symmetric matrix $B \in \mathbb{R}^{d\times d}$ with $B \succcurlyeq \mu I$ for $\mu > 0$.

**(A3)** There exists $R \geq 0$ such that $\max\{\|A_i\|, \|B_i\|\} \leq R$ almost surely.

Under the assumptions stated above, we obtain the following convergence theorem for Gen-Oja with respect to the $\sin_B^2$ distance, as described in Section 2.

**Theorem 5.1** (Main Result). *Fix any $\delta > 0$ and $\epsilon_1 > 0$. Suppose that the step sizes are set to $\alpha_t = \frac{c}{\log(d^2\beta+t)}$ and $\beta_t = \frac{\gamma}{\Delta_\lambda(d^2\beta+t)}$ for $\gamma > 1/2$, $c > 1$ and*

$$\beta = \max\left(\frac{20\gamma^2\lambda_1^2}{\Delta_\lambda^2 d^2 \log\left(\frac{1+\delta/100}{1+\epsilon_1}\right)}, \frac{200\left(\frac{R}{\mu} + \frac{R^3}{\mu^2} + \frac{R^5}{\mu^3}\right)\log\left(1 + \frac{R^2}{\mu} + \frac{R^4}{\mu^2}\right)}{\delta\Delta_\lambda^2}\right).$$

*Suppose that the number of samples $n$ satisfy*

$$\frac{d^2\beta + n}{\log^{\frac{1}{\min(1,2\gamma\lambda_1/\Delta_\lambda)}}(d^2\beta + n)} \geq \left(\frac{cd}{\delta_1\min(1,\lambda_1)}\right)^{\frac{1}{\min(1,2\gamma\lambda_1/\Delta_\lambda)}} (d^3\beta + 1)\exp\left(\frac{c\lambda_1^2}{d^2}\right)$$

*Then, the output $v_n$ of Algorithm 1 satisfies,*

$$\sin_B^2(u_1, v_n) \leq \frac{(2+\epsilon_1)cd\|\sum_{i=1}^d \tilde{u}_i\tilde{u}_i^\top\|_2 \log\left(\frac{1}{\delta}\right)}{\delta^2\|\tilde{u}_1\|_2^2}\left(\frac{c\gamma^2 \log^3(d^2\beta+n)}{\Delta_\lambda^2(d^2\beta+n+1)} + \frac{cd}{\Delta_\lambda}\left(\frac{d^2\beta + \log^3(d^2\beta)}{d^2\beta+n+1}\right)^{2\gamma}\right),$$

*with probability at least $1 - \delta$ with $c$ depending polynomially on parameters of the problem $\lambda_1, \kappa_B, R, \mu$. The parameter $\delta_1$ is set as $\delta_1 = \frac{\epsilon_1}{2(2+\epsilon_1)}$.*

The above result shows that with probability at least $1 - \delta$, Gen-Oja converges in the $B$-norm to the right eigenvector, $u_1$, corresponding to the maximum eigenvalue of the matrix $B^{-1}A$. Further, Gen-Oja exhibits an $\tilde{O}(1/n)$ rate of convergence, which is known to be optimal for stochastic approximation algorithms even with convex objectives [24].

**Comparison with Streaming PCA.** In the setting where $B = I$, and $A \succeq 0$ is a covariance matrix, the principal generalized eigenvector problem reduces to performing PCA on the $A$. When compared with the results obtained for streaming PCA by [17], our corresponding results differ by a factor of dimension $d$ and problem dependent parameters $\lambda_1, \Delta_\lambda$. We believe that such a dependence is not inherent to Gen-Oja but a consequence of our analysis. We leave this task of obtaining a dimension free bound for Gen-Oja as future work.

**Gap-independent step size**: While the step size for the sequence $v_n$ in Gen-Oja depends on eigen-gap, which is a priori unknown, one can leverage recent results as in [29] to get around this issue by using a streaming average step size.

## 6 Proof Sketch

In this section, we detail out the two key ideas underlying the analysis of Gen-Oja to obtain the convergence rate mentioned in Theorem 5.1: a) controlling the non i.i.d. Markovian noise term which is introduced because of the coupled Markov chains in Gen-Oja and b) proving that a noisy power method with such Markovian noise converges to the correct solution.

**Controlling Markovian perturbations.** In order to better understand the sequence $v_t$, we rewrite the update as,

$$v'_t = v_{t-1} + \beta_t w_t = v_{t-1} + \beta_t(B^{-1}Av_{t-1} + \xi_t), \tag{13}$$

where $\xi_t = w_t - B^{-1}Av_{t-1}$ is the prediction error which is a Markovian noise. Note that the noise term is neither *mean zero* nor a *martingale difference* sequence. Instead, the noise term $\xi_t$ is dependent on all previous iterates, which makes the analysis of the process more involved. This framework with Markovian noise has been extensively studied by [6, 3].

From the update in Equation (13), we observe that Gen-Oja is performing an Oja update but with a controlled Markovian noise. However, we would like to highlight that classical techniques in the study of stochastic approximation with Markovian noise (as the *Poisson Equation* [6, 22]) were not enough to provide adequate control on the noise to show convergence.

In order to overcome this difficulty, we leverage the fast mixing of the chain $w_t^v$ for understanding the Markovian noise. While it holds that $\mathbb{E}[\|\xi_t\|_2] = O(1)$ (see Appendix C), a key part of our analysis is the following lemma, the proof of which can be found in Appendix B.

**Lemma 6.1.** . *For any choice of $k > 4\frac{\lambda_1(B)}{\mu\alpha}\log(\frac{1}{\beta_{t+k}})$, and assuming that $\|w_s\| \le W_s$ for $t \le s \le t+k$ we have that*

$$\|\mathbb{E}[\epsilon_{t+k}|\mathcal{F}_t]\|_2 = O(\beta_t k^2 \alpha_t W_{t+k})$$

Lemma 6.1 uses the fast mixing of $w_t$ to show that $\|\mathbb{E}[\xi_t|\mathcal{F}_{t-r}]\|_2 = \tilde{O}(\beta_t)$ where $r = O(\log t)$, i.e., the magnitude of the expected noise is small conditioned on $\log(t)$ steps in the past.

**Analysis of Oja's algorithm.** The usual proofs of convergence for stochastic approximation define a Lyapunov function and show that it decreases sufficiently at each iteration. Oftentimes control on the per step rate of decrease can then be translated into a global convergence result. Unfortunately in the context of PCA, due to the non-convexity of the Raleigh quotient, the quality of the estimate $v_t$ cannot be related to the previous $v_{t-1}$. Indeed $v_t$ may become orthogonal to the leading eigenvector. Instead [17] circumvent this issue by leveraging the randomness of the initialization and adopt an operator view of the problem. We take inspiration from this approach in our analysis of Gen-Oja. Let $G_i = w_i v_{i-1}^\top$ and $H_t = \prod_{i=1}^t (I + \beta_i G_i)$, Gen-Oja's update can be equivalently written as

$$v_t = \frac{H_t v_0}{\|H_t v_0\|_2^2},$$

pushing, for the analysis only, the normalization step at the end. This point of view enables us to analyze the improvement of $H_t$ over $H_{t-1}$ since allows one to interpret Oja's update as one step of power method on $H_t$ starting on a random vector $v_0$. We present here an easy adaptation of [17, Lemma 3.1] that takes into account the special geometry of the generalized eigenvector problem and the asymmetry of $B^{-1}A$. The proof can be found in Appendix A.

**Lemma 6.2.** *Let $H \in \mathbb{R}^{d \times d}$, $(u_i)_{i=1}^d$ and $(\tilde{u}_i)_{i=1}^d$ be the corresponding right and left eigenvectors of $B^{-1}A$ and $w \in \mathbb{R}^d$ chosen uniformly on the sphere, then with probability $1 - \delta$ (over the randomness in the initial iterate)*

$$\sin_B^2(u_i, Hw) \leq \frac{C \log(1/\delta)}{\delta} \frac{\text{Tr}(HH^\top \sum_{j \neq i} \tilde{u}_j \tilde{u}_j^\top)}{\tilde{u}_i^\top HH^\top \tilde{u}_i}, \tag{14}$$

*for some universal constant $C > 0$.*

This lemma has the virtue of highly simplifying the challenging proof of convergence of Oja's algorithm. Indeed we only have to prove that $H_t$ will be close to $\prod_{i=1}^t (I + \beta_i B^{-1}A)$ for $t$ large enough which can be interpreted as an analogue of the law of large numbers for the multiplication of matrices. This will ensure that $\text{Tr}(H_t H_t^\top \sum_{j \neq i} \tilde{u}_j \tilde{u}_j^\top)$ is relatively small compared to $\tilde{u}_i^\top H_t H_t^\top \tilde{u}_i$ and be enough with Lemma 6.2 to prove Theorem 5.1. The proof follows the line of [17] with two additional tedious difficulties: the Markovian noise is neither unbiased nor independent of the previous iterates, and the matrix $B^{-1}A$ is no longer symmetric, which is precisely why we consider the left eigenvector $\tilde{u}_i$ in the right-hand side of Eq. (14). We highlight two key steps:

- First we show that $\mathbb{E} \, \text{Tr}(H_t H_t^\top \sum_{j \neq i} \tilde{u}_j \tilde{u}_j^\top)$ grows as $O(\exp(2\lambda_2 \sum_{i=1}^t \beta_i))$, which implies by Markov's inequality the same bound on $\text{Tr}(H_t H_t^\top \sum_{j \neq i} \tilde{u}_j \tilde{u}_j^\top)$ with constant probability. See Lemmas E.2 for more details.
- Second we show that $\text{Var} \, \tilde{u}_i^\top H_t H_t^\top \tilde{u}_i$ grows as $O(\exp(4\lambda_1 \sum_{i=1}^t \beta_i))$ and $\mathbb{E} \tilde{u}_i^\top HH^\top \tilde{u}_i$ grows as $O(\exp(2\lambda_1 \sum_{i=1}^t \beta_i))$ which implies by Chebshev's inequality the same bound for $\tilde{u}_i^\top HH^\top \tilde{u}_i$ with constant probability. See Lemmas E.3 and E.5 for more details.

# 7 Application to Canonical Correlation Analysis

Consider two random vectors $X \in \mathbb{R}^d$ and $Y \in \mathbb{R}^d$ with joint distribution $P_{XY}$. The objective of canonical correlation analysis in the population setting is to find the canonical correlation vectors $\phi, \psi \in \mathbb{R}^{d,d}$ which maximize the correlation

$$\max_{\phi, \psi} \frac{\mathbb{E}[(\phi^\top X)(\psi^\top Y)]}{\sqrt{\mathbb{E}[(\phi^\top X)^2] \mathbb{E}[(\psi^\top Y)^2]}}.$$

This problem is equivalent to maximizing $\phi^\top \mathbb{E}[XY^\top]\psi$ under the constraint $\mathbb{E}[(\phi^\top X)^2] = \mathbb{E}[(\psi^\top Y)^2] = 1$ and admits a closed form solution: if we define $T = \mathbb{E}[XX^\top]^{-1/2}\mathbb{E}[XY^\top]\mathbb{E}[YY^\top]^{-1/2}$, then the solution is $(\phi_*, \psi_*) = (\mathbb{E}[XX^\top]^{-1/2}a_1 \mathbb{E}[YY^\top]^{-1/2}b_1)$ where $a_1, b_1$ are the left and right principal singular vectors of $T$. By the KKT conditions, there exist $\nu_1, \nu_2 \in \mathbb{R}$ such that this solution satisfies the stationarity equation

$$\mathbb{E}[XY^\top]\psi = \nu_1 \mathbb{E}[XX^\top]\phi \quad \text{and} \quad \mathbb{E}[YX^\top]\phi = \nu_2 \mathbb{E}[YY^\top]\psi.$$

Using the constraint conditions we conclude that $\nu_1 = \nu_2$. This condition can be written (for $\lambda = \nu_1$) in the matrix form of Eq. (3). As a consequence, finding the largest generalized eigenvector for the matrices $(A, B)$ will recover the canonical correlation vector $(\phi, \psi)$. Solving the associated generalized streaming eigenvector problem, we obtain the following result for estimating the canonical correlation vector whose proof easily follows from Theorem 5.1 (setting $\gamma = 6$).

**Theorem 7.1.** *Assume that $\max\{\|X\|, \|Y\|\} \leq R$ a.s., $\min\{\lambda_{\min}(\mathbb{E}[XX^\top]), \lambda_{\min}(\mathbb{E}[YY^\top])\} = \mu > 0$ and $\sigma_1(T) - \sigma_2(T) = \Delta > 0$. Fix any $\delta > 0$, let $\epsilon_1 \geq 0$, and suppose the step sizes are set to $\alpha_t = \frac{1}{2R^2 \log(d^2\beta + t)}$ and $\beta_t = \frac{6}{\Delta(d^2\beta + t)}$ and*

$$\beta = \max \left( \frac{720\sigma_1^2}{\Delta^2 d^2 \log\left(\frac{1+\delta/100}{1+\epsilon_1}\right)}, \frac{200 \left(\frac{R}{\mu} + \frac{R^3}{\mu^2} + \frac{R^5}{\mu^3}\right) \frac{1}{\delta} \log(1 + \frac{R^2}{\mu} + \frac{R^4}{\mu^2})}{\Delta^2} \right)$$

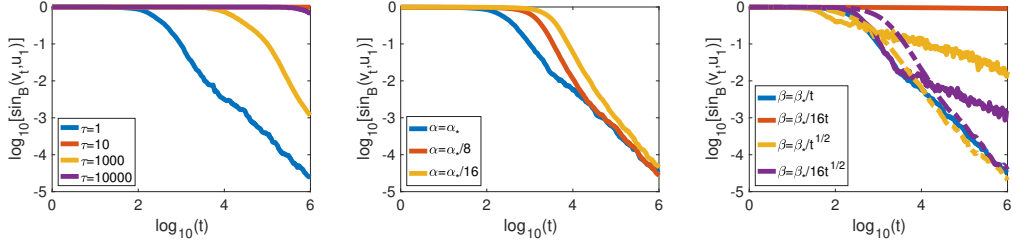

Figure 1: Synthetic Generalized Eigenvalue problem. Left: Comparison with two-steps methods. Middle: Robustness to step size $\alpha_t$. Right: Robustness to step size $\beta_t$ (Streaming averaged Gen-Oja is dashed).

*Suppose that the number of samples $n$ satisfy*

$$\frac{d^2\beta + n}{\log^{\frac{1}{\min(1,12\lambda_1/\Delta_\lambda)}}(d^2\beta + n)} \geq \left(\frac{cd}{\delta_1 \min(1,\lambda_1)}\right)^{\frac{1}{\min(1,12\lambda_1/\Delta_\lambda)}} (d^3\beta + 1)\exp\left(\frac{c\lambda_1^2}{d^2}\right)$$

*Then the output $(\phi_t, \psi_t)$ of Algorithm 1 applied to $(A,B)$ defined above satisfies,*

$$\sin^2_B((\phi_*, \psi_*),(\phi_t,\psi_t)) \leq \frac{(2+\epsilon_1)cd^2\log\left(\frac{1}{\delta}\right)}{\delta^2\|\tilde{u}_1\|_2^2}\frac{\log^3(d^2\beta + n)}{\Delta^2(d^2\beta + n + 1)},$$

*with probability at least $1 - \delta$ with $c$ depending on parameters of the problem and independent of $d$ and $\Delta$ where $\delta_1 = \frac{\epsilon_1}{2(2+\epsilon_1)}$.*

We can make the following observations:

- The convergence guarantee are comparable with the sample complexity obtained by the ERM ($t = \tilde{O}(d/(\varepsilon\Delta^2)$ for sub-Gaussian variables and $t = \tilde{O}(1/(\varepsilon\Delta^2\mu^2)$ for bounded variables)[12].

- The sample complexity in [12] is better in term of the dependence on $d$. They obtain the same rates as the ERM. We are unable to explicitly compare our bounds with [4] since they work in the gap free setting and their computational complexity is $O(d^2)$.

## 8 Simulations

Here we illustrate the practical utility of Gen-Oja on a synthetic, streaming generalized eigenvector problem. We take $d = 20$ and $T = 10^6$. The streams $(A_t, B_t) \in (\mathbb{R}^{d\times d})^2$ are normally-distributed with covariance matrix $A$ and $B$ with random eigenvectors and eigenvalues decaying as $1/i$, for $i = 1, \ldots, d$. Here $R^2$ denotes the radius of the streams with $R^2 = \max\{\operatorname{Tr} A, \operatorname{Tr} B\}$. All results are averaged over ten repetitions.

**Comparison with two-steps methods.** In the left plot of Figure 1 we compare the behavior of Gen-Oja to different two-steps algorithms. Since the method by [4] is of complexity $O(d^2)$, we compare Gen-Oja to a method which alternates between one step of Oja's algorithm and $\tau$ steps of averaged stochastic gradient descent with constant step size $1/2R^2$. Gen-Oja is converging at rate $O(1/t)$ whereas the other methods are very slow. For $\tau = 10$, the solution of the inner loop is too inaccurate and the steps of Oja are inefficient. For $\tau = 10000$, the output of the sgd steps is very accurate but there are too few Oja iterations to make any progress. $\tau = 1000$ seems an optimal parameter choice but this method is slower than Gen-Oja by an order of magnitude.

**Robustness to incorrect step-size $\alpha$.** In the middle plot of Figure 1 we compare the behavior of Gen-Oja for step size $\alpha \in \{\alpha_*, \alpha_*/8, \alpha_*/16\}$ where $\alpha_* = 1/R^2$. We observe that Gen-Oja converges at a rate $O(1/t)$ independently of the choice of $\alpha$.

**Robustness to incorrect step-size $\beta_t$.** In the right plot of Figure 1 we compare the behavior of Gen-Oja for step size $\beta_t \in \{\beta_*/t, \beta_*/16t, \beta_*/\sqrt{i}, \beta_*/16\sqrt{i}\}$ where $\beta_*$ corresponds to the minimal error after one pass over the data. We observe that Gen-Oja is not robust to the choice of the constant

for step size $\beta_t \propto 1/t$. If the constant is too small, the rate of convergence is arbitrary slow. We observe that considering the streaming average of [29] on Gen-Oja with a step size $\beta_t \propto 1/\sqrt{t}$ enables to recover the fast $O(1/t)$ convergence while being robust to constant misspecification.

# 9    Conclusion

We have proposed and analyzed a simple online algorithm to solve the streaming generalized eigenvector problem and applied it to CCA. This algorithm, inspired by two-time-scale stochastic approximation achieves a fast $O(1/t)$ convergence. Considering recovering the $k$-principal generalized eigenvector (for $k > 1$) and obtaining a slow convergence rate $O(1/\sqrt{t})$ in the gap free setting are promising future directions. Finally, it would be worth considering removing the dimension dependence in our convergence guarantee.

# Acknowledgements

We gratefully acknowledge the support of the NSF through grant IIS-1619362. AP acknowledges Huawei's support through a BAIR-Huawei PhD Fellowship. This work was supported in part by the Mathematical Data Science program of the Office of Naval Research under grant number N00014-18-1-2764. This work was partially supported by AFOSR through grant FA9550-17-1-0308.

## Footnotes

[2]Note that we consider here the largest *signed* value of $\lambda_i$

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
