[Supplementary Material]

# Contents

# A  Proof of Lemma 6.2

We prove here the Lemma 6.2 which is an easy adaptation of [17, Lemma 3.1]. We first recall it.

**Lemma A.1.** *Let $H \in \mathbb{R}^{d \times d}$, $(u_i)_{i=1}^d$ and $(\tilde{u}_i)_{i=1}^d$ the corresponding right and left eigenvectors of $B^{-1}A$ and $w \in \mathbb{R}^d$ chosen uniformly on the sphere, then with probability $1 - \delta$ (over the randomness in the initial iterate)*

$$\sin_B^2(u_i, Hw) \leq \frac{C \log(1/\delta)}{\delta} \frac{\operatorname{Tr}(HH^\top \sum_{j \neq i} \tilde{u}_j \tilde{u}_j^\top)}{\tilde{u}_i^\top HH^\top \tilde{u}_i},$$

*for some universal constant $C > 0$.*

*Proof.* We follow the proof of [17]. Given a $B$-normalized right eigenvector $u_i$ of $B^{-1}A$ and $w = \frac{g}{\|g\|_2}$ for $g \sim \mathcal{N}(0, I)$, we consider:

$$\sin_B^2(u_i, Hw) = 1 - \frac{(u_i^\top BHw)^2)}{w^\top H^\top BHw} = \frac{g^\top H^\top B^{1/2} \left[ I - B^{1/2} u_i u_i^\top B^{1/2} \right] B^{1/2} Hg}{g^\top H^\top BHg}.$$

Moreover following Lemma G.3 and denoting by $\hat{u}_i$ the corresponding orthonormal family of eigenvectors of the symmetric matrix $B^{-1/2} A B^{-1/2}$, we have that $u_i = B^{-1/2} \hat{u}_i$. This yields:

$$\left[ I - B^{1/2} u_i u_i^\top B^{1/2} \right] = \left[ I - \hat{u}_i \hat{u}_i^\top \right] = \sum_{j \neq i} \hat{u}_j \hat{u}_j^\top$$

Using now that the left eigenvectors of $B^{-1}A$ are given by $\tilde{u}_i = Bu_i$, we get

$$\sin_B^2(u_i, Hw) = \frac{g^\top H^\top B^{1/2} \left[ \sum_{j \neq i} \hat{u}_j \hat{u}_j^\top \right] B^{1/2} Hg}{g^\top H^\top BHg} = \frac{g^\top H^\top \left[ \sum_{j \neq i} \tilde{u}_j \tilde{u}_j^\top \right] Hg}{g^\top H^\top BHg}.$$

We may bound the denominator by

$$g^\top H^\top BHg \geq g^\top H^\top B^{1/2} \hat{u}_i \hat{u}_i^\top B^{1/2} Hg = g^\top H^\top \tilde{u}_i \tilde{u}_i^\top Hg = (\tilde{u}_i^\top Hg)^2 \geq \frac{\delta}{C_1} \tilde{u}_i^\top HH^\top \tilde{u}_i,$$

where the last inequality follows as $\tilde{u}_i^\top Hg$ is a Gaussian random vector with variance $\|H^\top \tilde{u}_i\|_2^2$. We can also bound the numerator as

$$g^\top H^\top \left[ \sum_{j \neq i} \tilde{u}_j \tilde{u}_j^\top \right] Hg \leq C_2 \log(1/\delta) \operatorname{Tr}[H^\top \sum_{j \neq i} \tilde{u}_j \tilde{u}_j^\top H],$$

since $w^\top H^\top \left[ \sum_{j \neq i} \tilde{u}_j \tilde{u}_j^\top \right] Hw$ is a $\chi^2$ random variable with $\operatorname{Tr}[H^\top \sum_{j \neq i} \tilde{u}_j \tilde{u}_j^\top H]$ degrees of freedom. Therefore it exists a universal constant $C > 0$ such that

$$\sin_B^2(u_i, Hw) \leq C \frac{\log(1/\delta)}{\delta} \frac{\operatorname{Tr}[H^\top \sum_{j \neq i} \tilde{u}_j \tilde{u}_j^\top H]}{\tilde{u}_i^\top HH^\top \tilde{u}_i},$$

with probability $1 - \delta$. $\qquad\square$

# B  Deviation bounds for fast-mixing Markov Chain

In this section, we prove an upper bound on $\|\mathbb{E}[\epsilon_{t+k}|\mathcal{F}_t]\|_2$, where $\epsilon_t = (w_t - B^{-1}Aw_{t-1})v_{t-1}^\top$ and $\mathcal{F}_t = \sigma(w_0, \cdots, w_t)$ denotes the $\sigma$-algebra generated by $w_0, \cdots, w_t$. For the purpose of this section, we denote the pointwise upperbound on $\|w_t\|_2$ by $W_t$. To begin with, we consider bounding the error term considering a fixed step-size $\alpha_t = \alpha$ in order to keep the analysis cleaner. In Lemma B.4, we bound the deviation of chains with step-size $\alpha_t = O(c/\log(d^2\beta + t))$ and fixed step size over a short horizon of length $O(\log^2(1/\beta_t))$

In order to prove the requisite bound, consider the following Markov chain given by,

$$\theta_{k+1} = \theta_k - \eta[f'(\theta_{(k)}) + \epsilon_{k+1}], \tag{15}$$

where $f : \mathbb{R}^d \to \mathbb{R}$ is some strongly convex function. We make use of the following proposition highlighting the fast-mixing property of constant step-size stochastic gradient descent from [11].

**Proposition B.1.** *For any step size $\alpha \in (0, 2/L_\theta)$, the markov chain given by $(\theta_k)_{k \geq 0}$ defined by recursion* (15)*, admits a unique stationary distribution $\pi \in \mathcal{P}(\mathbb{R}^d)$. In addition, for all $\theta \in \mathbb{R}^d, k \in \mathbb{N}$, we have,*

$$W_2^2(R^k(\theta, \cdot), \pi) \leq (1 - 2\mu_\theta \eta (1 - \eta L_\theta/2))^k \int_{\mathbb{R}^d} \|\theta - \theta'\|_2^2 d\pi(\theta'), \tag{16}$$

*where $L_\theta$ and $\mu_\theta$ are the smoothness and the strong convexity parameters of $f$ respectively.*

Now, consider the Markov chain given by

$$w_t^{k+1} = w_t^k - \alpha(B_k w_t^k - A_k v_t), \tag{17}$$

where $\mathbb{E}[B_k] = B, \mathbb{E}[A_k] = A, w_t^0 = w_t$ where $w_t$ is as given by Algorithm 1. Equation (17) represents the update step for the $k^{th}$ step of a Markov chain starting at $w_t$ and performing stochastic gradient updates on $f_t(w) = 1/2 w^\top B w - w^\top A v_t$.

For this function $f_t$, the smoothness constant $L = \lambda_B$. Further, proposition B.1 guarantees the existence of a unique stationary distribution $\pi$ and we have that under the stationary distribution,

$$\mathbb{E}_\pi[w_t^k] = B^{-1} A v_t. \tag{18}$$

**Lemma B.2.** *For the Markov chain given by* (17) *with any step size $\alpha \in (0, 2/\lambda_B)$, for any $k > \frac{\log(\frac{\lambda_1}{\epsilon})}{\mu\alpha\left(1 - \frac{\alpha\lambda_B}{2}\right)}$, we have*

$$\|\mathbb{E}[w_t^k - B^{-1} A v_t]|\mathcal{F}_t\|_2^2 \leq \epsilon$$

*Proof.* We know from (18), $B^{-1} A v_t = \mathbb{E}_\pi[w_t^k]$. Now, we consider the term $\|\mathbb{E}[w_t^k - B^{-1} A v_t]|\mathcal{F}_t\|_2^2$,

$$\begin{aligned}
\|\mathbb{E}[w_t^k - B^{-1} A v_t]|\mathcal{F}_t\|_2^2 &= \|\mathbb{E}[w_t^k] - \mathbb{E}_\pi[w]|\mathcal{F}_t\|_2^2 \\
&= \|\mathbb{E}_{\Gamma(R^k(w_t, \cdot), \pi)}[w_t^k - w]\|_2^2 \\
&\overset{\zeta_1}{\leq} \mathbb{E}_{\Gamma(R^k(w_t, \cdot), \pi)}[\|w_t^k - w\|_2^2] \\
&\overset{\zeta_2}{=} W_2^2(R^k(w_t, \cdot), \pi) \\
&\overset{\zeta_3}{\leq} (1 - 2\mu\alpha(1 - \alpha\lambda_B/2))^k \lambda_1^2,
\end{aligned}$$

where $R^k(w_t, \cdot)$ denotes the $k$-step transition kernel of the Markov chain beginning from $w_t$, $\Gamma(R^k(w_t, \cdot), \pi)$ denotes any coupling of the distributions $R^k(w_t, \cdot)$ and $\pi$ and $\mathbb{E}_{\Gamma(\cdot, \cdot)}$ denotes the expectation under the joint distribution, conditioned on $\mathcal{F}_t$. Now, $\zeta_1$ follows from Jenson's inequality, $\zeta_2$ follows by setting $\Gamma(R^k(w_t, \cdot), \pi)$ to the coupling attaining the infimum in the wasserstein bound and $\zeta_3$ follows by using proposition (B.1). The lemma now follows by setting $k > \frac{\log(\frac{\lambda_1}{\epsilon})}{\mu\alpha\left(1 - \frac{\alpha\lambda_B}{2}\right)}$. [see, e.g., 30, for more properties of $W_2$] $\square$

**Deviation bound for** $\|v_t - v_{t+k}\|_2$: We now bound the deviation of $v_{t+k}$ from $v_t$ if we execute $k$ steps of the algorithm sarting from $v_t$,

$$\|v_t - v_{t+k}\|_2 \leq \sum_{i=0}^{k-1} \|v_{t+i} - v_{t+i+1}\|_2. \tag{19}$$

Now, for a single step of the algorithm, using the contractivity of the projection

$$\|v_i - v_{i+1}\|_2 \leq \|v_i - \frac{v'_{i+1}}{\|v'_{i+1}\|}\|_2 \leq \|v_i - v'_{i+1}\|_2 \leq W_{i+1}\beta_{i+1}.$$

Using the above bound in (19), we obtain,

$$\|v_t - v_{t+k}\|_2 \leq W_{t+k} \sum_{i=0}^{k-1} \beta_{t+i+1} \leq W_{t+k} k \beta_t, \tag{20}$$

by using the fact that $\beta_t$ is a decreasing sequence.

**Deviation bound for Coupled Chains**: Consider the sequence $(w_{t+i})_{i=0}^{k}$ as generated by Algorithm 1, assuming a constant step-size $\alpha$, and the sequence $(w_t^i)_{i=1}^{k}$ generated by the recurrence (17) in the case when both have the same randomness with respect to the sampling of the matrices $A_{t+i}, B_{t+i}$. We now obtain a bound on $\|\mathbb{E}[w_t^k - w_{t+k}]|\mathcal{F}_t\|_2$.

$$
\begin{aligned}
\|\mathbb{E}[w_t^k - w_{t+k}]|\mathcal{F}_t\|_2 &= \|\mathbb{E}\left[\mathbb{E}[(I - \alpha B_{t+k})(w_t^{k-1} - w_{t+k-1}) - \alpha A_{t+k}(v_t - v_{t+k-1})]\,|\mathcal{F}_{t+k-1}]|\mathcal{F}_t\right\|_2 \\
&= \|\mathbb{E}\left[(I - \alpha B)(w_t^{k-1} - w_{t+k-1}) - \alpha A(v_t - v_{t+k-1})\right]|\mathcal{F}_t\|_2 \\
&\vdots \\
&= \alpha \left\|\mathbb{E}\left[\sum_{i=0}^{k-1}(I - \alpha B)^i A(v_t - v_{t+k-1-i})|\mathcal{F}_t\right]\right\|_2 \\
&\leq \alpha\mathbb{E}\left[\sum_{i=0}^{k-1}\left\|(I - \alpha B)^i A(v_t - v_{t+k-1-i})\right\|_2 |\mathcal{F}_t\right] \\
&\leq \alpha\lambda_A W_{t+k} k \sum_{i=0}^{k-1}(1 - \alpha\mu)^i \beta_{t+k-1-i} \\
&\leq \frac{\lambda_A W_{t+k} k \beta_t}{\mu},
\end{aligned}
\tag{21}
$$

where we expand the terms using the recursion and bound the geometric series by using that $\alpha\mu \leq 1$.

**Lemma B.3.** *For any choice of $k > \frac{\log(\frac{1}{\beta_t})}{2\mu\alpha\left(1 - \frac{\alpha\lambda_B}{2}\right)}$, we have that*

$$
\|\mathbb{E}[\epsilon_{t+k}|\mathcal{F}_t]\|_2 \leq \left(\frac{\lambda_A W_{t+k} k}{\mu} + \lambda_1(1 + 2W_{t+k}k) + W_{t+k}^2 k\right)\beta_t = O(W_{t+k}^2 k\beta_t)
$$

*Proof.* Consider the term $\|\mathbb{E}[\epsilon_{t+k}|\mathcal{F}_t]\|_2$,

$$
\begin{aligned}
\|\mathbb{E}[\epsilon_{t+k}|\mathcal{F}_t]\|_2 &= \|\mathbb{E}[(w_{t+k} - B^{-1}Av_{t+k-1})v_{t+k-1}^\top|\mathcal{F}_t]\|_2 \\
&\leq \underbrace{\|\mathbb{E}[(w_{t+k} - B^{-1}Av_{t+k-1})v_t^\top|\mathcal{F}_t]\|_2}_{(I)} + \underbrace{\|\mathbb{E}[(w_{t+k} - B^{-1}Av_{t+k-1})(v_{t+k-1} - v_t)^\top|\mathcal{F}_t]\|_2}_{(II)}.
\end{aligned}
$$

We first analyze term (I) in the expansion above.

$$
\begin{aligned}
\|\mathbb{E}[(w_{t+k} - B^{-1}Av_{t+k-1})v_t^\top|\mathcal{F}_t]\|_2 &= \|\mathbb{E}[(w_{t+k} - w_t^k) + (w_t^k - B^{-1}Av_t) \\
&\quad + (B^{-1}Av_t - B^{-1}Av_{t+k-1}))v_t^\top|\mathcal{F}_t]\|_2 \\
&\leq \|\mathbb{E}[(w_{t+k} - w_t^k)|\mathcal{F}_t]v_t^\top\|_2 + \|\mathbb{E}[(w_t^k - B^{-1}Av_{t+k-1}))|\mathcal{F}_t]v_t^\top\|_2 \\
&\leq \|\mathbb{E}[(w_{t+k} - w_t^k)|\mathcal{F}_t]\|_2 + \|\mathbb{E}[(w_t^k - B^{-1}Av_t))|\mathcal{F}_t]\|_2 \\
&\quad + \|\mathbb{E}[(B^{-1}Av_t - B^{-1}Av_{t+k-1}))|\mathcal{F}_t]\|_2 \\
&\overset{\zeta_1}{\leq} \frac{\lambda_A W_{t+k} k}{\mu}\beta_t + \lambda_1\beta_t + \lambda_1 W_{t+k}k\beta_t \\
&= (\frac{\lambda_A W_{t+k} k}{\mu} + \lambda_1(1 + W_{t+k}k))\beta_t,
\end{aligned}
\tag{22}
$$

where $\zeta_1$ follows from using lemma B.2 with $k > \frac{\log(\frac{1}{\beta_t})}{2\mu\alpha\left(1 - \frac{\alpha\lambda_B}{2}\right)}$, bound in (20) and bound in (21).

We now look at term (II) in the expansion.

$$
\begin{aligned}
\|\mathbb{E}[(w_{t+k} - B^{-1}Av_{t+k-1})(v_{t+k-1} - v_t)^\top|\mathcal{F}_t]\|_2 &\leq (W_{t+k} + \lambda_1)\|v_{t+k-1} - v_t\| \\
&\leq W_{t+k}(W_{t+k} + \lambda_1)k\beta_t.
\end{aligned}
\tag{23}
$$

Combininig the bounds in (22) and (23), we get the desired result. $\qquad\square$

The bound we proved above hold for any fixed fixed step-size $\alpha$. However, in order to obtain the sharpest convergence result for our algorithm, we would require the step size $\alpha_t = \frac{c}{\log(d^2\beta+t)}$ for some constant $\beta$. We provide the following lemma which accomodates for this change.

In order to get a bound on the noise term with a logarithmically decaying step size, in addition to the previous analysis, we consider processes $(\hat{w}_{t+i})_{i=1}^k$ and $(\hat{v}_{t+i})_{i=1}^k$ which evolve with the same random matrices $A_{t+i}$ and $B_{t+i}$, but with a step size of $\alpha_{t+i} = \alpha_t = \frac{c}{\log(d^2\beta+t)}$.

**Pointwise bound on** $\|\hat{w}_{t+k}\|_2$: We can obtain a pointwise bound on $\|\hat{w}_{t+k}\|_2$ using the simple recursive evaluation:

$$\|\hat{w}_{t+k}\| \leq \|I - \alpha_t B_{t+k}\|_2 \|\hat{w}_{t+k-1}\|_2 + \alpha_t \lambda_A$$
$$\leq W_t + k\alpha_t \lambda_A, \tag{24}$$

where the final inequality follows from recursing on $\|\hat{w}_{t+k-1}\|$ and using the assumption that $B_i \succeq 0$.

**Deviation bound for** $\|v_{t+k} - \hat{v}_{t+k}\|_2$: We can obtain a bound on this quantity as follows:

$$\|v_{t+k} - \hat{v}_{t+k}\|_2 \leq \|v_{t+k} - v'_{t+k}\|_2 + \|\hat{v}'_{t+k} - \hat{v}_{t+k}\|_2 + \|v'_{t+k} - \hat{v}'_{t+k}\|_2$$
$$\leq 2\beta_{t+k} W_{t+k} + 2\beta_{t+k}\|\hat{w}_{t+k}\|_2 + \|\beta_{t+k}(w_{t+k} - \hat{w}_{t+k})\|_2 + \|v_{t+k-1} - \hat{v}_{t+k-1}\|_2$$
$$\leq 2\left(\sum_{i=1}^k \beta_{t+i}(W_{t+i} + \|\hat{w}_{t+i}\|_2)\right) + \sum_{i=1}^k \beta_{t+i}\|w_{t+i} - \hat{w}_{t+i}\|_2$$
$$\leq 3\beta_t k(2W_{t+k} + k\alpha_t \lambda_A), \tag{25}$$

where the final bound is obtained using $\|w_{t+k}\|_2 \leq W_{t+k}$ and $\|\hat{w}_{t+k}\|_2 \leq W_{t+k} + k\alpha_t \lambda_A$ from Equation (24)

**Lemma B.4.** *For any choice of* $k > \frac{\log(\frac{1}{\beta_t})}{2\mu\alpha_t\left(1-\frac{\alpha_t\lambda_B}{2}\right)}$ *and* $\alpha_t \in (0, 2/\lambda_B)$ *of the form* $\alpha_t = \frac{c}{\log(d^2\beta+t)}$, *we have that*

$$\|\mathbb{E}[\epsilon_{t+k}|\mathcal{F}_t]\|_2 \leq \left(\frac{\lambda_A W_{t+k} k}{\mu} + \lambda_1(1 + 2W_{t+k}k) + W_{t+k}^2 k\right)\beta_t$$
$$+ \frac{\lambda_B W_{t+k}k\alpha_t\beta_t}{c\mu\gamma} + \frac{\lambda_A k\alpha_t\beta_t}{c\mu\gamma} + \frac{3\lambda_A \beta_t k(2W_{t+k} + k\alpha_t\lambda_A)}{\mu}$$
$$+ (2W_{t+k} + k\alpha_t\lambda_A)W_{t+k}k\beta_t.$$

*In other words, we get that* $\|\mathbb{E}[\epsilon_{t+k}|\mathcal{F}_t]\|_2 = O(\beta_t k^2 \alpha_t W_{t+k})$.

*Proof.* In continuation from Lemma B.3, we consider bounding the deviation of the process $\hat{w}_{t+k}$ from the process $w_{t+k}$. The extra components in the error term $\epsilon_t$ remain the same and we ignore them for clarity of this lemma.

$$\|\mathbb{E}[(w_{t+k}-\hat{w}_{t+k})v_{t+k-1}^\top|\mathcal{F}_t]\|_2 \leq \underbrace{\|\mathbb{E}[(w_{t+k} - \hat{w}_{t+k})v_t^\top|\mathcal{F}_t]\|_2}_{(I)} + \underbrace{\|\mathbb{E}[(w_{t+k} - \hat{w}_{t+k})(v_{t+k-1} - v_t)^\top|\mathcal{F}_t]\|_2}_{(II)}$$
$$\tag{26}$$

We proceed by first analyzing term (I) in Equation (26).

$$\|\mathbb{E}[(w_{t+k} - \hat{w}_{t+k})v_t^\top|\mathcal{F}_t]\|_2 = \|\mathbb{E}[\mathbb{E}[((I - \alpha_{t+k}B_{t+k})w_{t+k-1} + \alpha_{t+k}A_{t+k}v_{t+k-1})$$
$$- ((I - \alpha_t B_{t+k})\hat{w}_{t+k-1} + \alpha_t A_{t+k}\hat{v}_{t+k-1})|\mathcal{F}_{t+k-1}]v_t^\top|\mathcal{F}_t]\|_2$$
$$= \|\mathbb{E}[((I - \alpha_{t+k}B)w_{t+k-1} + \alpha_{t+k}Av_{t+k-1})$$
$$- ((I - \alpha_t B)\hat{w}_{t+k-1} + \alpha_t A\hat{v}_{t+k-1})|\mathcal{F}_t]v_t^\top\|_2$$
$$= \|\mathbb{E}[(\alpha_t - \alpha_{t+k})Bw_{t+k-1} + (I - \alpha_t B)(w_{t+k-1} - \hat{w}_{t+k-1})$$
$$+ (\alpha_{t+k} - \alpha_t)Av_{t+k-1} + \alpha_t A(v_{t+k-1} - \hat{v}_{t+k-1}))|\mathcal{F}_t]v_t^\top\|_2$$
$$\leq \left\|\mathbb{E}\left[\sum_{i=1}^k (\alpha_t - \alpha_{t+i})(I - \alpha_t B)^{k-i}Bw_{t+i-1}|\mathcal{F}_t\right]v_t^\top\right\|_2$$

$$+ \left\| \mathbb{E}\left[ \sum_{i=1}^{k} (\alpha_{t+i} - \alpha_t)(I - \alpha_t B)^{k-i} A v_{t+i-1} | \mathcal{F}_t \right] v_t^\top \right\|_2$$

$$+ \alpha_t \left\| \mathbb{E}\left[ \sum_{i=1}^{k} (I - \alpha_t B)^{k-i} A (v_{t+i-1} - \hat{v}_{t+i-1}) | \mathcal{F}_t \right] v_t^\top \right\|_2$$

$$\leq \frac{(\alpha_t - \alpha_{t+k})\lambda_B W_{t+k}}{\alpha_t \mu} + \frac{(\alpha_t - \alpha_{t+k})\lambda_A}{\alpha_t \mu} + \frac{\lambda_A \|v_{t+k-1} - \hat{v}_{t+k-1}\|_2}{\mu}$$

$$\leq \frac{\lambda_B W_{t+k} k \alpha_t}{c\mu(d\beta + t)} + \frac{\lambda_A k \alpha_t}{c\mu(d\beta + t)} + \frac{3\lambda_A \beta_t k(2W_{t+k} + k\alpha_t \lambda_A)}{\mu}$$

$$\leq \frac{\lambda_B W_{t+k} k \alpha_t \beta_t}{c\mu b} + \frac{\lambda_A k \alpha_t \beta_t}{c\mu b} + \frac{3\lambda_A \beta_t k(2W_{t+k} + k\alpha_t \lambda_A)}{\mu} \quad (27)$$

where the second last inequality follows using Jensen's ineuality along with a trinagle inequality and using the fact that $B \succeq \mu I$ and the last equality follows from using the form of $\beta_t = \frac{b}{d^2\beta + t}$ for some constant $b$.

We now consider term (II) in Equation (26).

$$\|\mathbb{E}[(w_{t+k} - \hat{w}_{t+k})(v_{t+k-1} - v_t)^\top | \mathcal{F}_t]\|_2 \leq (2W_{t+k} + k\alpha_t\lambda_A)W_{t+k}k\beta_t, \quad (28)$$

by using Jensen's inequality along with bound (20). Combining (27) and (28) with (26), and using Lemma B.3, we obtain the desired result. □

Note that in order to prove the final convergence for Algorithm 1, we use the form of the step sizes $\alpha_t$ and $\beta_t$ as mentioned in this section.

In the following sections we denote by $r_t = \frac{1}{2\mu\alpha_t \left(1 - \frac{\alpha_t \lambda_B}{2}\right)} \log^2(\frac{1}{\beta_t})$ and $\mathcal{A}_t$ to be such that:

$$\mathcal{A}_t r_t \beta_t \geq \|\mathbb{E}\left[\epsilon_{t+k} | \mathcal{F}_t\right]\| \quad (29)$$

When $\alpha_t = \frac{c}{\log(d^2\beta + t)}$, $r_t$ will be $O(\log^3(1/\beta_t))$ and when $\alpha_t$ is contant, $r_t$ will be $O(\log^2(1/\beta_t))$.

## C    Controlling Markov Chain $w_t$

For the purpose of this section, we stick with bounds $R_A, R_B$ the maximum of which equals $R$ in the main paper. In this section we provide a bound on the norm of the markov chain $w_t$. We start by showing the $p$ moments of the norms of $w_t$ are bounded as long as $\alpha_t = \alpha$ a small enough constant $\forall t$. Ultimately we will use a time dependent $\alpha_t$ as defined in the previous section, but for warm up we start by showing some lemmas that bring out the behavior of $w_t$ when $\alpha_t = \alpha$ for all $t$. The proofs for a moving $\alpha_t$ will follow a similar though technically involved arguments.

**Lemma C.1.** *For $\alpha \leq 1/R_B^2$ we have*

$$\mathbb{E}[\|w_t\|_2^2] \leq \left[ (1 - \mu\alpha/2)^t \|w_0\|_2 + 2\frac{R_A^2}{\mu} \right]^2.$$

*If, in addition we assume that $\alpha \leq \frac{2}{R_B^2(p-2)}$ for $p \geq 3$ we have:*

$$\mathbb{E}\left[\|w_t\|_2^p\right] \leq \left[ (1 - \mu\alpha/4)^t \|w_0\|_2 + 4\frac{R_A^2}{\mu} \right]^p.$$

*Proof.* We first expand $w_{t+1} = (I - \alpha B_{t+1})w_t + \alpha A_{t+1}v_t$ and use the Minkowski inequality on $L_2$-norm (denoted by $\|\|_{L_2}$) to obtain:

$$\|w_{t+1}\|_{L_2} \leq \|(I - \alpha B_t)w_t\|_{L_2} + \|\alpha A_{t+1}v_t\|_{L_2}$$

We directly have that $\|\alpha A_{t+1}v_t\|_{L_2} \leq \alpha R_A^2$ almost surely and we can directly compute for $\alpha < 1/R_B^2$:

$$\|(I - \alpha B_{t+1})w_t\|_{L_2}^2 \quad = \quad \mathbb{E}[w_t^\top (I - \alpha B_{t+1})^2 w_t] = \mathbb{E}[w_t^\top (I - 2\alpha B_{t+1} + \alpha^2 B_{t+1}^2)w_t]$$

$$\overset{(1)}{\leq} \quad \mathbb{E}[w_t^\top (I - \alpha B_{t+1}) w_t] \leq \mathbb{E}[w_t^\top (I - \alpha \mathbb{E}[B_{t+1}|\mathcal{F}_{\sqcup}]) w_t] \leq (1 - \alpha\mu)\mathbb{E}[\|w_t\|_2^2],$$

where (1) follows as $B_{t+1} \preccurlyeq R_B^2 I$. We obtain expanding the recursion ( and using $\sqrt{1-x} \leq 1-x/2$ for $x \geq 0$):

$$\|w_t\|_{L_2} \leq (1 - \alpha\mu/2)^t \|w_0\|_{L_2} + \alpha R_A^2 \sum_{i=0}^{t-1} (1 - \alpha\mu/2)^i.$$

We conclude

$$\|w_t\|_{L_2} \leq (1 - \mu\alpha/2)^t \|w_0\|_{L_2} + 2\frac{R_A^2}{\mu}.$$

We consider now $p \geq 3$. We expand again $w_{t+1} = (I - \alpha B_{t+1})w_t + \alpha A_{t+1} v_t$ and use now the Minkowski inequality on $L_p$-norm on $(\mathbb{R}^d, \|\|_2)$ (denoted by $\|\|_{L_p}$ and defined by $\|x\|_{L_p} = (\mathbb{E}[\|x\|_2^p])^{1/p}$) to obtain:

$$\|w_{t+1}\|_{L_p} \leq \|(I - \alpha B_t)w_t\|_{L_p} + \|\alpha A_{t+1} v_t\|_{L_p}$$

We then compute for $\alpha < 1/R_B^2$

$$
\begin{aligned}
\|(I - \alpha B_{t+1})w_t\|_{L_p}^p &= \mathbb{E}[(w_t^\top (I - \alpha B_{t+1})^2 w_t)^{p/2}] = \mathbb{E}[(w_t^\top (I - 2\alpha B_{t+1} + \alpha^2 B_{t+1}^2)w_t)^{p/2}]\\
&\leq \mathbb{E}[(w_t^\top (I - \alpha B_{t+1})w_t)^{p/2}] \leq \mathbb{E}[\|w_t\|_2^p \left(1 - \alpha\frac{w_t^\top B_{t+1} w_t}{\|w_t\|_2^2}\right)^{p/2}]\\
&\overset{(1)}{\leq} \mathbb{E}[\|w_t\|_2^p \left(1 - p\alpha\frac{w_t^\top B_{t+1} w_t}{2\|w_t\|_2^2} + \alpha^2 \frac{p(p-2)}{8}\frac{(w_t^\top B_{t+1} w_t)^2}{\|w_t\|_2^4}\right)]\\
&\overset{(2)}{\leq} \mathbb{E}[\|w_t\|_2^p \left(1 - p\alpha\frac{w_t^\top B_{t+1} w_t}{2\|w_t\|_2^2} + \alpha^2 R_B^2 \frac{p(p-2)}{8}\frac{w_t^\top B_{t+1} w_t}{\|w_t\|_2^2}\right)]\\
&\leq \mathbb{E}[\|w_t\|_2^p \left(1 - \frac{p\alpha}{2}(1 - \alpha R_B^2 \frac{p-2}{4})\frac{w_t^\top B_{t+1} w_t}{\|w_t\|_2^2}\right)]\\
&\overset{(3)}{\leq} \mathbb{E}[\|w_t\|_2^p \left(1 - \frac{p\alpha}{2}(1 - \alpha R_B^2 \frac{p-2}{4})\mu\right)],
\end{aligned}
$$

where (1) follows as $(1-x)^p \leq (1 - px + p(p-1)/2x^2)$ for $x \in [0,1]$, (2) follows as $w_t^\top B_{t+1} w_t \leq R_B^2 \|w_t\|_2^2$ and (3) follows as $\mathbb{E}[B_{t+1}|\mathcal{F}_t] = B \succcurlyeq \mu I$. Then using $(1-x)^{1/p} \leq 1 - x/p$ for $x \geq 0$ yields

$$\|(I - \alpha B_{t+1})w_t\|_{L_p} \quad \leq \quad \|w_t\|_{L_p} \left(1 - \frac{\alpha}{2}(1 - \alpha R_B^2 \frac{p-2}{4})\mu\right).$$

Moreover

$$\|\alpha A_{t+1} v_t\|_{L_p} \leq \alpha R_A^2 \quad \text{a.s.}$$

And therefore

$$\|w_{t+1}\|_{L_p} \leq \|w_t\|_{L_p} \left(1 - \frac{\alpha}{2}(1 - \alpha R_B^2 \frac{p-2}{4})\mu\right) + \alpha R_A^2. \tag{30}$$

Let us denote by $\delta = \frac{\alpha}{2}(1 - \alpha R_B^2 \frac{p-2}{4})\mu$, then we directly obtain expanding the recursion:

$$\|w_t\|_{L_p} \leq (1 - \delta)^t \|w_0\|_{L_p} + \alpha R_A^2 \sum_{i=0}^{t-1} (1 - \delta)^i.$$

We conclude for $\alpha \leq \frac{2}{R_B^2(p-2)}$

$$\|w_t\|_{L_p} \leq (1 - \mu\alpha/4)^t \|w_0\|_{L_p} + 4\frac{R_A^2}{\mu}.$$

$\square$

As a corollary, we conclude that:

**Corollary C.1.1.** *If $p \geq 3$, $w_0$ is sampled from the unit sphere, and $\alpha$ satisfies $\alpha \leq \min(\frac{2}{R_B^2(p-2)}, \frac{4}{\mu})$ then:*

$$\mathbb{E}\left[\|w_t\|_2^p\right] \leq \left(1 + 4\frac{R_A^2}{\mu}\right)^p \tag{31}$$

We can leverage corollary C.1.1 to obtain the following control on the norms of $w_t$. As a warm up first we show that polynomial control on the norms of $w$ is possible.

**Lemma C.2.** *Let $\eta > 0$ and $b > 0$. If:*

$$p = \frac{1+a}{b}, \qquad c \geq \frac{\left(1 + 4\frac{R_A^2}{\mu}\right)}{\eta^{1/p}} \left(\sum_{j=1}^{\infty} \frac{1}{j^{1+a}}\right)^{1/p} \tag{32}$$

*Then whenever $\alpha \leq \min(\frac{2}{R_B^2(p-2)})$, we have that with probability $1 - \eta$, $\|w_t\| \leq ct^b$ for all $t \leq n$.*

*Proof.* By Corollary C.1.1 and Markov's inequality:

$$\Pr\left(\|w_t\|^p \geq c^p t^{bp}\right) \leq \frac{\mathbb{E}\left[\|w_t\|^p\right]}{c^p t^{bp}} \leq \left(\frac{1 + 4R_A^2/\mu}{c}\right)^p \frac{1}{t^{bp}} \leq \eta \left(\frac{1}{\sum_{j=1}^{\infty} \frac{1}{j^{1+a}}}\right) \frac{1}{t^{bp}}$$

The first inequality follows by Markov, the second by Corollary C.1.1 and the third by the definition of $c$, and $p$. Applying the union bound to all $w_t$ from $t = 1$ to $\infty$ yields the desired result. □

The lemma above implies that for any probability level $\eta$, whenever the step size $\alpha_t$ is a small enough constant, independent of time $t$, by picking $\alpha$ small enough, we can show pointwise control on the norms of $\|w_t\|$ with constant probability so that at time $t$, $\|w_t\| \leq ct^b$.

Notice that for a fixed $a$, $\sum_{j=1}^{\infty} \frac{1}{j^{1+a}}$ converges, and that in case $a \geq 1$, $\sum_{j=1}^{\infty} \frac{1}{j^{1+a}} < 10$ (an absolute constant).

We now proceed to show that in fact for any $\delta > 0$, there is a constant $\mathcal{C}(\delta, \mu, R_B, R_A, \log(d))$ such that with probability $1 - \delta$, $w_t < \mathcal{B}(\delta, \mu, R_B, R_A, \log(d))$ for all $t$ whenever the step size is $\alpha_t = \frac{c}{\log(d^2\beta+t)}$ with $\beta \geq 0$.

We start with the following observation:

**Lemma C.3.** *Let $t_0 \in \mathbb{N}$ and $t_1 = 2t_0$. Assume $\|w_{t_0}\| \leq \mathcal{B}$. Then for all $t_0 + k \in [t_0 + \frac{8\log(\mathcal{B})\log(d^2\beta+t_0)}{\mu c}, \cdots, t_1]$, the following holds:*

$$\mathbb{E}\left[\|w_{t_0+k}\|^{c_1\log(t_1)}\right] \leq (1 + \frac{8R_A^2}{\mu})^{c_1\log(t_1)}$$

*Where $\alpha_{t_0+k} = \frac{c}{\log(d^2\beta+t_0+k)}$, $t_0 \geq 2$. And $c, c_1$ are positive constants such that $c \leq \frac{1}{R_B^2 c_1}$.*

*Proof.* Mimicking the proof of Lemma C.1, the same result of said Lemma holds up to Equation 30 even if the step size $\alpha_{t_0+m} = \frac{c}{\log(d^2\beta+t_0+m)}$, therefore for any $m$:

$$\|w_{t_0+m+1}\|_{L_p} \leq \|w_{t_0+m}\|_{L_p} \left(1 - \frac{\alpha_{t_0+m}}{2}\left(1 - \alpha_{t_0+m}R_B^2\frac{p-2}{4}\right)\mu\right) + \alpha_{t_0+k}R_A^2$$

Let $\delta_{t_0+m} = \frac{\alpha_{t_0+m}}{2}\left(1 - \alpha_{t_0+m}R_B^2\frac{p-2}{4}\right)\mu$, we obtain the recursion:

$$\|w_{t_0+m+1}\|_{L_p} \leq \|w_{t_0+m}\|_{L_p}(1 - \delta_{t_0+m}) + \alpha_{t_0+m}R_A^2$$

Which for any $k$ can be expanded to:

$$\|w_{t_0+k}\|_{L_p} \leq \prod_{m=0}^{k-1}(1 - \delta_{t_0+m})\|w_{t_0}\|_{L_p} + R_A^2 \sum_{m'=0}^{k-1}\alpha_{t_0+m'}\prod_{j=m'+1}^{k-1}(1 - \delta_{t_0+j})$$

We now show that we can substitute all instances of $\delta_{t_0+k}$ in the upper bound with a fixed quantity, which will allow us to bound the whole expression afterwards.

Notice that $\alpha_{t_0+k}$ is decreasing and that $\delta_{t_0+k} \geq \frac{\alpha_{t_1}}{2}\left(1 - 2\alpha_{t_1}R_B^2\frac{p-2}{4}\right)\mu$. The later follows because by assumption $\alpha_{t_0+k} = \frac{c}{\log(d^2\beta+t_0+k)} \leq 2\frac{c}{\log(d^2\beta+t_1)} = 2\alpha_{t_1}$ (recall that $t_1 = 2t_0$, implying this is true as long as $t_0 \geq 2$) and therefore $\alpha_{t_1} \leq \alpha_{t_0+k} \leq 2\alpha_{t_1}$.

Define $\delta'_{t_1} := \frac{\alpha_{t_1}}{2}\left(1 - 2\alpha_{t_1}R_B^2\frac{p-2}{4}\right)\mu$. As a consequence:

$$\|w_{t_0+k}\|_{L_p} \leq \prod_{i=0}^{k-1}(1-\delta'_{t_1})\|w_{t_0}\|_{L_p} + 2R_A^2\alpha_{t_1}\sum_{m'=0}^{k-1}(1-\delta'_{t_1})^{m'}$$

$$\leq \prod_{i=0}^{k-1}(1-\delta'_{t_1})\|w_{t_0}\|_{L_p} + 2R_A^2\alpha_{t_1}\frac{1}{\delta'_{t_1}}$$

$$= (1-\delta'_{t_1})^k\|w_{t_0}\|_{L_p} + 2R_A^2\alpha_{t_1}\frac{1}{\delta'_{t_1}}$$

If $\alpha_{t_1} < \frac{1}{R_B^2(p-2)}$, then $\delta'_{t_1} > \frac{\alpha_{t_1}}{4}\mu$. Then:

$$\|w_{t_0+k}\|_{L_p} \leq (1-\mu\alpha_{t_1}/4)^k\|w_{t_0}\|_{L_p} + 8\frac{R_A^2}{\mu}$$

And therefore:

$$\mathbb{E}\left[\|w_{t_0+k}\|^p\right] \leq \left((1-\mu\alpha_{t_1}/4)^k\|w_{t_0}\|_{L_p} + 8\frac{R_A^2}{\mu}\right)^p$$

Notice that $(1-\mu\alpha_{t_1}/4)^k \leq \exp(-\frac{\mu\alpha_{t_1}k}{4})$ and therefore $(1-\mu\alpha_{t_1}/4)^k\|w_{t_0}\|_{L_p} \leq 1$ whenever $-\mu\alpha_{t_1}k/4 + \log(\mathcal{B}) \leq 0$. Since $2\log(d^2\beta+t_0) \geq \log(d^2\beta+t_1)$ (because $t_0 \geq 2$), the relationship $(1-\mu\alpha_{t_1}/4)^k\|w_{t_0}\|_{L_p} \leq 1$ holds (at least) whenever $k \geq \frac{8\log(\mathcal{B})\log(d^2\beta+t_0)}{\mu c}$.

Recall that $p = c_1\log(t_1)$. Since the above conditions require $\alpha_{t_1} < \frac{1}{R_B^2(p-2)}$ to hold, it is enough to ensure that:

$$\alpha_{t_1} = \frac{c}{\log(d^2\beta+t_1)} \leq \frac{1}{R_B^2 p} = \frac{1}{R_B^2 c_1\log(t_1)} < \frac{1}{R_B^2(p-2)} = \frac{1}{R_B^2(c_1\log(t_1)-2)}$$

It is enough to take $c \leq \frac{1}{R_B^2 c_1}$ to satisfy the bound. Putting all these relationships together:

$$\mathbb{E}\left[\|w_{t_0+k}\|^p\right] \leq \left(1 + 8\frac{R_A^2}{\mu}\right)^p$$

For $p = c_1\log(t_1)$ and for all $k$ such that $k \in [\frac{8\log(\mathcal{B})\log(d^2\beta+t_0)}{\mu c}, \cdots, t_0]$. $\qquad\square$

As a consequence of Lemma C.3, we have the following corollary:

**Corollary C.3.1.** *Let $t_0 \in \mathbb{N}$ and $t_1 = 2t_0$. Assume $\|w_{t_0}\| \leq \mathcal{B}$. Then for all $t_0 + k \in [t_0 + \frac{8\log(\mathcal{B})\log(d^2\beta+t_0)}{\mu c}, \cdots, t_1]$, the following holds:*

$$\mathbb{E}\left[\|w_{t_0+k}\|^{c_1\log(t_0+k)}\right] \leq (1 + \frac{8R_A^2}{\mu})^{c_1\log(t_0+k)}$$

*Where $\alpha_{t_0+k} = \frac{c}{\log(d^2\beta+t_0+k)}$, $t_0 \geq 2$. And $c, c_1$ are positive constants such that $c \leq \frac{1}{R_B^2 c_1}$.*

The proof of this result follows the exact same template as the proof of Lemma C.3, the only difference is the subtitution of $p$ with the desired $c_1\log(t_0 + k)$ wherever necessary.

Now we proceed to show that having control up to the $c_1\log(t)$ moments for $\|w_t\|$ implies boundedness of $w_t$ with high probability:

**Lemma C.4.** *Assume $\mathbb{E}\left[\|w_t\|^{c_1\log(t)}\right] \leq (1 + 8\frac{R_A^2}{\mu})^{c_1\log(t)}$, and $\delta > 0$, then for $\mathcal{B} \geq 2\left(1 + \frac{8R_A^2}{\mu}\right)\frac{1}{\delta}$, we have:*

$$\Pr\left(\|w_t\| \geq \mathcal{B}\right) \leq \frac{1}{t^{c_1}}\delta^{c_1\log(t)}$$

*Where $\log$ is base $2$.*

*Proof.* The proof follows from a simple application of Markov's inequality:

$$\Pr\left(\|w_t\| \geq \mathcal{B}\right) \leq \Pr\left(\|w_t\|^{c_1 \log(t)} \geq \mathcal{B}^{c_1 \log(t)}\right)$$

$$\leq \frac{1}{t^{c_1}} \delta^{c_1 \log(t)}$$

This concludes the proof. $\qquad\square$

We now show that if there is $t_0$ for which $\|w_t\| \leq \mathcal{B}$, for some large enough constant $\mathcal{B}$, then by leveraging Lemmas C.3 and C.4 then we can say that with any constant probability a large chunk of the $w_t$ are bounded provided $\alpha$ is time dependent $\alpha_t$ with $\alpha_t = \frac{c}{\log(d^2\beta+t)}$ for some constant $c$.

**Lemma C.5.** *Let $\delta > 0$, define $\eta := \frac{\sum_{j=1}^{\infty} \frac{1}{j^2}}{\delta}$, and let the step size $\alpha_t = \frac{c}{\log(d^2\beta+t)}$ with $c > 0$ satisfying $c \leq \frac{1}{2R_B^2}$. Assume there exists $t_0 \geq 2$ such that $\|w_{t_0}\| \leq \mathcal{B}$ with $\mathcal{B} \geq 2\left(1 + \frac{8R_A^2}{\mu}\right)\eta$. Define $t_1 = 2t_0$ and $t_{i+1} = 2t_i$ for all $i \geq 1$. With probability $1 - \delta$ it holds that for all $t \geq t_0$ such that $t \in [t_i + \frac{2\log(\mathcal{B})\log(d^2\beta+t_i)}{\mu}2R_B^2, \cdots, t_{i+1}]$ it follows that:*

$$\|w_t\| \leq \mathcal{B}$$

*Proof.* The proof is a simple application of Lemmas C.3 and C.4. Indeed, by Lemma C.3 and the assumptions on $w_{t_0}$ and the step size, conditioning on the event that $w_{t_0} \leq \mathcal{B}$, the $2\log(t_1)$ moments (and in fact the $2\log(t)$ moments as well) of $\|w_t\|$ for $t \in [t_0 + \frac{2\log(\mathcal{B})\log(d^2\beta+t_0)}{\mu}2R_B^2, \cdots, t_1]$ are bounded by $(1 + \frac{8R_A^2}{\mu})^{2\log(t_1)}$ (respectively $(1 + \frac{8R_A^2}{\mu})^{2\log(t)}$ for the $2\log(t)$ moments). This in turn implies by Lemma C.4, that conditional on $\|w_{t_0}\| \leq \mathcal{B}$, for any $t \in [t_0 + \frac{2\log(\mathcal{B})\log(d^2\beta+t_0)}{\mu}2R_B^2, \cdots, t_1]$ the probability that $\|w_t\|$ is larger than $\mathcal{B}$ is upper bounded by $\frac{1}{t^2}\frac{1}{\eta^{2\log(t)}} \leq \frac{1}{t^2}\frac{\delta}{\sum_{j=1}^{\infty}\frac{1}{j^2}}$ (this inequality follows because $\eta \geq 1$ and $2\log(t) \geq 1$ as well). Consequently, the probability that any $\|w_t\| > \mathcal{B}$ for $t \in [t_0 + \frac{2\log(\mathcal{B})\log(d^2\beta+t_0)}{\mu}2R_B^2, \cdots, t_1]$ can be bounded by the union bound as:

$$\frac{\delta}{\sum_{j=1}^{\infty}\frac{1}{j^2}} \sum_{t\in[t_0 + \frac{2\log(\mathcal{B})\log(d^2\beta+t_0)}{\mu}2R_B^2, \cdots, t_1]} \frac{1}{t^2}$$

Conditioning on $\|w_{t_1}\| \leq \mathcal{B}$ and repeating the argument, for all $i$, we obtain that the probability that there is any $t$ such that $\|w_t\| > \mathcal{B}$ and $t \in [t_i + \frac{2\log(\mathcal{B})\log(d^2\beta+t_i)}{\mu}2R_B^2, \cdots, t_{i+1}]$ is at most:

$$\frac{\delta}{\sum_{j=1}^{\infty}\frac{1}{j^2}} \sum_{i=0}^{\infty} \sum_{t\in[t_i + \frac{2\log(\mathcal{B})\log(d^2\beta+t_i)}{\mu}2R_B^2, \cdots, t_{i+1}]} \frac{1}{t^2} \leq \delta$$

This concludes the proof. $\qquad\square$

Now we show that in fact, for any $\delta \in (0,1)$, then, with probability $1 - \delta$, for all $t$, all $w_t$ are bounded (by a quantity that depends inversely on $\delta$). More formally:

**Lemma C.6.** *Define $R_A$ and $R_B$ such that $R_A = R_B \geq \frac{1}{2}$. Let*

$$\mathcal{B} = \max\left(1 + \frac{1}{R_B}, (1 + \frac{8R_A^2}{\mu})\frac{\sum_{j=1}^{\infty}\frac{1}{j^2}}{\delta}, 2, (5 + 72 \cdot \frac{\log^2(1+d^2\beta)R_B^3}{\mu^2})^2\right).$$

*If $\alpha_t = \frac{c}{\log(d^2\beta+t)}$ with $c = \frac{1}{2R_B^2}$ and $\|w_0\| = 1$, then with probability $1 - \delta$ for all $t$:*

$$\|w_t\| \leq \mathcal{B} + \frac{2\log(\mathcal{B})R_B}{\mu} := \mathcal{C}(\delta, \mu, R_B, R_A, \log(d))$$

*Proof.* Let $t_0 = \max(\left(\frac{4}{3} * \frac{4\log(1+d^2\beta)\log(\mathcal{B})R_B^2}{\mu}\right)^2, 2)$. Define $t_1 = 2t_0$ and in general for all $i \geq 1$, $t_i = 2t_{i-1}$.

- We start by showing that $t_0 \geq 4\frac{\log(\mathcal{B})\log(d^2\beta+t_0)R_B^2}{\mu}$, which will allow us to show that the interval $[t_0 + 4\frac{\log\log(\mathcal{B})\log(d^2\beta+t_0)R_B^2}{\mu}, \cdots, t_1]$ is nonempty.

First notice that for all $t \geq 1$, (and in particular for all $t \geq 2$), we have that:

$$\frac{t}{\log_2(t)} \geq \frac{3}{4}t^{\frac{1}{2}}$$

Therefore:

$$\frac{t_0}{\log(t_0)} \geq \frac{3}{4}t_0^{1/2} \geq \max(\left(\frac{4\log(1+d^2\beta)\log(\mathcal{B})R_B^2}{\mu}\right), 1) \geq \frac{4\log(1+d^2\beta)\log(\mathcal{B})R_B^2}{\mu}$$

And therefore, since $\log(t_0)\log(1+d^2\beta) \geq \log(d^2\beta+t_0)$:

$$t_0 \geq \frac{4\log(t_0)\log(1+d^2\beta)\log(\mathcal{B})R_B^2}{\mu} \geq \frac{4\log(d^2\beta+t_0)\log(\mathcal{B})R_B^2}{\mu}$$

Which implies the desired inequality.

- Now we see that $\|w_t\| \leq \mathcal{B}$ for all $t \leq t_0$.

We use a very rough bound on $w_t$. Recall that $w_t = (I - \alpha_{t-1}B_t)w_{t-1} + \alpha_{t-1}A_t v_t$. The following sequence of inequalities holds:

$$\|w_t\| \leq \|I - \alpha_{t-1}B_t\|\|w_{t-1}\| + \frac{1}{2R_B^2}\|A_t\|$$

$$\leq \|w_{t-1}\| + \frac{1}{2R_B}$$

This holds as long as $\|I - \alpha_{t-1}B_t\| \leq 1$, which is true since by assumption $B_t \succeq 0$ for all $t$ and therefore $\|\alpha_{t-1}B_t\| \leq \frac{1}{2R_B^2}R_B = \frac{1}{2R_B} \leq 1$. The last inequality follows because $R_B \geq \frac{1}{2}$. Consequently, $\|w_t\| \leq 1 + \frac{t}{2R_B}$ for $t \leq t_0$. We want to ensure $\mathcal{B} \geq 1 + \frac{t_0}{2R_B}$. Notice that:

$$1 + \frac{t_0}{2R_B} = 1 + \frac{\max(\left(\frac{4}{3}\frac{4\log(1+d^2\beta)\log(\mathcal{B})R_B^2}{\mu}\right)^2, 1)}{2R_B}$$

If $t_0 = 1$, this provides the condition $\mathcal{B} \geq 1 + \frac{1}{R_B}$. When the max defining $t_0$ is achieved at $\left(\frac{4}{3}\frac{4\log(1+d^2\beta)\log(\mathcal{B})R_B^2}{\mu}\right)^2$, we obtain the condition:

$$1 + \left(\frac{4^4}{2*3^2}\frac{\log^2(1+d^2\beta)\log^2(\mathcal{B})R_B^3}{\mu^2}\right) \leq \mathcal{B} \tag{33}$$

Since we already have $\mathcal{B} \geq 2$, it follows that $\log(\mathcal{B}) \geq 1$. And therefore, Equation 33 is satisfied as long as:

$$\log^2(\mathcal{B})\left(1 + \left(\frac{4^4}{2*3^2}\frac{\log^2(1+d^2\beta)R_B^3}{\mu^2}\right)\right) \leq \mathcal{B}$$

Notice that for all $x \geq 1$:

$$\frac{x}{\log^2(x)} \geq \frac{1}{5}x^{1/2}$$

Therefore, picking $\mathcal{B} \geq (5 + 72 \cdot \frac{\log^2(1+d^2\beta)R_B^3}{\mu^2})^2 \geq (5 + 5\frac{4^4}{2*3^2} \cdot \frac{\log^2(1+d^2\beta)R_B^3}{\mu^2})^2$ guarantees that Equation 33 is satisfied, (since $\mathcal{B}$ is also greater than 1).

- We can therefore invoke Lemma C.5 to the sequence $\{t_i\}$ and conclude that with probability $1 - \delta$ for all $t$ such that $t \in [t_i + \frac{4 \log(\mathcal{B}) \log(d^2 \beta + t_i) R_B^2}{\mu}, \cdots, t_{i+1}]$ for some $i$, we have $\|w_t\| \leq \mathcal{B}$ simultaneously for all such $t$. This uses the fact that $\mathcal{B} \geq \left(1 + \frac{8R_A^2}{\mu}\right)^{\frac{\sum_{j=1}^{\infty} \frac{1}{j^2}}{\delta}}$.

- The final step is to show a bound on $w_t$ for the remaining blocks.

For the remaining blocks notice that if $\|w_{t_i}\| \leq \mathcal{B}$, then by a crude bound since $\alpha_t = \frac{c}{\log(d^2 \beta + t)}$, with $c = \frac{1}{2R_B^2}$, at each step starting from $t_i$, $w_t$ grows by at most an additive $\frac{1}{\log(d^2 \beta + t_i)}$ factor:

$$\|w_t\| \leq \|I - \alpha_{t-1} B_t\| \|w_{t-1}\| + \frac{1}{2R_B^2 \log(d^2 \beta + t_i)} \|A_t\|$$

$$\leq \|w_{t-1}\| + \frac{1}{2R_B \log(d^2 \beta + t_i)}$$

For all $t \in [t_i + 1, \cdots, t_i + \frac{2 \log(\mathcal{B}) \log(d^2 \beta + t_i)}{\mu} 2R_B^2]$.

Since $\|w_{t_i}\| \leq \mathcal{B}$, we have that $\|w_t\| \leq \mathcal{B} + \frac{2 \log(\mathcal{B}) R_B}{\mu}$ for all $t \in [t_i + 1, \cdots, t_i + \frac{2 \log(\mathcal{B}) \log(d^2 \beta + t_i)}{\mu} 2R_B^2]$.

As desired. $\qquad\square$

**Notation for following sections**: Throughout the following sections we use the following notation:

We use the assumption that $\|\mathbb{E}[\epsilon_t | \mathcal{F}_{t-r_t}]\| \leq \mathcal{A}_t r_t \beta_t$ as proved in Section B where $r_t$ is the mixing time window at time $t$.

Also, as proved in Section C, we have that $\|w_t\| \leq W_t$ and consequently:

$$\|\epsilon_t\| \leq \|w_t - B^{-1} A v_{t-1}\| \leq W_t + \|B^{-1} A\| := B_{\epsilon_t} \qquad (34)$$

Additionally we also have that:
$$\|G_t\| \leq \lambda_1 + B_{\epsilon_t} := \mathcal{G}_t$$

Notice that $B_{\epsilon_t}$ and $\mathcal{G}_t$ are of the same order.

# D  Analysis burn in times

In order to provide a convergence analysis for Algorithm 1, we use Lemma A.1 and bound each of the terms appearing in it. To obtain those bounds, we use a mixing time argument that allows us to bound the expected error accumulated by terms of the form $\beta_t \left(\epsilon_t H_{t-1} H_{t-1}^\top + H_{t-1} H_{t-1}^\top \epsilon_t\right)$.

To control terms of this kind we deal with the set $\{t\}$ such that $t \geq r_t$ and the set of $\{t\}$ such that $t < r_t$ differently. Let $t_0 = \max t$ such that $t < r_t$. This value $t_0$ is finite because $r_t$ grows polylogarithmically.

Recall that $r_t = O(\log^3(\frac{1}{\beta_t}))$ where $\beta_t = \frac{b}{d^2 \beta + 1}$. We define $r_t := \log^3(\frac{1}{\beta_t}) \mathcal{C}_r$. Where $\mathcal{C}_r$ is a constant capturing all the missing dependencies between $r_t$ and $A, B$. Let's start with an auxiliary lemma:

**Lemma D.1.** *Let $c > 0$ be some constant. If $x \geq 6!c$ then, $x^{\frac{1}{3}} \geq \log(cx)$.*

*Proof.* Observe that $x^{\frac{1}{3}} \geq \log(cx)$ iff $\exp(x^{\frac{1}{3}}) \geq cx$. Let's write the left hand side using its taylor series:

$$\exp(x^{\frac{1}{3}}) = \sum_{i=0}^{\infty} \frac{x^{\frac{i}{3}}}{i!}$$

Notice that $\sum_{i=0}^{\infty} \frac{x^{\frac{i}{3}}}{i!} \geq \frac{x^2}{6!}$, which in turn implies that if $\frac{x^2}{6!} \geq cx$ and therefore $x \geq 6!c$, then $\exp(x^{\frac{1}{3}}) \geq cx$, as desired. $\qquad\square$

We provide an upper bound for $t_0$:

**Lemma D.2.** *The breakpoint $t_0$ satisfies:*

$$t_0 = \max(\mathcal{B}_1(b, \mathcal{C}_r), \mathcal{C}_r \left(\log(d^2\beta) + \log(b) - 1\right)^3)$$

*Where $\mathcal{B}_1(b, \mathcal{C}_r) := 1440\frac{\mathcal{C}_r^2}{b}$ is a constant dependent only on $b$ and $\mathcal{C}_r$.*

*Proof.* We would like to show $t_0$ satisfies the property that for all $t \geq t_0$, it follows that $t \geq \mathcal{C}_r \log^3(\frac{1}{\beta_t})$. This is true iff $t^{\frac{1}{3}} - \mathcal{C}_r^{\frac{1}{3}} \log(\frac{1}{\beta_t}) \geq 0$. The following sequence of equalities holds:

$$t^{\frac{1}{3}} - \mathcal{C}_r^{\frac{1}{3}} \log(\frac{1}{\beta_t}) = t^{\frac{1}{3}} - \mathcal{C}_r^{\frac{1}{3}} \log(d^2\beta + t) + \log(b)\mathcal{C}_r^{\frac{1}{3}}$$

$$= t^{\frac{1}{3}} - \mathcal{C}_r^{\frac{1}{3}} \log(\frac{d^2\beta + t}{t}) - \mathcal{C}_r^{\frac{1}{3}} \log(t) + \log(b)\mathcal{C}_r^{\frac{1}{3}}$$

$$= t^{\frac{1}{3}} - \mathcal{C}_r^{\frac{1}{3}} \log(\frac{d^2\beta}{t} + 1) - \mathcal{C}_r^{\frac{1}{3}} \log(t) + \log(b)\mathcal{C}_r^{\frac{1}{3}}$$

We now massage this expression by considering two cases and making use of the following inequality: For $\log(1 + x) \leq \log(x) + 1$ if $x \geq 1$

Case 1 : $t \geq d^2\beta$

This implies that $\log(\frac{d^2\beta}{t} + 1) \leq \log(1 + 1) = 1$. The following inequalities hold:

$$t^{\frac{1}{3}} - \mathcal{C}_r^{\frac{1}{3}} \log(\frac{d^2\beta}{t} + 1) - \mathcal{C}_r^{\frac{1}{3}} \log(t) + \log(b)\mathcal{C}_r^{\frac{1}{3}} \geq t^{\frac{1}{3}} - \mathcal{C}_r^{\frac{1}{3}} - \mathcal{C}_r^{\frac{1}{3}} \log(t) + \log(b)\mathcal{C}_r^{\frac{1}{3}}$$

$$= t^{\frac{1}{3}} - \mathcal{C}_r^{\frac{1}{3}} \left(1 - \log(b) + \log(t)\right)$$

$$= t^{\frac{1}{3}} - \mathcal{C}_r^{\frac{1}{3}} \left(\log\left(\frac{2}{b}\right) + \log(t)\right)$$

$$= t^{\frac{1}{3}} - \mathcal{C}_r^{\frac{1}{3}} \left(\log\left(\frac{2t}{b}\right)\right)$$

Let $t = \mathcal{C}_r h$. Substituting into the previous equation, we would like to find a condition for $h$ such that $t^{\frac{1}{3}} - \mathcal{C}_r^{\frac{1}{3}} \left(\log\left(\frac{2t}{b}\right)\right) = \mathcal{C}_r^{\frac{1}{3}} \left(h^{\frac{1}{3}} - \log(\frac{2\mathcal{C}_r h}{b})\right) \geq 0$. This follows as long as $h \geq 6! \frac{2\mathcal{C}_r}{b} = 1440\frac{\mathcal{C}_r}{b}$ by Lemma D.1. Let $\mathcal{B}_1(b, \mathcal{C}_r) = 1440\frac{\mathcal{C}_r^2}{b}$.

We conclude that as long as we have $t \geq \mathcal{B}_1(b, \mathcal{C}_r)$ for some constant $\mathcal{B}_1(b, \mathcal{C}_r)$ depending on $\gamma$ and $\mathcal{C}_r$, we can guarantee that $t^{\frac{1}{3}} - \mathcal{C}_r^{\frac{1}{3}} \left(\log\left(\frac{2t}{b}\right)\right) \geq 0$.

Case 2 : $t < d^2\beta$.

This implies that $\log(\frac{d^2\beta}{t} + 1) \leq \log(\frac{d^2\beta}{t}) + 1$. The following inequalities hold:

$$t^{\frac{1}{3}} - \mathcal{C}_r^{\frac{1}{3}} \log(\frac{d^2\beta}{t} + 1) - \mathcal{C}_r^{\frac{1}{3}} \log(t) + \log(b)\mathcal{C}_r^{\frac{1}{3}} \geq t^{\frac{1}{3}} - \mathcal{C}_r^{\frac{1}{3}} \log(\frac{d^2\beta}{t}) - \mathcal{C}_r^{\frac{1}{3}} - \mathcal{C}_r^{\frac{1}{3}} \log(t) + \log(b)\mathcal{C}_r^{\frac{1}{3}}$$

$$= t^{\frac{1}{3}} - \mathcal{C}_r^{\frac{1}{3}} \log(d^2\beta) - \mathcal{C}_r^{\frac{1}{3}} + \log(b)\mathcal{C}_r^{\frac{1}{3}}$$

And therefore the last expression is greater than zero if $t \geq \mathcal{C}_r \left(\log(d\beta) + \log(b) - 1\right)^3$. As a consequence we get that as long as $t \geq t_0 = \max(\mathcal{B}_1(b, \mathcal{C}_r), \mathcal{C}_r \left(\log(d\beta) + \log(b) - 1\right)^3)$ we have that $t \geq \mathcal{C}_r \log^3(\frac{1}{\beta_t})$ as desired. □

Throughout the next sections, we use $t_0$ to denote this breakpoint.

# E   Analysis for Gen-Oja

In this section, we provide bounds on expectations of various terms appearing in Lemma A.1 which are required to obtain a convergence bound for Gen-Oja.

## E.1   Upper Bound on Operator Norm of $\mathbb{E}\left[H_t H_t^\top\right]$

We start by showing an upper bound for $\|\mathbb{E}\left[H_t H_t^\top\right]\|$.

**Lemma E.1.** *For all $t \geq 0$:*

$$\|\mathbb{E}\left[H_t H_t^\top\right]\| \leq \exp\left(2\sum_{i=1}^{t} \beta_i \lambda_1 + \beta_i^2 dr_i(\mathcal{A}_i + B_{\epsilon_i}\mathcal{G}_i + B_{\epsilon_i}^2)\mathcal{C}^{(1)} + \sum_{j=1}^{t_0} \beta_j 2dB_{\epsilon_j}\right)$$

*Where $\mathcal{C}^{(1)}$ is a constant. Assuming that for all $t \geq 0$, $\beta_t r_t \mathcal{G}_t < \frac{1}{4}$.*

*Proof.* We start by substituting the identity: $H_t = (I + \beta_t G_t)H_{t-1} = (I + \beta_t B^{-1}A + \beta_t \epsilon_t)H_{t-1}$ into the expectation:

$$
\begin{aligned}
\mathbb{E}\left[H_t H_t^\top\right] &= \mathbb{E}\left[(I + \beta_t B^{-1}A + \beta_t \epsilon_t)H_{t-1}H_{t-1}^\top(I + \beta_t B^{-1}A + \beta_t \epsilon_t)^\top\right] \\
&= (I + \beta_t B^{-1}A)\mathbb{E}\left[H_{t-1}H_{t-1}^\top\right](I + \beta_t B^{-1}A)^\top + \beta_t \mathbb{E}\left[\epsilon_t H_{t-1}H_{t-1}^\top + H_{t-1}H_{t-1}^\top \epsilon_t^\top\right] \\
&\quad + \beta_t^2 \mathbb{E}\left[\epsilon_t H_{t-1}H_{t-1}^\top \epsilon_t^\top\right]
\end{aligned}
$$

If we assume to have a series of upper bounds $\theta_1 \leq \cdots \leq \theta_{t-1}$ such that:

$$\mathbb{E}\left[B_u B_u^\top\right] \preceq \theta_u I \tag{35}$$

The following inequality holds:

$$(I + \beta_t B^{-1}A)\mathbb{E}\left[H_{t-1}H_{t-1}^\top\right](I + \beta_t B^{-1}A)^\top \preceq \theta_{t-1}(I + \beta_t B^{-1}A)(I + \beta_t B^{-1}A)^\top \tag{36}$$

Furthermore, we show how that $(I + \beta_t B^{-1}A)(I + \beta_t B^{-1}A)^\top \preceq (1 + \beta_t \lambda_1)^2 I$:

Indeed, let $v$ be an eigenvector of $B^{-\frac{1}{2}}AB^{-\frac{1}{2}}$ with eigenvalue $\lambda$ and denote $\tilde{v} = B^{\frac{1}{2}}v$. We show that $\tilde{v}$ is an eigenvector of $(I + \beta_t B^{-1}A)(I + \beta_t B^{-1}A)^\top$ with eigenvalue $(1 + \beta_t \lambda)^2$:

$$
\begin{aligned}
\tilde{v}^\top(I + \beta_t B^{-1}A)(I + \beta_t B^{-1}A)^\top \tilde{v} &= v^\top(B^{\frac{1}{2}} + \beta_t B^{-\frac{1}{2}}A)(B^{\frac{1}{2}} + \beta_t B^{-\frac{1}{2}}A)^\top v \\
&= v^\top(B^{\frac{1}{2}} + \beta_t B^{-\frac{1}{2}}A)B^{-\frac{1}{2}}B^{\frac{1}{2}}B^{\frac{1}{2}}B^{-\frac{1}{2}}(B^{\frac{1}{2}} + \beta_t B^{-\frac{1}{2}}A)^\top v \\
&= v^\top(I + \beta_t B^{-\frac{1}{2}}AB^{-\frac{1}{2}})B(I + \beta_t B^{-\frac{1}{2}}AB^{-\frac{1}{2}})^\top v \\
&= (1 + \beta_t \lambda)^2 v^\top B v \\
&= (1 + \beta_t \lambda)^2 \tilde{v}^\top \tilde{v}
\end{aligned}
$$

As a consequence, we conclude the set of eigenvalues of $(I + \beta_t B^{-1}A)(I + \beta_t B^{-1}A)^\top$ equals $\{(1 + \beta_t \lambda_i)^2\}_{i=1}^d$, since the set of eigenvalues of $B^{-\frac{1}{2}}AB^{-\frac{1}{2}}$ equals $\{\lambda_i\}_{i=1}^d$, the set of eigenvalues of $B^{-1}A$. Therefore we conclude that

$$(I + \beta_t B^{-1}A)(I + \beta_t B^{-1}A)^\top \preceq (1 + \beta_t \lambda_1)^2 I \tag{37}$$

We proceed to bound the remaining terms.

$$
\begin{aligned}
\mathbb{E}\left[\epsilon_t H_{t-1}H_{t-1}^\top \epsilon_t^\top\right] &\leq \mathbb{E}\left[\|\epsilon_t\|\|H_{t-1}H_{t-1}^\top\|\|\epsilon_t^\top\|\right] \\
&\leq B_{\epsilon_t}^2 \mathbb{E}\left[\|H_{t-1}H_{t-1}^\top\|\right] \\
&\leq B_{\epsilon_t}^2 \mathbb{E}\left[\text{Tr}(H_{t-1}H_{t-1}^\top)\right] \\
&\leq dB_{\epsilon_t}^2 \|\mathbb{E}\left[H_{t-1}H_{t-1}^\top\right]\| \\
&\leq dB_{\epsilon_t}^2 \theta_{t-1}
\end{aligned} \tag{38}
$$

The first step is a consequence of Cauchy Schwartz, the second step because of the uniform boundedness of $\epsilon_t$ and the last step is true because $H_{t-1}H_{t-1}^\top$ is a positive semidefinite matrix.

**Terms with a single** $\epsilon_t$: Let $H_{t-1} = \prod_{j=t-r_t+1}^{t-1}(I + \beta_j G_j)H_{t-r_t}$.

Define $H_{t-1}^{t-r_t+1} := \prod_{j=t-r_t+1}^{t-1}(I + \beta_j G_j)$ and $L_{t-1}^{t-r_t+1} := H_{t-1}^{t-r_t+1} - I$.

In order to control this term we start by bounding $\|L_{t-1}^{t-r_t+1}\|$. For this we use a crude bound.

$$L_{t-1}^{t-r_t-1} = \sum_{k=1}^{r_t}\left(\sum_{i_1>\cdots>i_k\in[t-r_t-1,\cdots,t-1]}\left[\prod_{j=1}^{k}\beta_{i_j}G_{i_j}\right]\right) \tag{39}$$

For any $k \in [1,\cdots,r_t]$:

$$\left\|\sum_{i_1>\cdots>i_k\in[t-r_t-1,\cdots,t-1]}\left[\prod_{j=1}^{k}\beta_{i_j}G_{i_j}\right]\right\| \leq \sum_{i_1>\cdots>i_k\in[t-r_t-1,\cdots,t-1]}\left[\prod_{j=1}^{k}\|\beta_{i_j}G_{i_j}\|\right]$$
$$\leq \sum_{i_1>\cdots>i_k\in[t-r_t-1,\cdots,t-1]}\mathcal{G}_t^k\beta_{t-r_t}^k$$
$$\leq [r_t\mathcal{G}_t\beta_{t-r_t}]^k$$

The first follows from the triangle inequality, the second because of the uniform boundedness assumptions at the beginning of the section and the third because $\binom{r_t}{k} \leq r_t^k$.

For all $t \geq 0$, since the step size condition holds:

$$[r_t\mathcal{G}_t\beta_{t-r_t}]^k \leq [2r_t\mathcal{G}_t\beta_t]^k \leq 2r_t\mathcal{G}_t\beta_t \leq \frac{1}{2}$$

Putting these rough bounds together we conclude that:

$$\|L_{t-1}^{t-r_t-1}\| \leq \sum_{k=1}^{r_t}[2r_t\mathcal{G}_t\beta_t]^k = [2r_t\mathcal{G}_t\beta_t]\frac{1-[2r_t\mathcal{G}_t\beta_t]^k}{1-[2r_t\mathcal{G}_t\beta_t]} \leq 2[2r_t\mathcal{G}_t\beta_t] = 4r_t\mathcal{G}_t\beta_t, \tag{40}$$

where we have used that $1/(1-x) \leq 2x$ for $x \in [0,1/2]$. We can write $H_t = (I+L_{t-1}^{t-r_t+1})H_{t-r_t} = H_{t-r_t} + L_{t-1}^{t-r_t+1}H_{t-r_t}$. Substituting this equation into $\mathbb{E}\left[\epsilon_t H_{t-1}H_{t-1}^\top + H_{t-1}H_{t-1}^\top\epsilon_t^\top\right]$ gives us:

$$\mathbb{E}\left[\epsilon_t H_{t-1}H_{t-1}^\top + H_{t-1}H_{t-1}^\top\epsilon_t^\top\right] = \mathbb{E}\left[\epsilon_t(H_{t-r_t} + L_{t-1}^{t-r_t+1}H_{t-r_t})(H_{t-r_t} + L_{t-1}^{t-r_t+1}H_{t-r_t})^\top\right]$$
$$+ \mathbb{E}\left[(H_{t-r_t} + L_{t-1}^{t-r_t+1}H_{t-r_t})(H_{t-r_t} + L_{t-1}^{t-r_t+1}H_{t-r_t})^\top\epsilon_t^\top\right]$$
$$= \mathbb{E}\left[\epsilon_t H_{t-r_t}H_{t-r_t}^\top\right] + \mathbb{E}\left[\epsilon_t H_{t-r_t}H_{t-r_t}^\top(L_{t-1}^{t-r_t+1})^\top\right]$$
$$+ \mathbb{E}\left[\epsilon_t L_{t-1}^{t-r_t+1}H_{t-r_t}H_{t-r_t}^\top\right] + \mathbb{E}\left[\epsilon_t L_{t-1}^{t-r_t+1}H_{t-r_t}H_{t-r_t}^\top(L_{t-1}^{t-r_t+1})^\top\right]$$
$$+ \mathbb{E}\left[H_{t-r_t}H_{t-r_t}^\top\epsilon_t^\top\right] + \mathbb{E}\left[H_{t-r_t}H_{t-r_t}^\top(L_{t-1}^{t-r_t+1})^\top\epsilon_t^\top\right]$$
$$+ \mathbb{E}\left[L_{t-1}^{t-r_t+1}H_{t-r_t}H_{t-r_t}^\top\epsilon_t^\top\right] + \mathbb{E}\left[L_{t-1}^{t-r_t+1}H_{t-r_t}H_{t-r_t}^\top(L_{t-1}^{t-r_t+1})^\top\epsilon_t^\top\right]$$

We focus first on bounding the terms of this expansion containing $L_{t-1}^{t-r_t+1}$. We analyze the term $\mathbb{E}\left[\epsilon_t L_{t-1}^{t-r_t+1}H_{t-r_t}H_{t-r_t}^\top\right]$.

$$\|\mathbb{E}\left[\epsilon_t L_{t-1}^{t-r_t+1}H_{t-r_t}H_{t-r_t}^\top\right]\| \leq \mathbb{E}\left[\|\epsilon_t\|\|L_{t-1}^{t-r_t+1}\|\|H_{t-r_t}H_{t-r_t}^\top\|\right]$$
$$\leq B_{\epsilon_t}4r_t\mathcal{G}_t\beta_t\mathbb{E}\left[\|H_{t-r_t}H_{t-r_t}^\top\|\right]$$
$$\leq B_{\epsilon_t}4r_t\mathcal{G}_t\beta_t\mathbb{E}\left[\text{Tr}(H_{t-r_t}H_{t-r_t})\right]$$
$$\leq B_{\epsilon_t}4r_t\mathcal{G}_t\beta_t d\|\mathbb{E}\left[H_{t-r_t}H_{t-r_t}\right]\|$$

All other terms containing $L_{t-1}^{t-r_t+1}$ can be bounded in the same way. Combining these terms, we obtain the following bound for the sum of all these terms:

$$\|\mathbb{E}\left[\epsilon_t H_{t-1}H_{t-1}^\top + H_{t-1}H_{t-1}^\top\epsilon_t^\top\right]\| \leq B_{\epsilon_t}4\mathcal{G}_t dr_t\beta_t(4 + 8\mathcal{G}_t r_t\beta_t)$$

$$= 16B_{\epsilon_t}\mathcal{G}_t dr_t\beta_t + 32dB_{\epsilon_t}\mathcal{G}_t^2 r_t\beta_t^2$$
$$\leq 8dB_{\epsilon_t}\mathcal{G}_t\beta_t + 16B_{\epsilon_t}\mathcal{G}_t d\beta_t r_t$$
$$= 8dB_{\epsilon_t}\mathcal{G}_t\beta_t(2r_t + 1)$$

The last inequality holds because of the step size condition. It remains to bound the terms $\mathcal{E}\left[\epsilon_t H_{t-r_t}H_{t-r_t}^\top\right]$ and $\mathcal{E}\left[H_{t-r_t}H_{t-r_t}^\top\epsilon_t^\top\right]$.

By assumption, we know $\|\mathbb{E}\left[\epsilon_t|\mathcal{F}_{t-r_t}\right]\| \leq \mathcal{A}_t\beta_t r_t$ and therefore:

$$\|\mathbb{E}\left[\epsilon_t H_{t-r_t}H_{t-r_t}^\top\right]\| \leq \mathbb{E}\left[\|\mathbb{E}\left[\epsilon_t|\mathcal{F}_{t-r_t}\right]H_{t-r_t}H_{t-r_t}^\top\|\right]$$
$$\leq \mathbb{E}\left[\|\mathbb{E}\left[\epsilon_t|\mathcal{F}_{t-r_t}\right]\|\|H_{t-r_t}H_{t-r_t}^\top\|\right]$$
$$\leq \mathcal{A}_t r_t\beta_t\mathbb{E}\left[\|H_{t-r_t}H_{t-r_t}^\top\|\right]$$
$$\leq \mathcal{A}_t r_t\beta_t\mathbb{E}\left[\text{Tr}(H_{t-r_t}H_{t-r_t}^\top)\right]$$
$$\leq d\cdot\mathcal{A}_t r_t\beta_t\mathbb{E}\left[\|H_{t-r_t}H_{t-r_t}^\top\|\right]$$
$$\leq d\cdot\mathcal{A}_t r_t\beta_t\theta_{t-r_t}$$
$$\leq d\cdot\mathcal{A}_t r_t\beta_t\theta_{t-1}$$

Combining the last bounds we get that whenever $t > t_0$:

$$\|\mathbb{E}\left[\epsilon_t H_{t-1}H_{t-1}^\top + H_{t-1}H_{t-1}^\top\epsilon_t^\top\right]\| \leq \left(8dB_{\epsilon_t}\mathcal{G}_t\beta_t(2r_t + 1) + d\mathcal{A}_t r_t\beta_t\right)\theta_{t-1} \qquad (41)$$

Also, whenever $t \leq t_0$, we have that,

$$\|\mathbb{E}\left[\epsilon_t H_{t-1}H_{t-1}^\top + H_{t-1}H_{t-1}^\top\epsilon_t^\top\right]\| \leq 2dB_{\epsilon_t}\theta_{t-1} \qquad (42)$$

Combining the bound of equation 41 with equations 36, 38, and 42 yields for $t > t_0$

$$\|\mathbb{E}\left[H_t H_t^\top\right]\| \leq \theta_{t-1}\|(I + \beta_t B^{-1}A)(I + \beta_t B^{-1}A)^\top\| + \theta_{t-1}\beta_t^2\left(8dB_{\epsilon_t}\mathcal{G}_t(2r_t + 1) + d\mathcal{A}_t r_t\right)$$
$$+ \theta_{t-1}\beta_t^2 dB_\epsilon^2$$
$$\leq \theta_{t-1}\left(1 + 2\beta_t\lambda_1 + \beta_t^2(\Lambda_1 + dB_\epsilon^2)\right) + \beta_t^2\left(8dB_{\epsilon_t}\mathcal{G}_t(2r_t + 1) + d\mathcal{A}_t r_t\right))$$

where $\Lambda_1 = \lambda_1^2$. This gives us a recursion of the form:

$$\theta_t = \theta_{t-1}\left(1 + 2\beta_t\lambda_1 + \beta_t^2 dr_t(\mathcal{A}_t + B_{\epsilon_t}\mathcal{G}_t)\mathcal{C}^{(1)}\right) \qquad (43)$$

where $\mathcal{C}^{(1)}$ is the smallest constant depending on $\Lambda_1$ such that:

$$dr_t(\mathcal{A}_t + B_{\epsilon_t}\mathcal{G}_t + B_{\epsilon_t}^2)\mathcal{C}^{(1)} \geq \Lambda_1 + dB_\epsilon^2 + 8dB_{\epsilon_t}\mathcal{G}_t(2r_t + 1) + d\mathcal{A}_t r_t \qquad (44)$$

Similarly, whenever $t \leq t_0$, we have that

$$\|\mathbb{E}\left[H_t H_t^\top\right]\| \leq \theta_{t-1}\|(I + \beta_t B^{-1}A)(I + \beta_t B^{-1}A)^\top\| + \theta_{t-1}\beta_t * 2dB_{\epsilon_t}$$
$$+ \theta_{t-1}\beta_t^2 dB_\epsilon^2$$
$$\leq \theta_{t-1}\left(1 + 2\beta_t\lambda_1 + \beta_t^2(\Lambda_1 + dB_\epsilon^2) + \beta_t * 2dB_{\epsilon_t}\right)$$
$$\leq \theta_{t-1}\left(1 + 2\beta_t\lambda_1 + \beta_i^2 dr_i(\mathcal{A}_i + B_{\epsilon_i}\mathcal{G}_i + B_{\epsilon_i}^2)\mathcal{C}^{(1)} + \beta_t * 2dB_{\epsilon_t}\right)$$

Using the inequality $(1+x) \leq \exp(x)$ for $x \geq 0$, and noting that $\theta_0 = 1$ we obtain the desired result:

$$\theta_t \leq \exp(\sum_{i=1}^{t} 2\beta_i\lambda_1 + \beta_i^2 dr_i(\mathcal{A}_i + B_{\epsilon_i}\mathcal{G}_i + B_{\epsilon_i}^2)\mathcal{C}^{(1)} + \sum_{j=1}^{t_0} \beta_j 2dB_{\epsilon_j})$$

$\square$

### E.2 Orthogonal Subspace: Upper Bound on Expectation of $\text{Tr}(V_\perp^\top H_t H_t^\top V_\perp)$

In this section, we provide a bound on $\mathbb{E}\left[\text{Tr}(V_\perp^\top H_t H_t^\top V_\perp)\right]$.

**Lemma E.2.** *For all $t > 0$ and $\beta_t$ is such that $\beta_t \mathcal{G}_t r_t < \frac{1}{4}$ (which can be obtained by appropriately controlling the constant $\beta$ in the step size).,*

$$\mathbb{E}\left[\mathrm{Tr}(V_\perp^\top H_t H_t^\top V_\perp)\right]$$

$$\leq \exp\left(\sum_{j=1}^t 2\beta_j \lambda_2 + \beta_j^2 \lambda_2^2\right)\left(\mathrm{Tr}(V_\perp V_\perp^\top) + d\|V_\perp V_\perp^\top\|_2 \sum_{i=1}^t \left(r_i \beta_i^2 \mathcal{S}_i \mathcal{C}^{(2)} + \mathbb{1}(i \leq t_0)\beta_i * 2dB_{\epsilon_i}\right)\right)\cdot$$

$$\exp\left(\sum_{j=1}^i 2\beta_j(\lambda_1 - \lambda_2) + \beta_j^2(\mathcal{S}_j dr_j \mathcal{C}^{(1)} - \lambda_2^2) + \sum_{j=1}^{\min(i,t_0)} \beta_j 2 * dB_{\epsilon_j}\right)\right)$$

*where the $V_\perp$ matrix contains in its columns $\tilde{u}_2, \ldots, \tilde{u}_d$, where each $\tilde{u}_1 = Bu_i$ is the unnormalized left eigenvector of the matrix $B^{-1}A$ and $\mathcal{S}_i = (\mathcal{A}_i + B_{\epsilon_i}\mathcal{G}_i + B_{\epsilon_i}^2)$ for all $i$.*

*Proof.* Let $\gamma_t = \mathbb{E}\left[\mathrm{Tr}(V_\perp^\top H_t H_t^\top V_\perp)\right]$. By definition:

$$\gamma_t = \mathrm{Tr}(\mathbb{E}\left[H_t H_t^\top\right]V_\perp V_\perp^\top)$$

$$= \underbrace{\mathrm{Tr}(\mathbb{E}\left[H_{t-1}H_{t-1}^\top\right](I + \beta_t B^{-1}A)^\top V_\perp V_\perp^\top(I + \beta_t B^{-1}A))}_{\spadesuit}$$

$$+ \underbrace{\mathrm{Tr}(\beta_t \mathbb{E}\left[\epsilon_t H_{t-1}H_{t-1}^\top + H_{t-1}H_{t-1}^\top \epsilon_t^\top\right]V_\perp V_\perp^\top + \beta_t^2 \mathbb{E}\left[\epsilon_t H_{t-1}H_{t-1}^\top \epsilon_t^\top)\right]V_\perp V_\perp^\top)}_{\square}$$

We focus on term $\spadesuit$:

$$\underbrace{(I + \beta_t B^{-1}A)^\top V_\perp V_\perp^\top(I + \beta_t B^{-1}A)}_{\spadesuit_0} = \underbrace{V_\perp V_\perp^\top + \beta_t(B^{-1}A)^\top V_\perp V_\perp^\top + \beta_t V_\perp V_\perp^\top(B^{-1}A)}_{\spadesuit_1}$$

$$+ \beta_t^2(B^{-1}A)^\top V_\perp V_\perp^\top B^{-1}A$$

**Analysis of $\spadesuit_1$:** We begin by noting that the columns of $V_\perp$ contain the vectors $\tilde{u}_i$ which are the unnormalized left eigenvectors of $B^{-1}A$ and therefore,

$$V_\perp^\top(B^{-1}A) = V_\perp^\top \Lambda,$$

where $\Lambda$ is a diagonal matrix with $\Lambda_{i,i} = \lambda_{i+1} \ \forall i = 2 \ldots d$. Noting that $V_\perp V_\perp^\top \Lambda \preceq \lambda_2 V_\perp V_\perp^\top$, we obtain,

$$\spadesuit_1 \preceq V_\perp V_\perp^\top(1 + 2\beta_t \lambda_2). \tag{45}$$

Following a similar argument, we obtain that,

$$(B^{-1}A)^\top V_\perp V_\perp^\top B^{-1}A \preceq \lambda_2^2 V_\perp V_\perp^\top. \tag{46}$$

Combining Eqs (45) and (46), we obtain,

$$\spadesuit \leq \mathrm{Tr}\left(\mathbb{E}[H_{t-1}H_{t-1}^T]V_\perp V_\perp^\top(1 + 2\beta_t \lambda_2 + \beta_t^2 \lambda_2^2)\right)$$

The terms corresponding to $\square$ can also be bounded by bonding the operator norms of its two constituent expectations. In the same way as in Lemma E.1, let $H_{t-1} = (I + L_{t-1}^{t-r_t+1})H_{t-r_t}$. Note that $V_\perp V_\perp^\top \preceq \|V_\perp V_\perp^\top\|_2 I$ and we bound the normalized term $\square/\|V_\perp V_\perp^\top\|_2$.

$$\frac{\mathrm{Tr}(\mathbb{E}\left[\epsilon_t H_{t-1}H_{t-1}^\top + H_{t-1}H_{t-1}^\top \epsilon_t^\top\right]V_\perp V_\perp^\top)}{\|V_\perp V_\perp^\top\|_2} \leq \mathrm{Tr}\left(\mathbb{E}\left[\epsilon_t H_{t-1}H_{t-1}^\top + H_{t-1}H_{t-1}^\top \epsilon_t^\top\right]\right)$$

$$= \underbrace{\mathrm{Tr}(\mathbb{E}\left[H_{t-r_t}H_{t-r_t}^\top(\epsilon_t + \epsilon_t^\top)\right])}_{\Gamma_1} + \underbrace{\mathrm{Tr}(\mathbb{E}\left[H_{t-r_t}H_{t-r_t}^\top((L_{t-1}^{t-r_t+1})^\top \epsilon_t^\top + \epsilon_t L_{t-1}^{t-r_t+1})\right])}_{\Gamma_2}$$

$$+ \underbrace{\mathrm{Tr}(\mathbb{E}\left[H_{t-r_t}H_{t-r_t}^\top((L_{t-1}^{t-r_t+1})^\top \epsilon_t + \epsilon_t^\top L_{t-1}^{t-r_t+1})\right])}_{\Gamma_3}$$

$$+ \underbrace{\mathrm{Tr}(\mathbb{E}\left[H_{t-r_t}H_{t-r_t}^\top((L_{t-1}^{t-r_t+1})^\top \epsilon_t L_{t-1}^{t-r_t+1} + (L_{t-1}^{t-r_t+1})^\top \epsilon_t^\top L_{t-1}^{t-r_t+1})\right])}_{\Gamma_4}$$

Recall that $\|L_{t-1}^{t-r_t+1}\| \le 4r_t\mathcal{G}_t\beta_t$. As a consequence:

$$(L_{t-1}^{t-r_t+1})^\top \epsilon_t^\top + \epsilon_t L_{t-1}^{t-r_t+1} \preceq 2B_{\epsilon_t} * 4r_t\mathcal{G}_t\beta_t I = 8B_{\epsilon_t}r_t\mathcal{G}_t\beta_t I$$

$$(L_{t-1}^{t-r_t+1})^\top \epsilon_t + \epsilon_t^\top L_{t-1}^{t-r_t+1} \preceq 2B_{\epsilon_t} * 4r_t\mathcal{G}_t\beta_t I = 8B_{\epsilon_t}r_t\mathcal{G}_t\beta_t I$$

$$(L_{t-1}^{t-r_t+1})^\top \epsilon_t L_{t-1}^{t-r_t+1} + (L_{t-1}^{t-r_t+1})^\top \epsilon_t^\top L_{t-1}^{t-r_t+1} \preceq 2(4r_t\mathcal{G}_t\beta_t)^2 B_{\epsilon_t} I \preceq 8B_{\epsilon_t}r_t\mathcal{G}_t\beta_t I$$

The second inequality in the last line follows from the step size condition. Therefore:

$$
\begin{aligned}
\Gamma_2 + \Gamma_3 + \Gamma_4 &\le 32B_{\epsilon_t}r_t\mathcal{G}_t\beta_t \operatorname{Tr}(\mathbb{E}\left[H_{t-r_t}H_{t-r_t}^\top\right]) \\
&\le 32B_{\epsilon_t}r_t\mathcal{G}_t\beta_t d\|\mathbb{E}\left[H_{t-r_t}H_{t-r_t}^\top\right]\| \\
&\le 32B_{\epsilon_t}r_t\mathcal{G}_t\beta_t d\theta_{t-r_t} \\
&\le 32B_{\epsilon_t}r_t\mathcal{G}_t\beta_t d\theta_{t-1}
\end{aligned}
$$

We proceed to bound $\Gamma_1$. We know that $\|\mathbb{E}\left[\epsilon_t|\mathcal{F}_{t-r_t}\right]\| = \mathcal{A}_t\beta_t r_t$ and therefore $\mathbb{E}\left[\epsilon_t + \epsilon_t^\top|\mathcal{F}_{t-r_t}\right] \preceq 2\mathcal{A}_t\beta_t r_t I$

$$
\begin{aligned}
\Gamma_1 = \operatorname{Tr}(\mathbb{E}\left[H_{t-r_t}H_{t-r_t}^\top(\epsilon_t + \epsilon_t^\top)\right]) &= \mathbb{E}\left[\operatorname{Tr}(H_{t-r_t}H_{t-r_t}^\top\mathbb{E}\left[\epsilon_t + \epsilon_t^\top|\mathcal{F}_{t-r_t}\right])\right] \\
&\le 2\mathcal{A}_t\beta_t r_t\mathbb{E}\left[H_{t-r_t}H_{t-r_t}^\top\right] \\
&\le 2\mathcal{A}_t\beta_t r_t d\|\mathbb{E}\left[H_{t-r_t}H_{t-r_t}^\top\right]\| \\
&\le 2\mathcal{A}_t\beta_t r_t d\theta_{t-r_t} \\
&\le 2\mathcal{A}_t\beta_t r_t d\theta_{t-1}
\end{aligned}
$$

The last inequalities follow from the same argument as in equation 41, where $\theta_{t-1}$ is the upper bound obtained in the previous lemma for $\|\mathbb{E}\left[H_{t-1}H_{t-1}^\top\right]\|$.

As a consequence, whenever $t \ge t_0$ the first term in $\square/\|V_\perp V_\perp^\top\|_2$ can be bounded by:

$$\frac{\operatorname{Tr}(\mathbb{E}\left[\epsilon_t H_{t-1}H_{t-1}^\top + H_{t-1}H_{t-1}^\top\epsilon_t^\top\right] V_\perp V_\perp^\top)}{\|V_\perp V_\perp^\top\|_2} \le 32B_{\epsilon_t}r_t\mathcal{G}_t\beta_t d\theta_{t-1} + 2\mathcal{A}_t\beta_t r_t d\theta_{t-1}$$

For the case when $t < t_0$:

$$
\begin{aligned}
\frac{\operatorname{Tr}\left(\mathbb{E}\left[\epsilon_t H_{t-1}H_{t-1}^\top + H_{t-1}H_{t-1}^\top\epsilon_t^\top\right] V_\perp V_\perp^\top\right)}{\|V_\perp V_\perp^\top\|_2} &= \frac{\mathbb{E}\left[\operatorname{Tr}\left((H_{t-1}H_{t-1}^\top)(V_\perp V_\perp^\top\epsilon_t + \epsilon_t^\top V_\perp V_\perp^\top)\right)\right]}{\|V_\perp V_\perp^\top\|_2} \\
&\le 2B_{\epsilon_t}\operatorname{Tr}(\mathbb{E}\left[H_{t-1}H_{t-1}^\top\right]) \\
&\le 2dB_{\epsilon_t}\|\mathbb{E}\left[H_{t-1}H_{t-1}^\top\right]\| \\
&\le 2dB_{\epsilon_t}\theta_{t-1}
\end{aligned}
$$

where the first inequality follows because $\|V_\perp V_\perp^\top\epsilon_t + \epsilon_t^\top V_\perp V_\perp^\top\| \le 2B_{\epsilon_t}\|V_\perp V_\perp^\top\|_2$.

And the second term in $\square/\|V_\perp V_\perp^\top\|_2$ can be bounded for all $t$:

$$
\begin{aligned}
\frac{\operatorname{Tr}(\mathbb{E}\left[\epsilon_t H_{t-1}H_{t-1}^\top\epsilon_t^\top\right] V_\perp V_\perp^\top)}{\|V_\perp V_\perp^\top\|_2} &\le \operatorname{Tr}(\mathbb{E}\left[\epsilon_t H_{t-1}H_{t-1}^\top\epsilon_t^\top\right]) \\
&= \operatorname{Tr}(\mathbb{E}\left[H_{t-1}H_{t-1}^\top\epsilon_t^\top\epsilon_t\right]) \\
&\le B_{\epsilon_t}^2\operatorname{Tr}(\mathbb{E}\left[H_{t-1}H_{t-1}^\top\right]) \\
&\le dB_{\epsilon_t}^2\|\mathbb{E}\left[H_{t-1}H_{t-1}^\top\right]\| \\
&\le dB_{\epsilon_t}^2\theta_{t-1}
\end{aligned}
$$

Let $C^{(2)}$ be a constant such that $dr_t(\mathcal{A}_t + B_{\epsilon_t}\mathcal{G}_t + B_{\epsilon_t}^2)C^{(2)} \ge 32dB_{\epsilon_t}r_t\mathcal{G}_t + 2d\mathcal{A}_t r_t + dB_\epsilon^2$.

The last inequalities follow from the same argument as in equation 38. We conclude that whenever $t > t_0$:

$$
\begin{aligned}
\square &= \beta_t\operatorname{Tr}(\mathbb{E}\left[\epsilon_t H_{t-1}H_{t-1}^\top + H_{t-1}H_{t-1}^\top\epsilon_t^\top\right] V_\perp V_\perp^\top) + \beta_t^2\operatorname{Tr}(\mathbb{E}\left[\epsilon_t H_{t-1}H_{t-1}^\top\epsilon_t^\top\right] V_\perp V_\perp^\top) \\
&\le dr_t\beta_t^2(\mathcal{A}_t + B_{\epsilon_t}\mathcal{G}_t + B_{\epsilon_t}^2)C^{(2)}\theta_{t-1}\|V_\perp V_\perp^\top\|_2
\end{aligned}
$$

Combining ♠ with □, whenever $t > t_0$:

$$\gamma_t = ♠ + □ \leq \gamma_{t-1}(1 + 2\beta_t\lambda_2 + \beta_t\lambda_2^2) + dr_t\beta_t^2(\mathcal{A}_t + B_{\epsilon_t}\mathcal{G}_t + B_{\epsilon_t}^2)\mathcal{C}^{(2)}\theta_{t-1}\|V_\perp V_\perp^\top\|_2$$

On the other hand, for $t \leq t_0$:

$$□ = \beta_t \operatorname{Tr}(\mathbb{E}\left[\epsilon_t H_{t-1}H_{t-1}^\top + H_{t-1}H_{t-1}^\top \epsilon_t^\top\right]V_\perp V_\perp^\top) + \beta_t^2 \operatorname{Tr}(\mathbb{E}\left[\epsilon_t H_{t-1}H_{t-1}^\top \epsilon_t^\top\right]V_\perp V_\perp^\top)$$

$$\leq \left(dr_t\beta_t^2(\mathcal{A}_t + B_{\epsilon_t}\mathcal{G}_t + B_{\epsilon_t}^2)\mathcal{C}^{(2)} + \beta_t * dB_{\epsilon_t}\right)\theta_{t-1}\|V_\perp V_\perp^\top\|_2$$

And consequently:

$$\gamma_t = ♠ + □ \leq \gamma_{t-1}(1 + 2\beta_t\lambda_2 + \beta_t^2\lambda_2^2) + \left(dr_t\beta_t^2(\mathcal{A}_t + B_{\epsilon_t}\mathcal{G}_t + B_{\epsilon_t}^2)\mathcal{C}^{(2)} + \beta_t 2 * dB_{\epsilon_t}\right)\theta_{t-1}\|V_\perp V_\perp^\top\|_2$$

Using the bound for $\theta_{t-1}$ in Lemma E.1 and the inequality $1 + x \leq e^x$:

$$\gamma_t \leq \exp(2\beta_t\lambda_2 + \beta_t^2\lambda_2^2)\gamma_{t-1} +$$

$$\|V_\perp V_\perp^\top\|_2 \left(dr_t\beta_t^2(\mathcal{A}_t + B_{\epsilon_t}\mathcal{G}_t + B_{\epsilon_t}^2)C^{(2)} + \mathbb{1}(t \leq t_0)\beta_t * 2dB_{\epsilon_t}\right) \cdot$$

$$\exp\left(\sum_{i=1}^{t-1} 2\beta_i\lambda_1 + dr_i\beta_i^2(\mathcal{A}_i + B_{\epsilon_i}\mathcal{G}_i + B_{\epsilon_i}^2)\mathcal{C}^{(1)} + \sum_{j=1}^{\min(t,t_0)} \beta_j 2 * dB_{\epsilon_j}\right)$$

After doing recursion we obtain the upper bound,

$$\gamma_t \leq \sum_{i=1}^{t}\left[\|V_\perp V_\perp^\top\|_2\left(dr_i\beta_i^2(\mathcal{A}_i + B_{\epsilon_i}\mathcal{G}_i + B_{\epsilon_i}^2)C^{(2)} + \mathbb{1}(i \leq t_0)\beta_i 2dB_{\epsilon_i}\right)\exp\left(\sum_{j=i+1}^{t} 2\beta_j\lambda_2 + \beta_j^2\lambda_2^2\right)\cdot\right.$$

$$\left.\exp\left(\sum_{j=1}^{i} 2\beta_j\lambda_1 + dr_j\beta_j^2(\mathcal{A}_j + B_{\epsilon_j}\mathcal{G}_j + B_{\epsilon_j}^2)C^{(1)} + \sum_{j=1}^{\min(i,t_0)} \beta_j 2 * dB_{\epsilon_j}\right)\right]$$

$$+ \exp(\sum_{j=1}^{t} 2\beta_j\lambda_2 + \beta_j^2\lambda_2^2)\gamma_0$$

Where $\gamma_0 = \operatorname{Tr}(V_\perp V_\perp^\top)$. Let $\mathcal{S}_i = (\mathcal{A}_i + B_{\epsilon_i}\mathcal{G}_i + B_{\epsilon_i}^2)$

$$\gamma_t \leq \exp(\sum_{j=1}^{t} 2\beta_j\lambda_2 + \beta_j^2\lambda_2^2)\left(\operatorname{Tr}(V_\perp V_\perp^\top) + d\|V_\perp V_\perp^\top\|_2 \sum_{i=1}^{t}\left(r_i\beta_i^2\mathcal{S}_i\mathcal{C}^{(2)} + \mathbb{1}(i \leq t_0)\beta_i * 2dB_{\epsilon_i}\right)\cdot\right.$$

$$\left.\exp\left(\sum_{j=1}^{i} 2\beta_j(\lambda_1 - \lambda_2) + \beta_j^2(\mathcal{S}_j dr_j\mathcal{C}^{(1)} - \lambda_2^2) + \sum_{j=1}^{\min(i,t_0)} \beta_j 2 * dB_{\epsilon_j}\right)\right)$$

$$□$$

## E.3  Lower Bound on Expectation of $\tilde{u}_1^\top H_t H_t^\top \tilde{u}_1$

**Lemma E.3.** *For all $t \geq 0$ and $\beta_t \geq 0$ we have,*

$$\mathbb{E}[\tilde{u}_1^\top H_t H_t^\top \tilde{u}_1] \geq \|\tilde{u}_1\|_2^2 \exp\left(\sum_{i=1}^{t} 2\beta_i\lambda_1 - 4\beta_i^2\lambda_1^2\right) - d\|\tilde{u}_1\|_2^2 \sum_{i=1}^{t}\left((\beta_i^2 r_t(\mathcal{A}_t + B_{\epsilon_t}\mathcal{G}_t + B_{\epsilon_t}^2)\mathcal{C}^{(2)}\right.$$

$$\left. + \beta_i\mathbb{I}(t \leq t_0)(B_{\epsilon_t}))\exp\left(\sum_{j=1}^{i-1} 2\beta_j\lambda_1 + \beta_j^2 dr_j(\mathcal{A}_j + B_{\epsilon_j}\mathcal{G}_j + B_{\epsilon_j}^2)\mathcal{C}^{(1)} + \sum_{j=1}^{\min(t-1,t_0)} \beta_j 2dB_{\epsilon_j}\right)\right),$$
(47)

*where $\tilde{u}_1$ is the unnormalized left eigenvector corresponding to the maximum eigenvalue $\lambda_1$ of $(B^{-1}A)^\top$.*

*Proof.* Let $\gamma_t \triangleq \mathbb{E}[v^\top H_t H_t^\top v]$ where $v = \tilde{u}_1/\|\tilde{u}_1\|_2$ be the normalized left eigenvector and $\Sigma = B^{-1}A$. Since $H_t = (I + \beta_t G_t)$, we can obtain a bound on $\gamma_t$ as,

$$
\begin{aligned}
\gamma_t &= \mathbb{E}[v^\top (I + \beta_t G_t) H_{t-1} H_{t-1}^\top (I + \beta_t G_t)^\top v] \\
&= \mathbb{E}[v^\top (I + \beta_t \Sigma) H_{t-1} H_{t-1}^\top (I + \beta_t \Sigma)^\top v] + \beta_t \mathbb{E}[v^\top (\epsilon_t H_{t-1} H_{t-1}^\top + H_{t-1} H_{t-1}^\top \epsilon_t^\top) v] \\
&\quad + \beta_t^2 \mathbb{E}[v^\top (\epsilon_t + \Sigma) H_{t-1} H_{t-1}^\top (\epsilon_t + \Sigma)^\top v] - \beta_t^2 \mathbb{E}[v^\top \Sigma H_{t-1} H_{t-1}^\top \Sigma^\top v] \\
&\overset{\zeta_1}{\geq} \mathbb{E}[v^\top H_{t-1} H_{t-1}^\top v] + \beta_t \mathbb{E}[v^\top \Sigma H_{t-1} H_{t-1}^\top v] + \beta_t \mathbb{E}[v^\top H_{t-1} H_{t-1}^\top \Sigma^\top v] \\
&\quad + \beta_t \mathbb{E}[v^\top (\epsilon_t H_{t-1} H_{t-1}^\top + H_{t-1} H_{t-1}^\top \epsilon_t^\top) v] \\
&\overset{\zeta_2}{=} (1 + 2\lambda_1 \beta_t)\gamma_{t-1} + \beta_t \underbrace{\mathbb{E}[v^\top (\epsilon_t H_{t-1} H_{t-1}^\top + H_{t-1} H_{t-1}^\top \epsilon_t^\top) v]}_{(I)},
\end{aligned}
\tag{48}
$$

where $\zeta_1$ follows since $(\epsilon_t + \Sigma) H_{t-1} H_{t-1}^\top (\epsilon_t + \Sigma)^\top$ is a positive semi-definite matrix and $\zeta_2$ follows since $v$ is the top left eigenvector of $\Sigma$. Now, in order to bound term (I), we note that

$$
\mathbb{E}[v^\top (\epsilon_t H_{t-1} H_{t-1}^\top + H_{t-1} H_{t-1}^\top \epsilon_t^\top) v] \geq -\|\mathbb{E}[\epsilon_t H_{t-1} H_{t-1}^\top + H_{t-1} H_{t-1}^\top \epsilon_t^\top]\|_2.
$$

Using the bound obtained in (41), we get that for $t > t_0$,

$$
\mathbb{E}[v^\top (\epsilon_t H_{t-1} H_{t-1}^\top + H_{t-1} H_{t-1}^\top \epsilon_t^\top) v] \geq -\beta_t (8dB_{\epsilon_t} \mathcal{G}_t (2r_t + 1) + d\mathcal{A}_t r_t)\theta_{t-1},
$$

and for $t \leq t_0$, we have from equation (42)

$$
\mathbb{E}[v^\top (\epsilon_t H_{t-1} H_{t-1}^\top + H_{t-1} H_{t-1}^\top \epsilon_t^\top) v] \geq -2dB_{\epsilon_t}\theta_{t-1},
$$

Where $\theta_{t-1}$ is defined as in E.1. We next use the bound from lemma E.1 to lower bound $-\theta_{t-1}$,

$$
\begin{aligned}
\mathbb{E}[v^\top (\epsilon_t H_{t-1} H_{t-1}^\top + H_{t-1} H_{t-1}^\top \epsilon_t^\top) v] \geq &-\beta_t (8dB_{\epsilon_t} \mathcal{G}_t (2r_t + 1) + d\mathcal{A}_t r_t + \mathbb{I}(t \leq t_0)(2dB_{\epsilon_t})) \cdot \\
&\exp \left( \sum_{i=1}^{t-1} 2\beta_i \lambda_1 + \beta_i^2 dr_i (\mathcal{A}_i + B_{\epsilon_i} \mathcal{G}_i + B_{\epsilon_i}^2)\mathcal{C}^{(1)} + \sum_{j=1}^{\min(t-1,t_0)} \beta_j 2 dB_{\epsilon_j} \right).
\end{aligned}
$$

Recall that in Lemma E.2 we defined $C^{(2)}$ as a constant such that: $dr_t (\mathcal{A}_t + B_{\epsilon_t} \mathcal{G}_t + B_{\epsilon_t}^2)\mathcal{C}^{(2)} \geq 32dB_{\epsilon_t} r_t \mathcal{G}_t + 2d\mathcal{A}_t r_t + dB_\epsilon^2$, therefore:

$$
\begin{aligned}
\mathbb{E}[v^\top (\epsilon_t H_{t-1} H_{t-1}^\top + H_{t-1} H_{t-1}^\top \epsilon_t^\top) v] \geq &-\left(\beta_t dr_t (\mathcal{A}_t + B_{\epsilon_t} \mathcal{G}_t + B_{\epsilon_t}^2)\mathcal{C}^{(2)} + \mathbb{I}(t \leq t_0)(2dB_{\epsilon_t})\right) \cdot \\
&\exp \left( \sum_{i=1}^{t-1} 2\beta_i \lambda_1 + \beta_i^2 dr_i (\mathcal{A}_i + B_{\epsilon_i} \mathcal{G}_i + B_{\epsilon_i}^2)\mathcal{C}^{(1)} + \sum_{j=1}^{\min(t-1,t_0)} \beta_j 2 dB_{\epsilon_j} \right).
\end{aligned}
$$

Substituting the above in equation (48), we obtain the following recursion,

$$
\begin{aligned}
\gamma_t \geq (1 + 2\lambda_1 \beta_t)\gamma_{t-1} - &\left(\beta_t dr_t (\mathcal{A}_t + B_{\epsilon_t} \mathcal{G}_t + B_{\epsilon_t}^2)\mathcal{C}^{(2)} + \mathbb{I}(t \leq t_0)(2dB_{\epsilon_t})\right) \cdot \\
&\exp \left( \sum_{i=1}^{t-1} 2\beta_i \lambda_1 + \beta_i^2 dr_i (\mathcal{A}_i + B_{\epsilon_i} \mathcal{G}_i + B_{\epsilon_i}^2)\mathcal{C}^{(1)} + \sum_{j=1}^{\min(t-1,t_0)} \beta_j 2 dB_{\epsilon_j} \right).
\end{aligned}
$$

Using the inequality $1 + x \geq \exp(x - x^2)$ for all $x \geq 0$, along with $\gamma_0 = 1$, we obtain,

$$
\begin{aligned}
\gamma_t \geq \exp \left( \sum_{i=1}^{t} 2\beta_i \lambda_1 - 4\beta_i^2 \lambda_1^2 \right) - d\sum_{i=1}^{t} \Big( (\beta_i^2 r_t (\mathcal{A}_t + B_{\epsilon_t} \mathcal{G}_t + B_{\epsilon_t}^2)\mathcal{C}^{(2)} + \beta_i \mathbb{I}(t \leq t_0)(B_{\epsilon_t})) \cdot \\
\exp \left( \sum_{j=1}^{i-1} 2\beta_j \lambda_1 + \beta_j^2 dr_j (\mathcal{A}_j + B_{\epsilon_j} \mathcal{G}_j + B_{\epsilon_j}^2)\mathcal{C}^{(1)} + \sum_{j=1}^{\min(t-1,t_0)} \beta_j 2 dB_{\epsilon_j} \right) \Big)
\end{aligned}
\tag{49}
$$

which concludes the proof of the lemma. $\qquad\square$

## E.4 Upper Bound on Variance of $\tilde{u}_1^\top H_t H_t^\top \tilde{u}_1$

In this section, we provide an upper bound on $\mathbb{E}\left[(v^\top H_t H_t^\top v)^2\right]$ which will be later used in order to lower bound the requisite term using the Chebychev Inequality. We first prove an upper bound on $\mathbb{E}\left[\text{Tr}(H_t H_t^\top H_t H_t^\top)\right]$ and use this in the next lemma to obtain the requisite bounds.

**Lemma E.4.** *For all $t \geq 0$:*

$$\mathbb{E}\left[\text{Tr}(H_t H_t^\top H_t H_t^\top)\right] \leq d \exp\left(\sum_{i=1}^{t} 4\lambda_1 \beta_i + d r_i (\mathcal{A}_i + B_{\epsilon_i}^2 + B_{\epsilon_i} \mathcal{G}_i) C^{(3)} \beta_i^2 + \sum_{j=1}^{\min(t,t_0)} \beta_j 2 * (\frac{101}{100})^3 B_{\epsilon_j}\right)$$

*As long as $\beta_t$ satisfies that for all $t$, $\|I + \beta_t B^{-1} A\| \leq \frac{101}{100}$, $\beta_t r_t \mathcal{G}_t < \frac{1}{4}$, and $\beta_t r_t \mathcal{B}_{\epsilon_t} < \frac{1}{4}$.*

*Proof.* We start by substituting the identity: $H_t = (I + \beta_t G_t) H_{t-1} = (I + \beta_t B^{-1} A + \beta_t \epsilon_t) H_{t-1}$.

Substituting this decomposition intro the trace we want to bound we obtain:

$$\mathbb{E}\left[\text{Tr}(H_t H_t^\top H_t H_t^\top)\right] = \text{Tr}\,\mathbb{E}\left[(I + \beta_t G_t) H_{t-1} H_{t-1}^\top (I + \beta_t G_t)^\top (I + \beta_t G_t) H_{t-1} H_{t-1}^\top (I + \beta_t G_t)^\top\right]$$

$$= \text{Tr}\,\mathbb{E}\left[\underbrace{H_{t-1} H_{t-1}^\top}_{\Gamma_1} \underbrace{(I + \beta_t G_t)^\top (I + \beta_t G_t)}_{\Gamma_2} \underbrace{H_{t-1} H_{t-1}^\top}_{\Gamma_1} \underbrace{(I + \beta_t G_t)^\top (I + \beta_t G_t)}_{\Gamma_2}\right]$$

$$\leq \text{Tr}\,\mathbb{E}\left[H_{t-1} H_{t-1}^\top H_{t-1} H_{t-1}^\top \underbrace{(I + \beta_t G_t)^\top (I + \beta_t G_t)(I + \beta_t G_t)^\top (I + \beta_t G_t)}_{\spadesuit}\right]$$

where last inequality follows from the trace inequality: $\text{Tr}\left(\Gamma_1 \Gamma_2 \Gamma_1 \Gamma_2\right) \leq \text{Tr}\left(\Gamma_1^2 \Gamma_2^2\right)$.

Expanding $\spadesuit$ yields:

$$\spadesuit = \underbrace{(I + \beta_t B^{-1} A)^\top (I + \beta_t B^{-1} A)(I + \beta_t B^{-1} A)^\top (I + \beta_t B^{-1} A)}_{\spadesuit_1}$$

$$+ \underbrace{\beta_t \left(\epsilon_t^\top (I + \beta_t B^{-1} A)(I + \beta_t B^{-1} A)^\top (I + \beta_t B^{-1} A) + (I + \beta_t B^{-1} A)^\top \epsilon_t (I + \beta_t B^{-1} A)^\top (I + \beta_t B^{-1} A)\right)}_{\spadesuit_2^{(1)}}$$

$$+ \underbrace{\beta_t \left((I + \beta_t B^{-1} A)^\top (I + \beta_t B^{-1} A)\epsilon_t^\top (I + \beta_t B^{-1} A) + (I + \beta_t B^{-1} A)^\top (I + \beta_t B^{-1} A)(I + \beta_t B^{-1} A)^\top \epsilon_t\right)}_{\spadesuit_2^{(2)}}$$

$$+ \spadesuit_3,$$

where $\spadesuit_3$ contains all terms with at least two $\epsilon_t$. Additionally, $\spadesuit_3$ is a symmetric matrix with norm satisfying:

$$\|\spadesuit_3\| \overset{\gamma_1}{\leq} \beta_t^2 \binom{4}{2} \|\epsilon_t\|^2 \|I + \beta_t B^{-1} A\|^2 + \beta_t^3 \binom{4}{3} \|\epsilon_t\|^3 \|I + \beta_t B^{-1} A\| + \beta_t^4 \binom{4}{4} \|\epsilon_t\|^4$$

$$\overset{\gamma_2}{\leq} \beta_t^2 * 6 B_{\epsilon_t}^2 (\frac{101}{100})^2 + \beta_t^3 * 4 * B_\epsilon^3 (\frac{101}{100}) + \beta_t^4 B_{\epsilon_t}^4$$

$$\overset{\gamma_3}{\leq} \beta_t^2 * 6 B_{\epsilon_t}^2 (\frac{101}{100})^2 + \beta_t^2 B_{\epsilon_t}^2 (\frac{101}{100}) + \beta_t^2 B_{\epsilon_t}^2 \frac{1}{16}$$

$$\leq 8 \beta_t^2 B_{\epsilon_t}^2$$

where the inequality $\gamma_1$ follows from triangle, and $\gamma_2, \gamma_3$ from the step size condition. Recall that:

$$\text{Tr}\,\mathbb{E}\left[H_{t-1} H_{t-1}^\top H_{t-1} H_{t-1}^\top \spadesuit\right] = \text{Tr}\,\mathbb{E}\left[H_{t-1} H_{t-1}^\top H_{t-1} H_{t-1}^\top \left(\spadesuit_1 + (\spadesuit_2^{(1)} + \spadesuit_2^{(2)}) + \spadesuit_3\right)\right]$$

Since, as shown in equation Equation 37 we have that $(I + \beta_t B^{-1} A)(I + \beta_t B^{-1} A)^\top \preceq (1 + \beta_t \lambda_1)^2 I$. then, $\spadesuit_1 \preceq (1 + \beta_t \lambda_1)^4 I$ (this is because $(I + \beta_t B^{-1} A)(I + \beta_t B^{-1} A)^\top$ and $(I + \beta_t B^{-1} A)^\top (I + \beta_t B^{-1} A)$ have the same eigenvalues. And therefore $\spadesuit_1 \preceq (1 + \beta_t \lambda_1)^4 I \preceq (1 + 4\beta_t \lambda_1 + 11\beta_t^2 \max(\lambda_1^4, 1)) I$ and $\spadesuit_3 \preceq 8\beta_t^2 B_{\epsilon_t}^2 I$, thus implying:

$$\text{Tr}\,\mathbb{E}\left[H_{t-1} H_{t-1}^\top H_{t-1} H_{t-1}^\top (\spadesuit_1 + \spadesuit_3)\right] \leq (1 + 4\beta_t \lambda_1 + 11\beta_t^2 (\lambda_1^4 \vee 1) + 8\beta_t^2 B_{\epsilon_t}^2) \text{Tr}\,\mathbb{E}\left[H_{t-1} H_{t-1}^\top H_{t-1} H_{t-1}^\top\right]$$

It only remains to bound the term $\mathrm{Tr}\,\mathbb{E}\Big[\underbrace{H_{t-1}H_{t-1}^\top H_{t-1}H_{t-1}^\top(\spadesuit_2^{(1)}+\spadesuit_2^{(2)})}_{\Gamma}\Big]$. Notice that $\spadesuit_2^{(1)}+$

$\spadesuit_2^{(2)}$ is a symmetric matrix. Therefore, whenever $t\le t_0$:

$$\|\spadesuit_2^{(1)}+\spadesuit_2^{(2)}\|\le 2\beta_t B_{\epsilon_t}\|I+\beta_t B^{-1}A\|^3\le 2\beta_t(\frac{101}{100})^3 B_{\epsilon_t}$$

And also whenever $t\le t_0$ :

$$\mathrm{Tr}\,\mathbb{E}\Big[H_{t-1}H_{t-1}^\top H_{t-1}H_{t-1}^\top(\spadesuit_2^{(1)}+\spadesuit_2^{(2)})\Big]\le 2\beta_t B_{\epsilon_t}\|I+\beta_t B^{-1}A\|^3$$
$$\le 2(1.01)^3\beta_t\,\mathrm{Tr}\,\mathbb{E}\Big[H_{t-1}H_{t-1}^\top H_{t-1}H_{t-1}^\top\Big]$$

We will use similar arguments to what we used in previous sections to bound these types of terms for the case when $t>t_0$:

Let $H_{t-1}=(I+L_{t-1}^{t-r_t+1})H_{t-r_t}$ as in Lemma E.1, therefore:

$$\mathrm{Tr}\,\mathbb{E}\Big[H_{t-1}H_{t-1}^\top H_{t-1}H_{t-1}^\top(\spadesuit_2^{(1)}+\spadesuit_2^{(2)})\Big]=\mathrm{Tr}\,\mathbb{E}\big[(I+L_{t-1}^{t-r_t+1})H_{t-r_t}H_{t-r_t}^\top(I+L_{t-1}^{t-r_t+1})^\top$$
$$(I+L_{t-1}^{t-r_t+1})H_{t-r_t}H_{t-r_t}^\top(I+L_{t-1}^{t-r_t+1})^\top(\spadesuit_2^{(1)}+\spadesuit_2^{(2)})\big]$$

We can now expand the right hand side of the last equation into different types of terms. We start by bounding the term that does not contain any $L_{t-1}^{t-r_t+1}$ nor $(L_{t-1}^{t-r_t+1})^\top$. It is easy to see that $\|\mathbb{E}\big[\spadesuit_2^{(1)}+\spadesuit_2^{(2)}|\mathcal{F}_{t-r_t}\big]\|\le\beta_t^2\mathcal{A}_t r_t*(\frac{101}{100})^3*4$. This follows because $\|\mathbb{E}\left[\epsilon_t|\mathcal{F}_{t-r_t}\right]\|\le \mathcal{A}_t\beta_t r_t$, and an operator bound on each of the remaining 3 terms in each of the four factors by $\|I+\beta_t B^{-1}A\|\le\frac{101}{100}$. With these observations and using the fact that $\spadesuit_2^{(1)}+\spadesuit_2^{(2)}$ is a symmetric matrix, we can bound the following term:

$$\mathbb{E}\Big[\mathrm{Tr}\Big(H_{t-r_t}H_{t-r_t}^\top H_{t-r_t}H_{t-r_t}^\top(\spadesuit_2^{(1)}+\spadesuit_2^{(2)})\Big)|\mathcal{F}_{t-r_t}\Big]$$
$$=\mathrm{Tr}\Big(H_{t-r_t}H_{t-r_t}^\top H_{t-r_t}H_{t-r_t}^\top\mathbb{E}\Big[\spadesuit_2^{(1)}+\spadesuit_2^{(2)}|\mathcal{F}_{t-r_t}\Big]\Big)$$
$$\le\beta_t^2\mathcal{A}_t r_t*\left(\frac{101}{100}\right)^3*4\,\mathrm{Tr}(H_{t-r_t}H_{t-r_t}^\top H_{t-r_t}H_{t-r_t}^\top)$$
$$\le\beta_t^2\mathcal{A}_t r_t*5\,\mathrm{Tr}(H_{t-r_t}H_{t-r_t}^\top H_{t-r_t}H_{t-r_t}^\top)$$

For the terms of $\Gamma$ containing $L_{t-1}^{t-r_t+1}$ components we use a simple bound. Notice that $\|\spadesuit_2^{(1)}+\spadesuit_2^{(2)}\|\le\beta_t(\frac{101}{100})^3*4*B_{\epsilon_t}$. And recall just as in Equation 40, $\|L_{t-1}^{t-r_t+1}\|\le 4r_t\mathcal{G}_t\beta_t$ and therefore $\|(L_{t-1}^{t-r_t+1})^\top L_{t-1}^{t-r_t+1}\|\le 16r_t^2\mathcal{G}_t^2\beta_t^2$. We look at the term containing four copies of $L_{t-1}^{t-r_t+1}$ terms:

Let $O_1=\mathrm{Tr}\Big(L_{t-1}^{t-r_t+1}H_{t-r_t}H_{t-r_t}^\top(L_{t-1}^{t-r_t+1})^\top L_{t-1}^{t-r_t+1}H_{t-r_t}H_{t-r_t}^\top(L_{t-1}^{t-r_t+1})^\top(\spadesuit_2^{(1)}+\spadesuit_2^{(1)})\Big)$.

$$O_1\le\beta_t(\frac{101}{100})^3*4*B_{\epsilon_t}\,\mathrm{Tr}\left(L_{t-1}^{t-r_t+1}H_{t-r_t}H_{t-r_t}^\top(L_{t-1}^{t-r_t+1})^\top L_{t-1}^{t-r_t+1}H_{t-r_t}H_{t-r_t}^\top(L_{t-1}^{t-r_t+1})^\top\right)$$
$$=\beta_t(\frac{101}{100})^3*4*B_{\epsilon_t}\,\mathrm{Tr}\left([H_{t-r_t}H_{t-r_t}^\top(L_{t-1}^{t-r_t+1})^\top L_{t-1}^{t-r_t+1}H_{t-r_t}H_{t-r_t}^\top](L_{t-1}^{t-r_t+1})^\top L_{t-1}^{t-r_t+1}\right)$$
$$\le\beta_t^3(\frac{101}{100})^3*4B_{\epsilon_t}*16r_t^2\mathcal{G}_t^2\,\mathrm{Tr}\left(H_{t-r_t}H_{t-r_t}^\top(L_{t-1}^{t-r_t+1})^\top L_{t-1}^{t-r_t+1}H_{t-r_t}H_{t-r_t}^\top\right)$$
$$=\beta_t^3(\frac{101}{100})^3*4B_{\epsilon_t}*16r_t^2\mathcal{G}_t^2\,\mathrm{Tr}\left(H_{t-r_t}H_{t-r_t}^\top H_{t-r_t}H_{t-r_t}^\top(L_{t-1}^{t-r_t+1})^\top L_{t-1}^{t-r_t+1}\right)$$
$$\le\beta_t^5(\frac{101}{100})^3*4B_{\epsilon_t}*16^2r_t^4\mathcal{G}_t^4\,\mathrm{Tr}\left(H_{t-r_t}H_{t-r_t}^\top H_{t-r_t}H_{t-r_t}^\top\right)$$
$$\le\beta_t^2 17B_{\epsilon_t}*r_t\mathcal{G}_t\,\mathrm{Tr}\left(H_{t-r_t}H_{t-r_t}^\top H_{t-r_t}H_{t-r_t}^\top\right)$$

where the last inequality follows from the step size conditions. We now look at the following term in $\Gamma$ that has three $L_{t-1}^{t-r_t+1}$ terms:

$$O_2=\mathrm{Tr}(H_{t-r_t}H_{t-r_t}^\top(L_{t-1}^{t-r_t+1})^\top L_{t-1}^{t-r_t+1}H_{t-r_t}H_{t-r_t}^\top(L_{t-1}^{t-r_t+1})^\top(\spadesuit_2^{(1)}+\spadesuit_2^{(2)}))+$$

$$\text{Tr}(L_{t-1}^{t-r_t+1} H_{t-r_t} H_{t-r_t}^\top (L_{t-1}^{t-r_t+1})^\top L_{t-1}^{t-r_t+1} H_{t-r_t} H_{t-r_t}^\top (\spadesuit_2^{(1)} + \spadesuit_2^{(2)}))$$

Since $\|(L_{t-1}^{t-r_t+1})^\top(\spadesuit_2^{(1)} + \spadesuit_2^{(2)}) + (\spadesuit_2^{(1)} + \spadesuit_2^{(2)})L_{t-1}^{t-r_t+1}\| \le \beta_t^2(\frac{101}{100})^3 B_{\epsilon_t} r_t \mathcal{G}_t$. Using a similar series of inequalities as in the case above we obtain a bound:

$$
\begin{aligned}
O_2 &= \text{Tr}\left(H_{t-r_t}H_{t-r_t}^\top(L_{t-1}^{t-r_t+1})^\top L_{t-1}^{t-r_t+1}H_{t-r_t}H_{t-r_t}^\top(L_{t-1}^{t-r_t+1})^\top(\spadesuit_2^{(1)} + \spadesuit_2^{(2)})+\right.\\
&\quad \left. L_{t-1}^{t-r_t+1}H_{t-r_t}H_{t-r_t}^\top(L_{t-1}^{t-r_t+1})^\top L_{t-1}^{t-r_t+1}H_{t-r_t}H_{t-r_t}^\top(\spadesuit_2^{(1)} + \spadesuit_2^{(2)})\right)\\
&= \text{Tr}\left(H_{t-r_t}H_{t-r_t}^\top(L_{t-1}^{t-r_t+1})^\top L_{t-1}^{t-r_t+1}H_{t-r_t}H_{t-r_t}^\top[(L_{t-1}^{t-r_t+1})^\top(\spadesuit_2^{(1)} + \spadesuit_2^{(2)}) + (\spadesuit_2^{(1)} + \spadesuit_2^{(2)})L_{t-1}^{t-r_t+1}]\right)\\
&\le \beta_t^2 * 32 * (\frac{101}{100})^3 B_{\epsilon_t}\mathcal{G}_t r_t \text{Tr}\left(H_{t-r_t}H_{t-r_t}^\top(L_{t-1}^{t-r_t+1})^\top L_{t-1}^{t-r_t+1}H_{t-r_t}H_{t-r_t}^\top\right)\\
&= \beta_t^2 * 32 * (\frac{101}{100})^3 B_{\epsilon_t}\mathcal{G}_t r_t \text{Tr}\left(H_{t-r_t}H_{t-r_t}^\top H_{t-r_t}H_{t-r_t}^\top(L_{t-1}^{t-r_t+1})^\top L_{t-1}^{t-r_t+1}\right)\\
&\le \beta_t^4 16 * 32(\frac{101}{100})^3 B_{\epsilon_t}\mathcal{G}_t^3 r_t^3 \text{Tr}\left(H_{t-r_t}H_{t-r_t}^\top H_{t-r_t}H_{t-r_t}^\top\right)\\
&\overset{\gamma_1}{\le} \beta_t^2 * 32(\frac{101}{100})^3 B_{\epsilon_t}\mathcal{G}_t r_t \text{Tr}\left(H_{t-r_t}H_{t-r_t}^\top H_{t-r_t}H_{t-r_t}^\top\right)\\
&\le \beta_t^2 * 33 B_{\epsilon_t}\mathcal{G}_t r_t \text{Tr}\left(H_{t-r_t}H_{t-r_t}^\top H_{t-r_t}H_{t-r_t}^\top\right)
\end{aligned}
$$

The last inequality $\gamma_1$ follows from the step size conditions.

$$
\begin{aligned}
O_2' &= \text{Tr}(L_{t-1}^{t-r_t+1}H_{t-r_t}H_{t-r_t}^\top(L_{t-1}^{t-r_t+1})^\top H_{t-r_t}H_{t-r_t}^\top(L_{t-1}^{t-r_t+1})^\top(\spadesuit_2^{(1)} + \spadesuit_2^{(2)}))+\\
&\quad \text{Tr}(L_{t-1}^{t-r_t+1}H_{t-r_t}H_{t-r_t}^\top L_{t-1}^{t-r_t+1}H_{t-r_t}H_{t-r_t}^\top(L_{t-1}^{t-r_t+1})^\top(\spadesuit_2^{(1)} + \spadesuit_2^{(2)}))
\end{aligned}
$$

Since $\|(L_{t-1}^{t-r_t+1})^\top(\spadesuit_2^{(1)} + \spadesuit_2^{(2)})L_{t-1}^{t-r_t+1}\| \le \beta_t^3(\frac{101}{100})^3 * 4 * B_{\epsilon_t} r_t^2 \mathcal{G}_t^2$:

$$
\begin{aligned}
O_2' &= \text{Tr}\left(L_{t-1}^{t-r_t+1}H_{t-r_t}H_{t-r_t}^\top(L_{t-1}^{t-r_t+1})^\top H_{t-r_t}H_{t-r_t}^\top(L_{t-1}^{t-r_t+1})^\top(\spadesuit_2^{(1)} + \spadesuit_2^{(2)})+\right.\\
&\quad \left. L_{t-1}^{t-r_t+1}H_{t-r_t}H_{t-r_t}^\top L_{t-1}^{t-r_t+1}H_{t-r_t}H_{t-r_t}^\top(L_{t-1}^{t-r_t+1})^\top(\spadesuit_2^{(1)} + \spadesuit_2^{(2)})\right)\\
&= \text{Tr}(H_{t-r_t}H_{t-r_t}^\top(L_{t-1}^{t-r_t+1} + (L_{t-1}^{t-r_t+1})^\top)H_{t-r_t}H_{t-r_t}^\top[(L_{t-1}^{t-r_t+1})^\top(\spadesuit_2^{(1)} + \spadesuit_2^{(2)})L_{t-1}^{t-r_t+1}])\\
&\overset{\gamma_1}{\le} \text{Tr}(H_{t-r_t}H_{t-r_t}^\top(L_{t-1}^{t-r_t+1} + (L_{t-1}^{t-r_t+1})^\top)H_{t-r_t}H_{t-r_t}^\top[(L_{t-1}^{t-r_t+1})^\top(\spadesuit_2^{(1)} + \spadesuit_2^{(2)})L_{t-1}^{t-r_t+1}])+\\
&\quad \text{Tr}(H_{t-r_t}H_{t-r_t}^\top\|(L_{t-1}^{t-r_t+1} + (L_{t-1}^{t-r_t+1})^\top)\|I H_{t-r_t}H_{t-r_t}^\top[(L_{t-1}^{t-r_t+1})^\top(\spadesuit_2^{(1)} + \spadesuit_2^{(2)})L_{t-1}^{t-r_t+1}])+\\
&\quad \text{Tr}(H_{t-r_t}H_{t-r_t}^\top\|(L_{t-1}^{t-r_t+1} + (L_{t-1}^{t-r_t+1})^\top)\|I H_{t-r_t}H_{t-r_t}^\top\|[(L_{t-1}^{t-r_t+1})^\top(\spadesuit_2^{(1)} + \spadesuit_2^{(2)})L_{t-1}^{t-r_t+1}]\|I)\\
&\overset{\gamma_2}{\le} \beta_t^3(\frac{101}{100})^3 * 4 * B_{\epsilon_t} r_t^2 \mathcal{G}_t^2 \text{Tr}(H_{t-r_t}H_{t-r_t}^\top(\|L_{t-1}^{t-r_t+1} + (L_{t-1}^{t-r_t+1})^\top\|I + L_{t-1}^{t-r_t+1}+\\
&\quad (L_{t-1}^{t-r_t+1})^\top)H_{t-r_t}H_{t-r_t}^\top) + \beta_t^3(\frac{101}{100})^3 * 4 * B_{\epsilon_t} r_t^2 \mathcal{G}_t^2 * 2 * 4r_t\mathcal{G}_t\beta_t \text{Tr}(H_{t-r_t}H_{t-r_t}^\top H_{t-r_t}H_{t-r_t}^\top)\\
&\le (\beta_t^3(\frac{101}{100})^3 * 4 * B_{\epsilon_t} r_t^2 \mathcal{G}_t^2 * 4 * 4r_t\mathcal{G}_t\beta_t + \beta_t^3(\frac{101}{100})^3 * 4 * B_{\epsilon_t} r_t^2 \mathcal{G}_t^2 * 2 * 4r_t\mathcal{G}_t\beta_t)\\
&\quad * \text{Tr}(H_{t-r_t}H_{t-r_t}^\top H_{t-r_t}H_{t-r_t}^\top)\\
&= \beta_t^4 * 72 * (\frac{101}{100})^3 r_t^3 \mathcal{G}_t^3 B_{\epsilon_t} \text{Tr}(H_{t-r_t}H_{t-r_t}^\top H_{t-r_t}H_{t-r_t}^\top)\\
&\le \beta_t^2 * 5 B_{\epsilon_t}\mathcal{G}_t r_t \text{Tr}(H_{t-r_t}H_{t-r_t}^\top H_{t-r_t}H_{t-r_t}^\top)
\end{aligned}
$$

where the inequality $\gamma_1$ follows because the sum of the two added terms is nonnegative. The inequality $\gamma_2$ follows by combining the first two terms in the previous expression and noting that $H_{t-r_t}H_{t-r_t}^\top H_{t-r_t}H_{t-r_t}^\top\|L_{t-1}^{t-r_t+1} + (L_{t-1}^{t-r_t+1})^\top\| + H_{t-r_t}H_{t-r_t}^\top(L_{t-1}^{t-r_t+1} + (L_{t-1}^{t-r_t+1})^\top)H_{t-r_t}H_{t-r_t}^\top \succeq 0$. The last inequality follows from the step size conditions. This finalizes the analysis for the components in $\Gamma$ having three $L_{t-1}^{t-r_t+1}$ terms.

We now look at the components of $\Gamma$ with two $L_{t-1}^{t-r_t+1}$ terms. Their sum equals:

$$\text{Tr}\left(((L_{t-1}^{t-r_t+1}H_{t-r_t}H_{t-r_t}^\top + H_{t-r_t}H_{t-r_t}^\top L_{t-1}^{t-r-t+1})^2 + (L_{t-1}^{t-r_t+1}H_{t-r_t}H_{t-r_t}^\top(L_{t-1}^{t-r_t+1})^\top + H_{t-r_t}H_{t-r_t}^\top)^2\right.$$

$$- H_{t-r_t} H_{t-r_t}^\top H_{t-r_t} H_{t-r_t}^\top - L_{t-1}^{t-r_t+1} H_{t-r_t} H_{t-r_t}^\top (L_{t-1}^{t-r_t+1})^\top L_{t-1}^{t-r_t+1} H_{t-r_t} H_{t-r_t}^\top (L_{t-1}^{t-r_t+1})^\top)(\spadesuit_2^{(1)} + \spadesuit_2^{(2)}))$$

We look at a generic term of $\Gamma$ having exactly two $L_{t-1}^{t-r_t+1}$ terms: Let

$$O_3 = \mathrm{Tr}(H_{t-r_t} H_{t-r_t}^\top (L_{t-1}^{t-r_t+1})^\top L_{t-1}^{t-r_t+1} H_{t-r_t} H_{t-r_t}^\top (\spadesuit_2^{(1)} + \spadesuit_2^{(2)})).$$

Then, we have that,

$$O_3 \leq d\|L_{t-1}^{t-r_t+1}\|^2 \|H_{t-r_t} H_{t-r_t}^\top\|^2 \|\spadesuit_2^{(1)} + \spadesuit_2^{(2)})\|$$

$$\leq d\beta_t^2 * 4^2 * r_t^2 * \mathcal{G}_t^2 * (\frac{101}{100})^3 * 4 * B_{\epsilon_t} \|(H_{t-r_t} H_{t-r_t}^\top)^2\|$$

$$\leq d\beta_t^2 17 \mathcal{G}_t B_{\epsilon_t} \mathrm{Tr}(H_{t-r_t} H_{t-r_t}^\top H_{t-r_t} H_{t-r_t}^\top)$$

The last inequality follows from a the step size conditions plus the fact that trace is larger than operator norm for a PSD matrix.

We now look at a generic term in $\Gamma$ with one $L_{t-1}^{t-r_t+1}$ term: Let

$$O_4 = \mathrm{Tr}(H_{t-r_t} H_{t-r_t}^\top (L_{t-1}^{t-r_t+1})^\top H_{t-r_t} H_{t-r_t}^\top (\spadesuit_2^{(1)} + \spadesuit_2^{(2)})).$$

Then, we have that,

$$O_4 \leq d\|L_{t-1}^{t-r_t+1}\| \|H_{t-r_t} H_{t-r_t}^\top\|^2 \|\spadesuit_2^{(1)} + \spadesuit_2^{(2)})\|$$

$$\leq d 4 r_t \mathcal{G}_t \beta_t (\frac{101}{100})^3 * 4 B_{\epsilon_t} \|(H_{t-r_t} H_{t-r_t}^\top)^2\|$$

$$\leq 17 d\beta_t^2 r_t \mathcal{G}_t B_{\epsilon_t} \mathrm{Tr}(H_{t-r_t} H_{t-r_t}^\top H_{t-r_t} H_{t-r_t}^\top)$$

Since there is a single term of type $O_1$, four of type $O_2$, six of type $O_3$ and four of type $O_4$, we obtain the bound whenever $t > t_0$:

$$\mathrm{Tr}\,\mathbb{E}\left[H_{t-1} H_{t-1}^\top H_{t-1} H_{t-1}^\top (\spadesuit_2^{(1)} + \spadesuit_2^{(2)})\right] \leq \beta_t^2 (5 r_t \mathcal{A}_t + 55 B_{\epsilon_t} \mathcal{G}_t r_t + 23 d\mathcal{G}_t B_{\epsilon_t} r_t) \cdot$$
$$\mathbb{E}\left[\mathrm{Tr}(H_{t-r_t} H_{t-r_t}^\top H_{t-r_t} H_{t-r_t}^\top)\right]$$

Therefore we obtain the following recursion:

$$\mathrm{Tr}(\mathbb{E}\left[H_t H_t^\top H_t H_t^\top\right]) \leq \mathrm{Tr}\,\mathbb{E}\left[H_{t-1} H_{t-1}^\top H_{t-1} H_{t-1}^\top \spadesuit\right]$$
$$\leq (1 + 4\beta_t \lambda_1 + 11\beta_t^2 \max(\lambda_1^4, 1) + 8\beta_t^2 B_{\epsilon_t}^2)\,\mathrm{Tr}\,\mathbb{E}\left[H_{t-1} H_{t-1}^\top H_{t-1} H_{t-1}^\top\right] +$$
$$\beta_t^2 (5 r_t \mathcal{A}_t + 55 B_{\epsilon_t} \mathcal{G}_t r_t + 23 d\mathcal{G}_t B_{\epsilon_t} r_t + 1(t \leq t_0) 2\beta_t (\frac{101}{100})^3 B_{\epsilon_t}) \cdot$$
$$\mathbb{E}\left[\mathrm{Tr}(H_{t-r_t} H_{t-r_t}^\top H_{t-r_t} H_{t-r_t}^\top)\right]$$

Let $C^{(3)}$ be a constant such that:

$$dr_t(\mathcal{A}_t + B_{\epsilon_t}^2 + B_{\epsilon_t} \mathcal{G}_t)C^{(3)} \geq (5 r_t \mathcal{A}_t + 55 B_{\epsilon_t} \mathcal{G}_t r_t + 23 d\mathcal{G}_t B_{\epsilon_t} r_t) + 11 \max(\lambda_1^4, 1) + 8 B_{\epsilon_t}^2$$

Let $\{\eta_i\}$ be a sequence of increasing upper bounds for $\mathbb{E}\left[\mathrm{Tr}(H_i H_i^\top H_i H_i^\top)\right]$. In other words,

$$\mathbb{E}\left[\mathrm{Tr}(H_i H_i^\top H_i H_i^\top)\right] \leq \eta_i \,\forall i$$

And $\eta_0 \leq \eta_1 \leq \eta_2 \leq \cdots$, where $\eta_0 = d$. Let $\mathcal{C}_t^{(3)} = E_\epsilon + \mathcal{D}_t^{(3)} + 11 \max(\lambda_1^4, 1)$. We can obtain a recursion of the form:

$$\eta_t \leq (1 + 4\beta_t \lambda_1 + \beta_t^2 dr_t(\mathcal{A}_t + B_{\epsilon_t}^2 + B_{\epsilon_t} \mathcal{G}_t)C^{(3)} + 1(t \leq t_0) 2\beta_t (\frac{101}{100})^3 B_{\epsilon_t})\eta_{t-1}$$

We conclude by applying the inequality $1 + x \leq \exp(x)$ for $x > 0$ and the initial condition $\eta_0 = d$:

$$\eta_t \leq d\exp(\sum_{i=1}^{t} 4\lambda_1 \beta_i + dr_i(\mathcal{A}_i + B_{\epsilon_i}^2 + B_{\epsilon_i} \mathcal{G}_i)C^{(3)} \beta_i^2 + \sum_{j=1}^{\min(t,t_0)} \beta_j 2 * (\frac{101}{100})^3 B_{\epsilon_j})$$

$\square$

**Lemma E.5.** *For $t > 0$, we have that*

$$\mathbb{E}\left[(\tilde{u}_i^\top H_t H_t^\top \tilde{u}_i)^2\right]$$

$$\leq \|\tilde{u}_1\|_2^4 \exp(\sum_{i=1}^t 4\lambda_1 \beta_i + 11\lambda_1^2\beta_t^2) + \|\tilde{u}_1\|_2^4 \sum_{i=1}^t \left( (\beta_i^2 dr_i(\mathcal{A}_i + B_{\epsilon_i}^2 + B_{\epsilon_i}\mathcal{G}_i)\mathcal{U}_2 + 1(i \leq t_0)\beta_i 4B_{\epsilon_i}) \cdot \right.$$

$$\left. \exp\left(\sum_{j=1}^i 4\lambda_1\beta_j + dr_j(\mathcal{A}_j + B_{\epsilon_j}^2 + \mathcal{G}_j B_{\epsilon_j})\mathcal{C}^{(3)}\beta_j^2 + \sum_{j=i}^{\min(t,t_0)} \beta_j 2 * (\frac{101}{100})^3 B_{\epsilon_j}\right)\right)$$

*where $\tilde{u}_1$ is the unnormalized left eigenvector corresponding to the maximum eigenvalue $\lambda_1$ of $B^{-1}A$. As long as $\beta_t$ follows that $\|I + \beta_t B^{-1}A\| \leq \frac{101}{100}, \beta_t B_{\epsilon_t} < 1$*

*Proof.* As in the previous lemma, we let $v = \tilde{u}_1/\|\tilde{u}_1\|_2$ denote the normalized left principal eigenvector. Let $H_t = (I + \beta_t G_t)H_{t-1} = (I + \beta_t B^{-1}A + \beta_t\epsilon_t)H_{t-1}$. The desired expectation can be written as:

$$\mathbb{E}\left[(v^\top H_t H_t^\top v)^2\right] = \mathbb{E}\left[v^\top(I + \beta_t G_t)H_{t-1}H_{t-1}^\top(I + \beta_t G_t)^\top vv^\top(I + \beta_t G_t)H_{t-1}H_{t-1}^\top(I + \beta_t G_t)^\top v\right]$$

$$= \mathbb{E}\left[\underbrace{v^\top(I + \beta_t B^{-1}A)H_{t-1}H_{t-1}^\top(I + \beta_t B^{-1}A)^\top vv^\top(I + \beta_t B^{-1}A)H_{t-1}H_{t-1}^\top(I + \beta_t B^{-1}A)^\top v}_{\Gamma_0}\right]$$

$$+ \mathbb{E}\left[\Gamma_1 + \Gamma_2 + \Gamma_3 + \Gamma_4\right]$$

where $\Gamma_i$ is the collection of terms in the expansion of $\mathbb{E}\left[(v^\top H_t H_t^\top v)^2\right]$ that have exactly $i$ terms of the form $\epsilon_t$.

Since $v$ is a left eigenvector of $B^{-1}A$, the term $\Gamma_0$ can be written as follows:

$$\mathbb{E}\left[\Gamma_0\right] = (1 + \beta_t\lambda_1)^4\mathbb{E}\left[v^\top H_{t-1}H_{t-1}^\top vv^\top H_{t-1}H_{t-1}^\top v\right]$$

$$\leq \exp(4\lambda_1\beta_t + 11\lambda_1^2\beta_t^2)\mathbb{E}\left[v^\top H_{t-1}H_{t-1}^\top vv^\top H_{t-1}H_{t-1}^\top v\right]$$

Now we bound the terms $\Gamma_i$ with $i \geq 2$. Each of these terms is formed of component terms with at least two $\beta_t\epsilon_t$ each. Let's look at a generic term like this one and bound it, for example one that has two terms of the form $\epsilon_t$:

$$|v^\top\beta_t\epsilon_t H_{t-1}H_{t-1}^\top\beta_t\epsilon_t vv^\top(I + \beta_t B^{-1}A)H_{t-1}H_{t-1}^\top(I + \beta_t B^{-1}A)^\top v| \leq \beta_t^2\|H_{t-1}H_{t-1}^\top\|^2 B_{\epsilon_t}^2\left(\frac{101}{100}\right)^2$$

$$\leq 2B_{\epsilon_t}^2\beta_t^2 \operatorname{Tr}(H_{t-1}H_{t-1}^\top H_{t-1}H_{t-1}^\top)$$

By a similar argument, and using the step size conditions $\beta_t\mathcal{B}_{\epsilon_t} < 1$, we can bound each of the terms in $\Gamma_2, \Gamma_3$ and $\Gamma_4$ and obtain (using the fact that $\beta_t < 1$):

$$\Gamma_2 + \Gamma_3 + \Gamma_4 \leq \beta_t^2 B_{\epsilon_t}^2\mathcal{U}_1 \operatorname{Tr}(H_{t-1}H_{t-1}^\top H_{t-1}H_{t-1}^\top) \tag{50}$$

For some universal constant $\mathcal{U}_1$ depending on $\frac{101}{100}$ and the number of component terms in $\Gamma_2, \Gamma_3$, and $\Gamma_4$. Therefore,

$$\mathbb{E}\left[\Gamma_2 + \Gamma_3 + \Gamma_4\right] \leq \beta_t^2 B_{\epsilon_t}^2\mathcal{U}_1\mathbb{E}\left[\operatorname{Tr}(H_{t-1}H_{t-1}^\top H_{t-1}H_{t-1}^\top)\right]$$

$$\leq \beta_t^2 B_{\epsilon-t}^2\mathcal{U}_1 d \exp\left(\sum_{i=1}^{t-1} 4\lambda_1\beta_i + dr_i(\mathcal{A}_i + B_{\epsilon_i}^2 + B_{\epsilon_i}\mathcal{G}_i)\mathcal{C}^{(3)}\beta_i^2\right.$$

$$\left. + \sum_{j=1}^{\min(t,t_0)} 2 * \beta_j(\frac{101}{100})^3 B_{\epsilon_j}\right)$$

**Bounding expectation of $\Gamma_1$:** We start by bounding the expectation of $\Gamma_1$ whenever $t \leq t_0$. Let's look at a generic term from $\Gamma_1$:

$$\mathcal{Z} := v^\top(I + \beta_t B^{-1}A)H_{t-1}H_{t-1}^\top\beta_t\epsilon_t vv^\top(I + \beta_t B^{-1}A)H_{t-1}H_{t-1}^\top(I + \beta_t B^{-1}A)^\top v$$

We bound this term naively:

$$\|\mathcal{Z}\| \leq \beta_t \|I + \beta_t B^{-1} A\|^3 \|H_{t-1} H_{t-1}^\top\|^2 B_{\epsilon_t}$$

$$\leq \beta_t \left(\frac{101}{100}\right)^3 \operatorname{Tr}(H_{t-1} H_{t-1}^\top H_{t-1} H_{t-1}^\top) B_{\epsilon_t}$$

There are exactly 4 terms of type $\mathcal{Z}$. Now we proceed to bound the expectation of $\Gamma_1$ whenever $t > t_0$: Let's look at a generic term from $\Gamma_1$:

$$v^\top (I + \beta_t B^{-1} A) H_{t-1} H_{t-1}^\top \beta_t \epsilon_t v v^\top (I + \beta_t B^{-1} A) H_{t-1} H_{t-1}^\top (I + \beta_t B^{-1} A)^\top v \quad (51)$$

In the same way as in previous lemmas, in order to obtain a bound for this term, we write $H_{t-1} = (I + L_{t-1}^{t-r_t+1}) H_{t-r_t}$ and substitute this equality in Equation 51. Recall that $\|L_{t-1}^{t-r_t+1}\| \leq 4 r_t \mathcal{G}_t \beta_t$. In this expansion, we bound all terms that have at least one $L_{t-1}^{t-r_t+1}$ using a simple bound. Let's look at a generic such term and bound it:

$$\spadesuit := |v^\top (I + \beta_t B^{-1} A) L_{t-1}^{t-r_t+1} H_{t-r_t} H_{t-r_t}^\top \beta_t \epsilon_t v v^\top (I + \beta_t B^{-1} A) H_{t-r_t} H_{t-r_t}^\top (I + \beta_t B^{-1} A)^\top v| \quad (52)$$

$$\spadesuit \leq 4 r_t \mathcal{G}_t \beta_t^2 \left(\frac{101}{100}\right)^3 \cdot \|H_{t-r_t} H_{t-r_t}^\top\|^2 B_\epsilon$$

$$\leq 2 r_t \mathcal{G}_t \beta_t^2 B_{\epsilon_t} \left(\frac{101}{100}\right)^3 \cdot \operatorname{Tr}(H_{t-r_t} H_{t-r_t}^\top H_{t-r_t} H_{t-r_t}^\top)$$

And therefore:

$$\mathbb{E}\left[\spadesuit\right] \leq 2 r_t \mathcal{G}_t \beta_t^2 B_{\epsilon_t} \left(\frac{101}{100}\right)^3 d \exp\left(\sum_{i=1}^{t-r_t} 4\lambda_1 \beta_i + d r_i (\mathcal{A}_i + B_{\epsilon_i}^2 + B_{\epsilon_i} \mathcal{G}_i) \mathcal{C}^{(3)} \beta_i^2\right)$$

$$\leq 2 r_t \mathcal{G}_t \beta_t^2 B_{\epsilon_t} \left(\frac{101}{100}\right)^3 d \exp\left(\sum_{i=1}^{t-1} 4\lambda_1 \beta_i + d r_i (\mathcal{A}_i + B_{\epsilon_i}^2 + B_{\epsilon_i} \mathcal{G}_i) \mathcal{C}^{(3)} \beta_i^2\right)$$

Using the step size condition, $\beta_t \mathcal{G}_t r_t \leq \frac{1}{4}$, all of the remaining terms with at least one $L_{t-1}^{t-r_t+1}$ can be upper bounded by a expression of order $O(\beta_t^2 r_t \mathcal{G}_t B_{\epsilon_t} \operatorname{Tr}(H_{t-r_t} H_{t-r_t}^\top H_{t-r_t} H_{t-r_t}^\top))$. This procedure will handle the terms in $\Gamma_1$ that after the subsitution $H_{t-1} = (I + L_{t-1}^{t-r_t+1}) H_{t-r_t}$ have at least one $L_{t-1}^{t-r_t+1}$.

The only terms remaining to bound are those coming from $\Gamma_1$, such that after substituting $H_{t-1} = (I + L_{t-1}^{t-r_t+1}) H_{t-r_t}$ do not involve any $L_{t-1}^{t-r_t+1}$. Let's look at a generic such term and bound its expectation:

$$\diamondsuit := \mathbb{E}\left[\underbrace{v^\top (I + \beta_t B^{-1} A) H_{t-r_t} H_{t-r_t}^\top \beta_t \epsilon_t v v^\top (I + \beta_t B^{-1} A) H_{t-r_t} H_{t-r_t}^\top (I + \beta_t B^{-1} A)^\top v}_{\diamondsuit_1}\right]$$

Recall that $\|\mathbb{E}[\epsilon_t | \mathcal{F}_{t-r_t}]\| \leq \mathcal{A}_t \beta_t r_t$. We bound $\diamondsuit$ by first bounding the norm of the conditional expectation of $\diamondsuit_1$:

$$\|\mathbb{E}\left[\diamondsuit_1 | \mathcal{F}_{t-r_t}\right]\| \leq \beta_t^2 O(r_t) \left(\frac{101}{100}\right)^3 \|H_{t-r_t} H_{t-r_t}^\top\|^2$$

$$\leq \beta_t^2 \mathcal{A}_t r_t \left(\frac{101}{100}\right)^3 \operatorname{Tr}(H_{t-r_t} H_{t-r_t}^\top H_{t-r_t} H_{t-r_t}^\top)$$

And therefore:

$$\diamondsuit = \mathbb{E}\left[\diamondsuit_1\right] \leq \mathbb{E}\left[\|\mathbb{E}\left[\diamondsuit_1 | \mathcal{F}_{t-r_t}\right]\|\right]$$

$$\leq \beta_t^2 \mathcal{A}_t r_t \left(\frac{101}{100}\right)^3 d \exp\left(\sum_{i=1}^{t-r_t} 4\lambda_1 \beta_i + d r_i (\mathcal{A}_i + B_{\epsilon_i}^2 + B_{\epsilon_i} \mathcal{G}_i) \mathcal{C}^{(3)} \beta_i^2 + \sum_{j=1}^{\min(t-r_t, t_0)} \beta_j 2 * (\frac{101}{100})^3 B_{\epsilon_j}\right)$$

$$\leq \beta_t^2 \mathcal{A}_t r_t \left(\frac{101}{100}\right)^3 d \exp\left(\sum_{i=1}^{t-1} 4\lambda_1 \beta_i + dr_i(\mathcal{A}_i + B_{\epsilon_i}^2 + B_{\epsilon_i}\mathcal{G}_i)\mathcal{C}^{(3)}\beta_i^2 + \sum_{j=1}^{\min(t,t_0)} \beta_j 2 * (\frac{101}{100})^3 B_{\epsilon_j}\right)$$

The last inequality follows from the results of E.4. Combining all these bounds yields for all $t$ we have:

$$\mathbb{E}\left[\Gamma_1 + \Gamma_2 + \Gamma_3 + \Gamma_4\right] \leq \left(\beta_t^2 dr_t(\mathcal{A}_t + B_{\epsilon_t}^2 + B_{\epsilon_t}\mathcal{G}_t)\mathcal{U}_2 + 1(t \leq t_0)\beta_t 4 * B_{\epsilon_t}(\frac{101}{100})^3\right)$$
$$\exp\left(\sum_{i=1}^{t-1} 4\lambda_1 \beta_i + dr_i(\mathcal{A}_i + B_{\epsilon_i}^2 + B_{\epsilon_i}\mathcal{G}_i)\mathcal{C}^{(3)}\beta_i^2 + \sum_{j=1}^{\min(t,t_0)} \beta_j 2 * (\frac{101}{100})^3\right)$$

where $\mathcal{U}_2$ is an absolute constant depending on $\frac{101}{100}$, and the number of terms in $\Gamma_1, \Gamma_2, \cdots, \Gamma_4$.
Combining all these terms we get a recursion of the form:

$$\mathbb{E}\left[(v^\top H_t H_t^\top v)^2\right] \leq \exp(4\lambda_1\beta_t + 11\lambda_1^2\beta_t^2)\mathbb{E}\left[(v^\top H_{t-1}H_{t-1}^\top v)^2\right] + \left(\beta_t^2 dr_t(\mathcal{A}_t + B_{\epsilon_t}^2 + B_{\epsilon_t}\mathcal{G}_t)\mathcal{U}_2 + \right.$$
$$1(t \leq t_0)\beta_t B_{\epsilon_t}(\frac{101}{100})^3\right) \exp\left(\sum_{i=1}^{t-1} 4\lambda_1\beta_i + dr_i(\mathcal{A}_i + B_{\epsilon_i}^2 + B_{\epsilon_i}\mathcal{G}_i)\mathcal{C}^{(3)}\beta_i^2\right.$$
$$\left. + \sum_{j=1}^{\min(t,t_0)} \beta_j(\frac{101}{100})^3 B_{\epsilon_j}\right)$$

After applying recursion on this equation we obtain:

$$\mathbb{E}\left[(v^\top H_t H_t^\top v)^2\right] \leq \exp(\sum_{i=1}^{t} 4\lambda_1\beta_i + 11\lambda_1^2\beta_t^2)$$
$$+ \sum_{i=1}^{t}\left(\beta_i^2 dr_i(\mathcal{A}_i + B_{\epsilon_i}^2 + B_{\epsilon_i}\mathcal{G}_i)\mathcal{U}_2 + 1(i \leq t_0)\beta_i 4 B_{\epsilon_i}(\frac{101}{100})^3\right) \exp\left(\sum_{j=1}^{i} 4\lambda_1\beta_j + \right.$$
$$dr_j(\mathcal{A}_j + B_{\epsilon_j}^2 + \mathcal{G}_j B_{\epsilon_j})\mathcal{C}^{(3)}\beta_j^2 + \sum_{j=i}^{\min(t,t_0)} \beta_j 2(\frac{101}{100})^3 B_{\epsilon_j}\right)$$

As desired. $\qquad\qquad\square$

# F  Convergence Analysis and Main Result

We reproduce the bounds that we will be requiring in this section from the previous ones. We begin by reporducing the lower bound of Lemma 5.3.

$$\frac{\mathbb{E}[\tilde{u}_1^\top H_n H_n^\top \tilde{u}_1]}{\|\tilde{u}_1\|_2^2} \geq \exp\left(\sum_{t=1}^{n} 2\beta_t\lambda_1 - 4\beta_t^2\lambda_1^2\right) -$$
$$\underbrace{d\sum_{t=1}^{n} c_1\left((\beta_t^2 r_t + \beta_t\mathbb{I}(t \leq t_0))\exp\left(\sum_{i=1}^{t} 2\beta_i\lambda_1 + c_2\beta_i^2 dr_i + c_3\sum_{i=1}^{t_0}\beta_i d\right)\right)}_{(I)},$$
$$\tag{53}$$

where we have merged previous explicit constants into $c_1$, $c_2$ and $c_3$, which throughout the course of this section might assume different values. Restating the bound from Lemma 5.4, we have,

$$\frac{\mathbb{E}\left[(\tilde{u}_1^\top H_n H_n^\top \tilde{u}_1)^2\right]}{\|\tilde{u}_1\|_2^4} \leq \exp\left(\sum_{t=1}^n 4\lambda_1\beta_t + 11\lambda_1^2\beta_t^2\right) +$$

$$\underbrace{c_1\sum_{t=1}^n\left((d\beta_t^2 r_t + \mathbb{I}(t \leq t_0)\beta_t)\exp\left(\sum_{i=1}^t 4\lambda_1\beta_i + c_2 dr_i\beta_i^2 + c_3\sum_{i=1}^{t_0}\beta_i\right)\right)}_{(II)}.$$

(54)

Note that as mentioned before in Section B, the term $r_t = O(\log^3(\beta_t^{-1}))$ and $t_0 = O(\log^3(d^2\beta))$. In the following, we substitute the step size $\beta_t = \frac{b}{d^2\beta+t}$, where $b, \beta$ are constants, implying that $r_t = O(\log^3(d^2\beta+t))$.

**Bounds on partial sums of series**: We begin by obtaining bounds on partial sums of some series which will be useful in our analysis. We first prove the following upper bound:

$$\sum_{i=1}^t 4\beta_i\lambda_1 = 4b\lambda_1\sum_{i=1}^t \frac{1}{d^2\beta+i} = 4b\lambda_1\sum_{i=d^2\beta+1}^{d^2\beta+t} \frac{1}{i} \leq 4b\lambda_1\log\left(\frac{d^2\beta+t}{d^2\beta}\right).$$

(55)

We next have the following lower bound:

$$\sum_{i=1}^t 4\beta_i\lambda_1 = 4b\lambda_1\sum_{i=1}^t \frac{1}{d^2\beta+i} = 4b\lambda_1\sum_{i=d^2\beta+1}^{d^2\beta+t} \frac{1}{i} \geq 4b\lambda_1\log\left(\frac{d^2\beta+t+1}{d^2\beta+1}\right).$$

(56)

We can obtain the following bound on the squared terms:

$$c\sum_{i=1}^t \beta_i^2\log^3(d^2\beta+i) = c\sum_{i=1}^t \frac{\log^3(d^2\beta+i)}{(d^2\beta+i)^2}$$

$$= c\sum_{i=d^2\beta+1}^{d^2\beta+t} \frac{\log^3(i)}{i^2} \leq c\int_{d^2\beta}^\infty \frac{\log^3(x)}{x^2}dx \leq c\frac{\log^3(d\beta)}{d^2\beta},$$

where $c$ is a constant which changes with inequality. Next, we proceed by bounding the excess terms in the exponent corresponding to the summation over the $t_0$ terms.

$$c\sum_{i=1}^{t_0}\beta_i \leq cb\log\left(\frac{d^2\beta+t_0}{d^2\beta}\right) \leq \frac{ct_0}{d^2\beta} \leq \frac{c\log^3(d\beta)}{d^2\beta} \leq \frac{c}{d},$$

(57)

where the last inequality follows since $\frac{log^3(x)}{(x)} \leq 2$.

**Bounds on $\mathbb{E}[v^\top H_n H_n^\top v]$ and $\mathbb{E}[(v^\top H_n H_n^\top v)^2]$**: We first proceed by providing upper bounds on Term $(I)$ in (53) and Term $(II)$ in (54).

$$d\sum_{t=1}^n c_1\left((\beta_t^2 r_t + \beta_t\mathbb{I}(t \leq t_0))\exp\left(\sum_{i=1}^t 2\beta_i\lambda_1 + c_2\beta_i^2 dr_i + c_3\sum_{i=1}^{t_0}\beta_i d\right)\right) \leq cd\sum_{t=1}^n\left((\beta_t^2 r_t + \beta_t\mathbb{I}(t \leq t_0))\right.$$

$$\left. *\left(\frac{d^2\beta+t}{d^2\beta}\right)^{2b\lambda_1}\right).$$

Similarly term $(II)$ by:

$$c_1\sum_{t=1}^n\left((d\beta_t^2 r_t + \mathbb{I}(t \leq t_0)\beta_t)\exp\left(\sum_{i=1}^t 4\lambda_1\beta_i + c_2 dr_i\beta_i^2 + c_3\sum_{i=1}^{t_0}\beta_i\right)\right)$$

$$\leq c\sum_{t=1}^n(d\beta_t^2 r_t + \beta_t\mathbb{I}(t \leq t_0))\left(\frac{d^2\beta+t}{d^2\beta}\right)^{4b\lambda_1}.$$

**Lemma F.1.** *For any $\delta_1 \in (0,1)$ and $n$ satisfying,*

$$\frac{d^2\beta + n}{\log^{\frac{4}{\min(1,2b\lambda_1)}}(d^2\beta + n)} \geq \max\left(\left(\frac{\exp(\frac{c\lambda_1^2}{d^2})}{\delta_1}\right)^{1/2b\lambda_1}(d^2\beta + 1),\right.$$

$$\left.\frac{cd\beta^{2b\lambda_1}}{\delta_1}\exp\left(\frac{c\lambda_1^2}{d^2}\right)\left(\left(1 + \frac{1}{d^2\beta}\right)^{2b\lambda_1} + d^2\beta\right), \frac{c\beta^2 d^3 \exp\left(\frac{c\lambda_1^2}{d^2}\right)}{\delta_1}\right)$$

*we have that*

$$\frac{\mathbb{E}[\tilde{u}_1^\top H_n H_n^\top \tilde{u}_1]}{\|\tilde{u}_1\|_2^2} \geq (1 - \delta_1)\exp\left(\sum_{t=1}^n 2\beta_t\lambda_1 - 4\beta_t^2\lambda_1^2\right),$$

*where $c$ depends polynomially on $b, \beta, \lambda_1$.*

*Proof.* We consider the term $\mathbb{E}[\tilde{u}_1^\top H_n H_n^\top \tilde{u}_1]$ from Equation (53),

$$\frac{\mathbb{E}[\tilde{u}_1^\top H_n H_n^\top \tilde{u}_1]}{\|\tilde{u}_1\|_2^2} \geq \exp\left(\sum_{t=1}^n 2\beta_t\lambda_1 - 4\beta_t^2\lambda_1^2\right) - cd\sum_{t=1}^n(\beta_t^2 r_t + \beta_t\mathbb{I}(t \leq t_0))\left(\frac{d^2\beta + t}{d^2\beta}\right)^{2b\lambda_1}$$

$$= (1 - \delta_1)\exp\left(\sum_{t=1}^n 2\beta_t\lambda_1 - 4\beta_t^2\lambda_1^2\right) - cd\sum_{t=1}^n(\beta_t^2 r_t + \beta_t\mathbb{I}(t \leq t_0))\left(\frac{d^2\beta + t}{d^2\beta}\right)^{2b\lambda_1}$$

$$+ \delta_1\exp\left(\sum_{t=1}^n 2\beta_t\lambda_1 - 4\beta_t^2\lambda_1^2\right)$$

$$\geq (1 - \delta_1)\exp\left(\sum_{t=1}^n 2\beta_t\lambda_1 - 4\beta_t^2\lambda_1^2\right) - \frac{c}{d^{4b\lambda_1 - 1}}\sum_{t=1}^n(\beta_t^2 r_t + \beta_t\mathbb{I}(t \leq t_0))\left(d^2\beta + t\right)^{2b\lambda_1}$$

$$+ \delta_1\exp\left(-\frac{c'\lambda_1^2}{d^2}\right)\left(\frac{d^2\beta + n + 1}{d^2\beta + 1}\right)^{2b\lambda_1}$$

$$\geq (1 - \delta_1)\exp\left(\sum_{t=1}^n 2\beta_t\lambda_1 - 4\beta_t^2\lambda_1^2\right) - \frac{c}{d^{4b\lambda_1 - 1}}\sum_{t=1}^n(\beta_t^2 r_t)\left(d^2\beta + t\right)^{2b\lambda_1}$$

$$+ \delta_1\exp\left(-\frac{c'\lambda_1^2}{d^2}\right)\left(\frac{d^2\beta + n + 1}{d^2\beta + 1}\right)^{2b\lambda_1} - \frac{c}{d^{4b\lambda_1 - 1}}\sum_{t=1}^{t_0}\left(d^2\beta + t\right)^{2b\lambda_1 - 1}$$

$$\overset{\zeta_1}{\geq} (1 - \delta_1)\exp\left(\sum_{t=1}^n 2\beta_t\lambda_1 - 4\beta_t^2\lambda_1^2\right) - \frac{c}{d^{4b\lambda_1 - 1}}\sum_{t=1}^n(\beta_t^2 r_t)\left(d^2\beta + t\right)^{2b\lambda_1}$$

$$+ \delta_1\exp\left(-\frac{c'\lambda_1^2}{d^2}\right)\left(\frac{d^2\beta + n + 1}{d^2\beta + 1}\right)^{2b\lambda_1} - \frac{c}{2b\lambda_1 d^{4b\lambda_1 - 1}}\left(d^2\beta + \log^3(d\beta)\right)^{2b\lambda_1}$$

$$\overset{\zeta_2}{\geq} (1 - \delta_1)\exp\left(\sum_{t=1}^n 2\beta_t\lambda_1 - 4\beta_t^2\lambda_1^2\right) - \frac{c}{d^{4b\lambda_1 - 1}}\sum_{t=1}^n(\beta_t^2 r_t)\left(d^2\beta + t\right)^{2b\lambda_1}$$

$$+ \delta_1\exp\left(-\frac{c'\lambda_1^2}{d^2}\right)\left(\frac{d^2\beta + n + 1}{d^2\beta + 1}\right)^{2b\lambda_1} - \frac{c\beta^{2b\lambda_1}d^{4b\lambda_1}}{d^{4b\lambda_1 - 1}}$$

$$\geq (1 - \delta_1)\exp\left(\sum_{t=1}^n 2\beta_t\lambda_1 - 4\beta_t^2\lambda_1^2\right) - \frac{c}{d^{4b\lambda_1 - 1}}\sum_{t=1}^n\log^3(d^2\beta + t)\left(d^2\beta + t\right)^{2b\lambda_1 - 2}$$

$$+ \delta_1\exp\left(-\frac{c'\lambda_1^2}{d^2}\right)\left(\frac{d^2\beta + n + 1}{d^2\beta + 1}\right)^{2b\lambda_1} - c\beta^{2b\lambda_1}d$$

$$\geq (1 - \delta_1)\exp\left(\sum_{t=1}^n 2\beta_t\lambda_1 - 4\beta_t^2\lambda_1^2\right) - \frac{c\log^3(d^2\beta + n)}{d^{4b\lambda_1 - 1}}\sum_{t=1}^n\left(d^2\beta + t\right)^{2b\lambda_1 - 2}$$

$$+ \delta_1 \exp\left(-\frac{c'\lambda_1^2}{d^2}\right)\left(\frac{d^2\beta + n + 1}{d^2\beta + 1}\right)^{2b\lambda_1} - c\beta^{2b\lambda_1} d,$$

where $\zeta_1$ from using $\sum_{i=1}^n i^\gamma \le n^{\gamma+1}/\gamma + 1$ for $\gamma > -1$ and $\zeta_2$ follows from the fact that $\log^3(x) \le cx$. We now consider the following three cases:

**Case 1:** $2b\lambda_1 < 1$

In this case we can lower bound the term $\frac{\mathbb{E}[\tilde{u}_1^\top H_n H_n^\top \tilde{u}_1]}{\|\tilde{u}_1\|_2^2}$ as,

$$\frac{\mathbb{E}[\tilde{u}_1^\top H_n H_n^\top \tilde{u}_1]}{\|\tilde{u}_1\|_2^2} \ge (1 - \delta_1) \exp\left(\sum_{t=1}^n 2\beta_t \lambda_1 - 4\beta_t^2 \lambda_1^2\right) - \frac{c \log^3(d^2\beta + n)}{d^{4b\lambda_1 - 1}(d^2\beta)^{(1-2b\lambda_1)}}$$

$$+ \delta_1 \exp\left(-\frac{c'\lambda_1^2}{d^2}\right)\left(\frac{d^2\beta + n + 1}{d^2\beta + 1}\right)^{2b\lambda_1} - c\beta^{2b\lambda_1} d$$

$$\ge (1 - \delta_1) \exp\left(\sum_{t=1}^n 2\beta_t \lambda_1 - 4\beta_t^2 \lambda_1^2\right) - \frac{c\beta^{2b\lambda_1} \log^3(d^2\beta + n)}{d\beta}$$

$$+ \delta_1 \exp\left(-\frac{c'\lambda_1^2}{d^2}\right)\left(\frac{d^2\beta + n + 1}{d^2\beta + 1}\right)^{2b\lambda_1} - c\beta^{2b\lambda_1} d$$

$$\ge (1 - \delta_1) \exp\left(\sum_{t=1}^n 2\beta_t \lambda_1 - 4\beta_t^2 \lambda_1^2\right) - cd\beta^{2b\lambda_1} \log^3(d^2\beta + n)$$

$$+ \delta_1 \exp\left(-\frac{c'\lambda_1^2}{d^2}\right)\left(\frac{d^2\beta + n + 1}{d^2\beta + 1}\right)^{2b\lambda_1}$$

$$\overset{\zeta_1}{\ge} (1 - \delta_1) \exp\left(\sum_{t=1}^n 2\beta_t \lambda_1 - 4\beta_t^2 \lambda_1^2\right),$$

where $\zeta_1$ follows by using that

$$\frac{d^2\beta + n}{\log^{3/2b\lambda_1}(d^2\beta + n)} \ge \left(\frac{cd}{\delta_1}\right)^{1/2b\lambda_1}(d^2\beta + 1).$$

**Case 2:** $2b\lambda_1 > 1$

In this case, we can lower bound the term $\frac{\mathbb{E}[\tilde{u}_1^\top H_n H_n^\top \tilde{u}_1]}{\|\tilde{u}_1\|_2^2}$ as,

$$\frac{\mathbb{E}[\tilde{u}_1^\top H_n H_n^\top \tilde{u}_1]}{\|\tilde{u}_1\|_2^2} \ge (1 - \delta_1) \exp\left(\sum_{t=1}^n 2\beta_t \lambda_1 - 4\beta_t^2 \lambda_1^2\right) - \frac{c \log^3(d^2\beta + n)}{d^{4b\lambda_1 - 1}}\frac{(d^2\beta + n)^{2b\lambda_1 - 1}}{2b\lambda_1 - 1}$$

$$+ \delta_1 \exp\left(-\frac{c'\lambda_1^2}{d^2}\right)\left(\frac{d^2\beta + n + 1}{d^2\beta + 1}\right)^{2b\lambda_1} - c\beta^{2b\lambda_1} d$$

$$\ge \left(\frac{d^2\beta + n}{d^2\beta + 1}\right)^{2b\lambda_1}\left(\delta_1 \exp\left(-\frac{c'\lambda_1^2}{d^2}\right) - c\beta^{2b\lambda_1} d\left(\frac{d^2\beta + 1}{d^2\beta + n}\right)\right)$$

$$- cd\beta^{2b\lambda_1}\left(1 + \frac{1}{d^2\beta}\right)^{2b\lambda_1}\frac{\log^3(d^2\beta + n)}{d^2\beta + n}\right) + (1 - \delta_1) \exp\left(\sum_{t=1}^n 2\beta_t \lambda_1 - 4\beta_t^2 \lambda_1^2\right)$$

$$\overset{\zeta_1}{\ge} (1 - \delta_1) \exp\left(\sum_{t=1}^n 2\beta_t \lambda_1 - 4\beta_t^2 \lambda_1^2\right),$$

where $\zeta_1$ follows by using that

$$\frac{d^2\beta + n}{\log^3(d^2\beta + n)} \ge \frac{cd\beta^{2b\lambda_1}}{\delta_1} \exp\left(\frac{c\lambda_1^2}{d^2}\right)\left(\left(1 + \frac{1}{d^2\beta}\right)^{2b\lambda_1} + d^2\beta\right)$$

**Case 3:** $2b\lambda_1 = 1$

In this case, we can lower bound the term $\frac{\mathbb{E}[\tilde{u}_1^\top H_n H_n^\top \tilde{u}_1]}{\|\tilde{u}_1\|_2^2}$ as,

$$\frac{\mathbb{E}[\tilde{u}_1^\top H_n H_n^\top \tilde{u}_1]}{\|\tilde{u}_1\|_2^2} \geq (1-\delta_1)\exp\left(\sum_{t=1}^n 2\beta_t\lambda_1 - 4\beta_t^2\lambda_1^2\right) - \frac{c\log^4(d^2\beta + n)}{d}$$

$$+ \delta_1 \exp\left(-\frac{c'\lambda_1^2}{d^2}\right)\left(\frac{d^2\beta + n + 1}{d^2\beta + 1}\right) - c\beta d$$

$$\overset{\zeta_1}{\geq} (1-\delta_1)\exp\left(\sum_{t=1}^n 2\beta_t\lambda_1 - 4\beta_t^2\lambda_1^2\right),$$

where $\zeta_1$ follows from using

$$\frac{d^2\beta + n}{\log^4(d^2\beta + n)} \geq \frac{c\beta^2 d^3 \exp\left(\frac{c\lambda_1^2}{d^2}\right)}{\delta_1}.$$

$\square$

**Lemma F.2.** *For any $\delta_2 \in (0,1)$ and $n$ satisfying,*

$$\frac{d^2\beta + n}{\log^{4\min(1, 1/4b\lambda_1)}(d^2\beta + n)} \geq \max\left(\frac{c(d^2\beta + 1)}{(\delta_2\log^3(d\beta))^{\frac{1}{4b\lambda_1}}}, \frac{c^{4b\lambda_1}}{\delta_2}(d^2\beta + n)\right),$$

*we have that,*

$$\frac{\mathbb{E}\left[(\tilde{u}_1^\top H_n H_n^\top \tilde{u}_1)^2\right]}{\|\tilde{u}_1\|_2^4} \leq (1+\delta_2)\exp\left(\sum_{t=1}^n 4\lambda_1\beta_t + 11\lambda_1^2\beta_t^2\right),$$

*where $c$ depends polynomially on $b, \beta, \lambda_1, \Delta_\lambda$.*

*Proof.* We consider the term $\mathbb{E}\left[(\tilde{u}_1^\top H_n H_n^\top \tilde{u}_1)^2\right]$ from Equation (54),

$$\frac{\mathbb{E}\left[(\tilde{u}_1^\top H_n H_n^\top \tilde{u}_1)^2\right]}{\|\tilde{u}_1\|_2^4} \leq \exp\left(\sum_{t=1}^n 4\lambda_1\beta_t + 11\lambda_1^2\beta_t^2\right) + c\sum_{t=1}^n(d\beta_t^2 r_t + \beta_t\mathbb{I}(t \leq t_0))\left(\frac{d^2\beta + t}{d^2\beta}\right)^{4b\lambda_1}$$

$$= c\sum_{t=1}^n(d\beta_t^2 r_t + \beta_t\mathbb{I}(t \leq t_0))\left(\frac{d^2\beta + t}{d^2\beta}\right)^{4b\lambda_1} - \delta_2\exp\left(\sum_{t=1}^n 4\lambda_1\beta_t + 11\lambda_1^2\beta_t^2\right)$$

$$+ (1+\delta_2)\exp\left(\sum_{t=1}^n 4\lambda_1\beta_t + 11\lambda_1^2\beta_t^2\right)$$

$$= c\sum_{t=1}^n(d\beta_t^2 r_t)\left(\frac{d^2\beta + t}{d^2\beta}\right)^{4b\lambda_1} - \delta_2\exp\left(\sum_{t=1}^n 4\lambda_1\beta_t + 11\lambda_1^2\beta_t^2\right)$$

$$+ cb\sum_{t=1}^{t_0}\frac{1}{d^2\beta + t}\left(\frac{d^2\beta + t}{d^2\beta}\right)^{4b\lambda_1} + (1+\delta_2)\exp\left(\sum_{t=1}^n 4\lambda_1\beta_t + 11\lambda_1^2\beta_t^2\right)$$

$$= c\sum_{t=1}^n(d\beta_t^2 r_t)\left(\frac{d^2\beta + t}{d^2\beta}\right)^{4b\lambda_1} - \delta_2\exp\left(\sum_{t=1}^n 4\lambda_1\beta_t + 11\lambda_1^2\beta_t^2\right)$$

$$+ \frac{cb}{(d^2\beta)^{4b\lambda_1}}\sum_{t=1}^{t_0}(d^2\beta + t)^{4b\lambda_1 - 1} + (1+\delta_2)\exp\left(\sum_{t=1}^n 4\lambda_1\beta_t + 11\lambda_1^2\beta_t^2\right)$$

$$\overset{\zeta_1}{\leq} c\sum_{t=1}^n(d\beta_t^2 r_t)\left(\frac{d^2\beta + t}{d^2\beta}\right)^{4b\lambda_1} - \delta_2\exp\left(\sum_{t=1}^n 4\lambda_1\beta_t + 11\lambda_1^2\beta_t^2\right) + \frac{c^{4b\lambda_1}}{\lambda_1}$$

$$+ (1+\delta_2)\exp\left(\sum_{t=1}^n 4\lambda_1\beta_t + 11\lambda_1^2\beta_t^2\right)$$

$$\leq \frac{cdb^2}{(d^2\beta)^{4b\lambda_1}} \sum_{t=1}^{n} \log^3(d^2\beta + t)(d^2\beta + t)^{4b\lambda_1 - 2} - \delta_2 \exp\left(\sum_{t=1}^{n} 4\lambda_1\beta_t + 11\lambda_1^2\beta_t^2\right)$$

$$+ \frac{c^{4b\lambda_1}}{\lambda_1} + (1 + \delta_2) \exp\left(\sum_{t=1}^{n} 4\lambda_1\beta_t + 11\lambda_1^2\beta_t^2\right)$$

$$\leq \frac{cdb^2 \log^3(d^2\beta + n)}{(d^2\beta)^{4b\lambda_1}} \sum_{t=1}^{n} (d^2\beta + t)^{4b\lambda_1 - 2} - \delta_2 \exp\left(\sum_{t=1}^{n} 4\lambda_1\beta_t + 11\lambda_1^2\beta_t^2\right) + \frac{c^{4b\lambda_1}}{\lambda_1}$$

$$+ (1 + \delta_2) \exp\left(\sum_{t=1}^{n} 4\lambda_1\beta_t + 11\lambda_1^2\beta_t^2\right)$$

$$\leq \frac{cdb^2 \log^3(d^2\beta + n)}{(d^2\beta)^{4b\lambda_1}} \sum_{t=1}^{n} (d^2\beta + t)^{4b\lambda_1 - 2} - \delta_2 \left(\frac{d^2\beta + n + 1}{d^2\beta + 1}\right)^{4b\lambda_1} + \frac{c^{4b\lambda_1}}{\lambda_1}$$

$$+ (1 + \delta_2) \exp\left(\sum_{t=1}^{n} 4\lambda_1\beta_t + 11\lambda_1^2\beta_t^2\right),$$

where $\zeta_1$ follows by using the fact that $\sum_{i=1}^{n} i^\gamma \leq n^{\gamma+1}/\gamma + 1$ for $\gamma > -1$. We consider now the following three cases as before:

**Case 1:** $4b\lambda_1 < 1$

In this case, we can upper bound the term $\frac{\mathbb{E}\left[(\tilde{u}_1^\top H_n H_n^\top \tilde{u}_1)^2\right]}{\|\tilde{u}_1\|_2^4}$ as,

$$\frac{\mathbb{E}\left[(\tilde{u}_1^\top H_n H_n^\top \tilde{u}_1)^2\right]}{\|\tilde{u}_1\|_2^4} \leq \frac{cb^2 \log^3(d^2\beta + n)}{d\beta} - \delta_2 \left(\frac{d^2\beta + n + 1}{d^2\beta + 1}\right)^{4b\lambda_1} + \frac{c^{4b\lambda_1}}{\lambda_1}$$

$$+ (1 + \delta_2) \exp\left(\sum_{t=1}^{n} 4\lambda_1\beta_t + 11\lambda_1^2\beta_t^2\right)$$

$$\overset{\zeta_1}{\leq} (1 + \delta_2) \exp\left(\sum_{t=1}^{n} 4\lambda_1\beta_t + 11\lambda_1^2\beta_t^2\right),$$

where $\zeta_1$ follows from using that

$$\frac{d^2\beta + n}{\log^{\frac{3}{4b\lambda_1}}(d^2\beta + n)} \geq \frac{c(d^2\beta + 1)}{(\delta_2 \log^3(d\beta))^{\frac{1}{4b\lambda_1}}}.$$

**Case 2:** $4b\lambda_1 > 1$

In this case, we can upper bound the term $\frac{\mathbb{E}\left[(\tilde{u}_1^\top H_n H_n^\top \tilde{u}_1)^2\right]}{\|\tilde{u}_1\|_2^4}$ as,

$$\frac{\mathbb{E}\left[(\tilde{u}_1^\top H_n H_n^\top \tilde{u}_1)^2\right]}{\|\tilde{u}_1\|_2^4} \leq \frac{cdb^2 \log^3(d^2\beta + n)}{(d^2\beta)^{4b\lambda_1}} \sum_{t=1}^{n} (d^2\beta + t)^{4b\lambda_1 - 2} - \delta_2 \left(\frac{d^2\beta + n + 1}{d^2\beta + 1}\right)^{4b\lambda_1} + \frac{c^{4b\lambda_1}}{\lambda_1}$$

$$+ (1 + \delta_2) \exp\left(\sum_{t=1}^{n} 4\lambda_1\beta_t + 11\lambda_1^2\beta_t^2\right)$$

$$\overset{\zeta_1}{\leq} \frac{cdb^2 \log^3(d^2\beta + n)}{(d^2\beta)^{4b\lambda_1}} (d^2\beta + n)^{4b\lambda_1 - 1} - \delta_2 \left(\frac{d^2\beta + n + 1}{d^2\beta + 1}\right)^{4b\lambda_1} + \frac{c^{4b\lambda_1}}{\lambda_1}$$

$$+ (1 + \delta_2) \exp\left(\sum_{t=1}^{n} 4\lambda_1\beta_t + 11\lambda_1^2\beta_t^2\right)$$

$$\overset{\zeta_2}{\leq} (1 + \delta_2) \exp\left(\sum_{t=1}^{n} 4\lambda_1\beta_t + 11\lambda_1^2\beta_t^2\right),$$

where $\zeta_2$ follows by using that

$$\frac{d^2\beta + n}{\log^3(d^2\beta + n)} \geq \frac{c^{4b\lambda_1}}{\delta_2}(d^2\beta + 1).$$

**Case 3:** $4b\lambda_1 = 1$

In this case, we can upper bound the term $\frac{\mathbb{E}\left[(\tilde{u}_1^\top H_n H_n^\top \tilde{u}_1)^2\right]}{\|\tilde{u}_1\|_2^4}$ as,

$$
\frac{\mathbb{E}\left[(\tilde{u}_1^\top H_n H_n^\top \tilde{u}_1)^2\right]}{\|\tilde{u}_1\|_2^4} \leq \frac{cdb^2\log^3(d^2\beta+n)}{(d^2\beta)}\sum_{t=1}^{n}(d^2\beta+t)^{-1} - \delta_2\left(\frac{d^2\beta+n+1}{d^2\beta+1}\right) + \frac{c}{\lambda_1}
$$

$$
+ (1+\delta_2)\exp\left(\sum_{t=1}^{n}4\lambda_1\beta_t + 11\lambda_1^2\beta_t^2\right)
$$

$$
\leq \frac{cdb^2\log^3(d^2\beta+n)}{(d^2\beta)}\log\left(\frac{d^2\beta+n}{d^2\beta}\right) - \delta_2\left(\frac{d^2\beta+n+1}{d^2\beta+1}\right) + \frac{c}{\lambda_1}
$$

$$
+ (1+\delta_2)\exp\left(\sum_{t=1}^{n}4\lambda_1\beta_t + 11\lambda_1^2\beta_t^2\right)
$$

$$
\overset{\zeta_1}{\leq} (1+\delta_2)\exp\left(\sum_{t=1}^{n}4\lambda_1\beta_t + 11\lambda_1^2\beta_t^2\right),
$$

where $\zeta_1$ holds due to

$$
\frac{d^2\beta+n}{\log^4(d^2\beta+n)} \geq \frac{cd}{\delta_2}.
$$

$\square$

**Convergence Theorem**: We begin by restating the bound obtained on $\mathbb{E}\left[\text{Tr}(V_\perp^\top H_t H_t^\top V_\perp)\right]$ in Lemma E.2,

$$
\mathbb{E}\left[\text{Tr}(V_\perp^\top H_n H_n^\top V_\perp)\right] \leq \exp\left(\sum_{t=1}^{n}2\beta_t\lambda_2 + \beta_t^2\lambda_2^2\right)
$$

$$
\left(\text{Tr}(V_\perp V_\perp^\top) + cd\|V_\perp V_\perp^\top\|_2 \sum_{t=1}^{n}\left(r_t\beta_t^2 + \mathbb{I}(t\leq t_0)\beta_t d\right)\exp\left(2\sum_{i=1}^{t}\beta_i(\lambda_1-\lambda_2) + cd\beta_i^2 r_i + c\sum_{i=1}^{t_0}\beta_i d\right)\right)
$$

$$
\overset{\zeta_1}{\leq} \exp\left(\sum_{t=1}^{n}2\beta_t\lambda_2 + \beta_t^2\lambda_2^2\right)\left(\text{Tr}(V_\perp V_\perp^\top)\right. \tag{58}
$$

$$
\left. + cd\|V_\perp V_\perp^\top\|_2 \sum_{t=1}^{n}\left(r_t\beta_t^2 + \mathbb{I}(t\leq t_0)\beta_t d\right)\exp\left(2\sum_{i=1}^{t}\beta_i(\lambda_1-\lambda_2) + cd\beta_i^2 r_i\right)\right), \tag{59}
$$

where $\zeta_1$ follows from using Equation (57).

**Theorem F.3** (Convergence Theorem). *Let $\delta > 0$ and the step sizes $\beta_i = \frac{b}{d^2\beta+i}$. The output $v_n$ of Algorithm 1 for $n$ satisfying the assumption in Lemma F.1 and F.2 is an $\epsilon$-approximation to $u_1$ with probability atleast $1-\delta$ where,*

$$
\underbrace{\sin_B^2(u_1, v_n)}_{\epsilon} \leq \frac{d\|V_\perp V_\perp^\top\|_2}{Q}\exp\left(5\lambda_1^2\sum_{t=1}^{n}\beta_t^2\right)\left(\exp\left(-2\Delta_\lambda\sum_{t=1}^{n}\beta_t\right)\right.
$$

$$
\left. + c\sum_{t=1}^{n}\left(r_t\beta_t^2 + \mathbb{I}(t\leq t_0)\beta_t d\right)\exp\left(-2\Delta_\lambda\sum_{i=t+1}^{n}\beta_i\right)\right),
$$

*where $\Delta_\lambda = \lambda_1 - \lambda_2$ and*

$$
Q = \frac{2\delta^2\|\tilde{u}_1\|_2^2}{(2+\epsilon_1)c\log(1/\delta)}\left(1 - \frac{1}{\sqrt{\delta}}\sqrt{(1+\epsilon_1)\exp\left(19\sum_{t=1}^{n}\beta_t^2\lambda_1^2\right) - 1}\right),
$$

*The constant $c$ occuring in the equations, as before depends polynomially on problem dependent paramters $b$, $\lambda_1$, $\Delta_\lambda$ and the parameters $\frac{\delta_1}{2} = \delta_2 = \frac{\epsilon_1}{2+\epsilon_1}$.*

*Proof.* First, using the Chebychev's inequality, we have:

$$\mathbb{P}\left[\left|\tilde{u}_1^\top H_n H_n^\top \tilde{u}_1 - \mathbb{E}\left[\tilde{u}_1^\top H_n H_n^\top \tilde{u}_1\right]\right| \geq \frac{1}{\delta}\sqrt{\mathrm{Var}[\tilde{u}_1^\top H_n H_n^\top \tilde{u}_1]}\right] \leq \delta.$$

With probability greater than $1 - \delta$, we have,

$$\tilde{u}_1^\top H_n H_n^\top \tilde{u}_1 \geq \mathbb{E}\left[\tilde{u}_1^\top H_n H_n^\top \tilde{u}_1\right] - \frac{1}{\sqrt{\delta}}\sqrt{\mathrm{Var}\left[\tilde{u}_1^\top H_n H_n^\top \tilde{u}_1\right]}$$

$$= \mathbb{E}\left[\tilde{u}_1^\top H_n H_n^\top \tilde{u}_1\right]\left(1 - \frac{1}{\sqrt{\delta}}\sqrt{\frac{\mathbb{E}[(\tilde{u}_1^\top H_n H_n^\top \tilde{u}_1)^2]}{\mathbb{E}[\tilde{u}_1^\top H_n H_n^\top \tilde{u}_1]^2} - 1}\right) \tag{60}$$

Now, using Lemma F.2, we have that,

$$\frac{\mathbb{E}\left[(\tilde{u}_1^\top H_n H_n^\top \tilde{u}_1)^2\right]}{\|\tilde{u}_1\|_2^4} \leq (1 + \delta_2)\exp\left(\sum_{t=1}^n 4\lambda_1\beta_t + 11\lambda_1^2\beta_t^2\right) \tag{61}$$

and using Lemma F.1, we have,

$$\frac{\mathbb{E}[\tilde{u}_1^\top H_n H_n^\top \tilde{u}_1]}{\|\tilde{u}_1\|_2^2} \geq (1 - \delta_1)\exp\left(\sum_{t=1}^n 2\beta_t\lambda_1 - 4\beta_t^2\lambda_1^2\right),$$

squaring the above, we obtain,

$$\frac{\mathbb{E}[\tilde{u}_1^\top H_n H_n^\top \tilde{u}_1]^2}{\|\tilde{u}_1\|_2^4} \geq (1 - \delta_1')\exp\left(\sum_{t=1}^n 4\beta_t\lambda_1 - 8\beta_t^2\lambda_1^2\right), \tag{62}$$

where $\delta_1' = 2\delta_1$. Setting $\delta_1' = \delta_2 = \frac{\epsilon_1}{2+\epsilon_1}$ and substituting bounds (61) and (62) in (60), we obtain,

$$\tilde{u}_1^\top H_n H_n^\top \tilde{u}_1 \geq \frac{2\|\tilde{u}_1\|_2^2}{2 + \epsilon_1}\exp\left(\sum_{t=1}^n 2\beta_t\lambda_1 - 4\beta_t^2\lambda_1^2\right)\left(1 - \frac{1}{\sqrt{\delta}}\sqrt{(1 + \epsilon_1)\exp\left(19\sum_{t=1}^n \beta_t^2\lambda_1^2\right) - 1}\right).$$

Further, using the Equation (58) along with Markov's inequality, we have with probability atleast $1 - \delta$

$$\mathrm{Tr}(V_\perp^\top H_n H_n^\top V_\perp) \leq \frac{1}{\delta}\exp\left(\sum_{t=1}^n 2\beta_t\lambda_2 + \beta_t^2\lambda_2^2\right)\left(\mathrm{Tr}(V_\perp V_\perp^\top)\right.$$

$$\left. + cd\|V_\perp V_\perp^\top\|_2\sum_{t=1}^n \left(r_t\beta_t^2 + \mathbb{I}(t \leq t_0)\beta_t d\right)\exp\left(2\sum_{i=1}^t \beta_i(\lambda_1 - \lambda_2) + cd\beta_i^2 r_i\right)\right).$$

Combining the above with Lemma 6.2, we have that the output $v_n$ of Algorithm 1 is an $\epsilon$-approximation to $u_1$ with probability atleast $1 - \delta$,

$$\epsilon \leq \frac{c\log(1/\delta)(2 + \epsilon_1)}{2\delta\|\tilde{u}_1\|_2^2}\frac{\exp\left(\sum_{t=1}^n -2\beta_t\lambda_1 + 4\beta_t^2\lambda_1^2\right)\mathrm{Tr}(V_\perp^\top H_n H_n V_\perp)}{\left(1 - \frac{1}{\sqrt{\delta}}\sqrt{(1 + \epsilon_1)\exp\left(19\sum_{t=1}^n \beta_t^2\lambda_1^2\right) - 1}\right)}$$

$$\leq \frac{d\|V_\perp V_\perp^\top\|_2}{Q}\exp\left(5\lambda_1^2\sum_{t=1}^n \beta_t^2\right)\left(\exp\left(-2\Delta_\lambda\sum_{t=1}^n \beta_t\right)\right.$$

$$\left. + c\sum_{t=1}^n \left(r_t\beta_t^2 + \mathbb{I}(t \leq t_0)\beta_t d\right)\exp\left(-2\Delta_\lambda\sum_{i=t+1}^n \beta_i\right)\right),$$

where $\Delta_\lambda = \lambda_1 - \lambda_2$ and

$$Q = \frac{2\delta^2\|\tilde{u}_1\|_2^2}{(2 + \epsilon_1)c\log(1/\delta)}\left(1 - \frac{1}{\sqrt{\delta}}\sqrt{(1 + \epsilon_1)\exp\left(19\sum_{t=1}^n \beta_t^2\lambda_1^2\right) - 1}\right).$$

$\square$

**Main Result**: Next, we state our main theorem and instantiate the parameters of our algorithm.

**Theorem F.4** (Main Result). *Fix any $\delta > 0$ and $\epsilon_1 > 0$. Suppose that the step sizes are set to $\alpha_t = \frac{c}{\log(d^2\beta+t)}$ and $\beta_t = \frac{\gamma}{\Delta_\lambda(d^2\beta+t)}$ for $\gamma > 1/2$ and*

$$\beta = \max\left( \frac{20\gamma^2\lambda_1^2}{\Delta_\lambda^2 d^2 \log\left(\frac{1+\delta/100}{1+\epsilon_1}\right)}, \frac{200\left(\frac{R}{\mu} + \frac{R^3}{\mu^2} + \frac{R^5}{\mu^3}\right)\log(1+\frac{R^2}{\mu}+\frac{R^4}{\mu^2})}{\delta\Delta_\lambda^2} \right).$$

*Suppose that the number of samples $n$ satisfy the assumptions of Lemma F.1 and F.2. Then, the output $v_n$ of Algorithm 1 satisfies,*

$$\sin_B^2(u_1, v_n) \le \frac{(2+\epsilon_1)cd\|\sum_{i=1}^d \tilde{u}_i\tilde{u}_i^\top\|_2 \log\left(\frac{1}{\delta}\right)}{\delta^2 \|\tilde{u}_1\|_2^2}\left( \left(\frac{d^2\beta+1}{d^2\beta+n+1}\right)^{2\gamma} + \frac{c\gamma^2\log^3(d^2\beta+n)}{\Delta_\lambda^2(d^2\beta+n+1)} \right.$$

$$\left. + \frac{cd}{\Delta_\lambda}\left(\frac{d^2\beta+\log^3(d^2\beta)}{d^2\beta+n+1}\right)^{2\gamma} \right),$$

*with probability at least $1-\delta$ with $c$ depending polynomially on parameters of the problem $\lambda_1, \kappa_B, R, \mu$. The parameters $\delta_1, \delta_2$ are set as $\delta_1 = \frac{\epsilon_1}{2(2+\epsilon_1)}$ and $\delta_2 = \frac{\epsilon_1}{2+\epsilon_1}$.*

*Proof.* With the step size $\beta_t = \frac{b}{d^2\beta+t}$, we set the parameter $b = \frac{\gamma}{\lambda_1-\lambda_2}$ and thus we get $\beta_t = \frac{\gamma}{\Delta_\lambda(d^2\beta+t)}$. Now, we have that

$$\sum_{t=1}^n \beta_t^2 \le \frac{\gamma^2}{\Delta_\lambda^2 d^2\beta}$$

and using the assumption that $\frac{\gamma^2\lambda_1^2}{\Delta_\lambda^2 d^2\beta} \le \frac{1}{19}\log\left(\frac{1+\frac{\delta}{100}}{1+\epsilon_1}\right)$, we obtain,

$$\sqrt{((1+\epsilon_1)\exp\left(19\sum_{t=1}^n \beta_t^2\lambda_1^2\right)-1)} \ge \frac{9}{10} \qquad \Rightarrow \qquad Q \ge \frac{c\delta^2\|\tilde{u}_1\|_2^2}{(2+\epsilon_1)\log(1/\delta)}. \qquad (63)$$

Using previous bounds on sums of partial harmonic sums, we have that,

$$\sum_{t=1}^n \beta_t \ge \frac{\gamma}{\Delta_\lambda}\log\left(\frac{d^2\beta+n+1}{d^2\beta+1}\right) \qquad \text{and} \qquad \sum_{i=t+1}^n \beta_i \ge \frac{\gamma}{\Delta_\lambda}\log\left(\frac{d^2\beta+n+1}{d^2\beta+t+1}\right).$$

Using these bounds, we obtain,

$$\exp\left(-2\Delta_\lambda\sum_{t=1}^n \beta_t\right) \le \left(\frac{d^2\beta+1}{d^2\beta+n+1}\right)^{2\gamma}. \qquad (64)$$

In order to bound the remaining terms from Theorem F.3, we note that,

$$c\sum_{t=1}^n \left(r_t\beta_t^2 + \mathbb{I}(t \le t_0)\beta_t d\right)\exp\left(-2\Delta_\lambda\sum_{i=t+1}^n \beta_i\right)$$

$$\le c\sum_{t=1}^n \left(r_t\beta_t^2 + \mathbb{I}(t \le t_0)\beta_t d\right)\left(\frac{d^2\beta+t+1}{d^2\beta+n+1}\right)^{2\gamma}$$

$$\le c\sum_{t=1}^n \frac{r_t\gamma^2}{(\Delta_\lambda)^2(d^2\beta+t)^2}\left(\frac{d^2\beta+t+1}{d^2\beta+n+1}\right)^{2\gamma} + cd\sum_{t=1}^{t_0} \frac{\gamma}{\Delta_\lambda(d^2\beta+t)}\left(\frac{d^2\beta+t+1}{d^2\beta+n+1}\right)^{2\gamma}$$

$$\le \frac{c\gamma^2\log^3(d^2\beta+n)}{\Delta_\lambda^2(2\gamma-1)(d^2\beta+n+1)} + \frac{cd}{\Delta_\lambda}\left(\frac{d^2\beta+\log^3(d^2\beta)}{d^2\beta+n+1}\right)^{2\gamma}, \qquad (65)$$

where the last bounds holds for any $\gamma > 1/2$. Substituting bounds (63),(64) and (65) in the result of Theorem F.3, we obtain that the output $v_n$ of Algorithm 1 satisfies,

$$\sin_B^2(u_1, v_n) \le \frac{(2+\epsilon_1)cd\|\sum_{i=1}^d \tilde{u}_i\tilde{u}_i^\top\|_2 \log\left(\frac{1}{\delta}\right)}{\delta^2 \|\tilde{u}_1\|_2^2}\left( \left(\frac{d^2\beta+1}{d^2\beta+n+1}\right)^{2\gamma} + \frac{c\gamma^2\log^3(d^2\beta+n)}{\Delta_\lambda^2(d^2\beta+n+1)} \right.$$

$$+\frac{cd}{\Delta_\lambda}\left(\frac{d^2\beta+\log^3(d^2\beta)}{d^2\beta+n+1}\right)^{2\gamma}\Bigg).$$

$\square$

# G  Auxiliary Properties

## G.1  Useful Trace Inequalities

In this section we enumerate some useful inequalities.

**Lemma G.1.**  *1. $\langle A, B\rangle \le \langle A, C\rangle$ for PSD matrices $A, B, C$ with $B \preceq C$.*

    *2. $\mathrm{Tr}(A^\top B) \le \frac{1}{2}\mathrm{Tr}(A^\top A + B^\top B)$ for all matrices $A, B \in \mathbb{R}^{m\times n}$.*

As a consequence:

**Corollary G.1.1.** *$\langle A, B\rangle \le \langle A, C\rangle$ for a PSD matrix $A$ and $B \preceq C$, with $B$ and $C$ symmetric.*

*Proof.* If $B$ is PSD, the result follows immediately from the previous lemma. Otherwise let $\lambda_{min}$ be the smallest eigenvalue of $B$. Let $B' = B + |\lambda_{min}|I$ and $C' = C + |\lambda_{min}|I$. The matrices $B'$ and $C'$ are PSD and satisfy $B' \preceq C'$. The result follows by applying the lemma above and rearranging the terms. $\square$

## G.2  Useful spectral norm Inequalities

In this section we enumerate some useful inequalities.

**Lemma G.2.** *If $0 \preceq B \preceq C$ and symmetric then $0 \preceq ABA^\top \preceq ACA^\top$.*

As a consequence:

**Corollary G.2.1.** *If $0 \preceq B \preceq C$ and symmetric then $\|ABA^\top\| \le \|ACA^\top\|$.*

## G.3  Properties concerning Eigenvectors of $B^{-1}A$

In this subsection, we highlight some important properties concerning the left and right eigenvectors of the matrix under consideration $B^{-1}A$.

As before, we let $\tilde{u}_1, \ldots, \tilde{u}_d$ denote the left eigenvetors and $u_1, \ldots, u_d$ denote the right eigenvectors of $B^{-1}A$.

**Lemma G.3.** *The right eigenvectors of the matrix $B^{-1}A$ satisfy the following:*

$$u_i^\top B u_j = 0 \quad \text{if } i \ne j.$$

*Proof.* Consider the symmetric matrix $C = B^{-1/2}AB^{-1/2}$. Let $u_1^C, \ldots, u_d^C$ be the eigenvectors of $C$. Notice that if $u_i^C$ is an eigenvector of $C$ with eigenvalue $\lambda_i$, then

$$B^{-1/2}(B^{-1/2}AB^{-1/2})u_i^C = \lambda_i B^{-1/2}u_i^C,$$

implying that $B^{-1/2}u_i^C$ is a right eigenvector of $B^{-1}A$, $u_i$. Therefore the eigenvector of $C$ are related to the righteigenvectors of $B^{-1}A$ as $B^{1/2}u_i = u_i^C$. Further, since the matrix $C$ is symmetric, its eigencvectors can be taken to form an orthogonal basis, and hence,

$$(u_i^C)^\top u_j^C = u_i^\top B u_j = 0 \quad \text{if } i \ne j.$$

$\square$

**Lemma G.4.** *Let $u_1$ denote the top right eigenvector of $B^{-1}A$. Then,*

$$\tilde{u}_i^\top u_1 = 0 \quad \text{for all } i \ge 2,$$

*where $\tilde{u}_i$ represent the left eigenvectors of the matrix $B^{-1}A$.*

*Proof.* We begin by noting that the left and right eigenvectors of the matrix $B^{-1}A$ are related as $\tilde{u}_i = Bu_i$, which follows from,

$$(B^{-1}A)B^{-1}\tilde{u}_i = B^{-1}(AB^{-1})\tilde{u}_i = \lambda_i B^{-1}\tilde{u}_i$$

As a consequence $B^{-1}\tilde{u}_i$ is a right eigenvector of $B^{-1}A$ and the lemma now follows from using Lemma G.3.

$\square$

As a consequence of Lemma G.4, we have the following corollary relating the orthogonal subspace of $u_1$ to the left eigenvectors $\tilde{u}_2, \ldots, \tilde{u}_d$.

**Corollary G.4.1.** *If $\lambda_1$ has multiplicity* 1*, the space orthogonal to $u_1$ is spanned by the vectors* $\{\tilde{u}_2, \ldots, \tilde{u}_d\}$.