[Reviews · NeurIPS 2018]

Reviewer 1



This paper addresses a problem of obtaining a principal generalized eigenvector in stochastic setting. The algorithm is considered as an extension of Oja algorithm to the stochastic setting, and its convergence proof is provided with its rate of convergence. The paper is well organized and reads well. The simulation done is this paper looks fair enough, and reveals the sensitivity of the parameter choice. Though it may not be the scope of this paper, it would be nice if one could provide an example machine learning case where the result of the proposed algorithm is used for feature extraction. It is unclear to me how small the error (e.g., Figure 1) should be to obtain a reasonable results, let's say, classification error. I understand that it will be different task-by-task, and should depend on the dimension (meaning of the error changes), so it may be difficult to provide the meaning of the error, but an example use case will be helpful. (Rebuttal) I believe that adding a paragraph for connecting the proposed approach to ML tasks will increase readers who will be interested in this topic. The author's rebuttal is satisfactory in this point. On the other hand, the second point was not clarified by the rebuttal. I will keep the score as it was.

Reviewer 2



I felt that the problem was well-motivated, and I liked the amount of intuition presented in the paper. In full disclosure, however, I did not verify all the algebra in the appendix. The review of previous literature suggests that while other algorithms for this problem may exist, the present one is significantly simpler as it avoids variance reduction and does not require a separate "SGD phase" for each Oja step. This seems to result in significantly faster convergence in the experiments section. Clarity: The paper is well-written, but there are a few very minor issues: 1. in equation (6) , did you actually mean this to be the definition of sin^2 rather than sin? 2. line 203: "is a the" => "is the" 3. line 207: "guarantee are" => "guarantee is" 4. In section 2, I think you should consider using u_i instead of v_i here, to avoid overlapping with the v_i of the algorithm, and be more consistent with the u_i of the previous notation paragraph. Originality: I don't know of any other algorithms for this problem, but I am not an expert of the literature in this area. Significance: The algorithm seems a significant improvement on prior works. However, it seems that a little more analysis of the iterate-averaging version of the algorithm is necessary to prove the more robust version of the algorithm, as otherwise it seems somewhat sensitive to the tuning of the learning rates.

Reviewer 3



Gen-Oja: A Simple and Efficient Algorithm for Streaming Generalized Eigenvector Computation Principal Generalized Eigenvector computation and Canonical Correlation Analysis in the stochastic setting. Propose a simple and efficient algorithm for these problems. Overall the approach appears to be reasonable and is scientific sound. The experimental part is however quite limited - the paper is widely a theoretical proposal and it remains unclear if it is really effective in a practical setting. The paper is also closely linked to streaming PCA and in parts it is not so clear if there is some strong novelty in this proposal or just a lot of mathematical wrapping. comments: - how critical is the i.i.d. condition after Eq. 5 - in my view it is a bit unlikely to assume that a data stream can be sampled in this way under realistic conditions - so one may only 'hope' that it holds, or it can be applied only in some scenaria - as the experiments are very limited the i.i.d. assumption is a key problem and it is not shown in the experiments that this assumption is not a particular problem - related work is discussed but not much addressed in the experimental part as outlined in the paper there is a lot related work - but not in your experiments - although the theory looks convincing it would be good to show the effectiveness also in a number of practical application. Numerical problems are common for such type of 'approximations' and behind theory the practical results may be different - it may also not harm to add some discussions to practical results (--> in the supplementary material) - I have not checked in detail the supplementary material --> journal submission? Comments - to author feedback: - thanks for addressing the comments - some points are clarified - I (now) understand that the authors are focusing on a theoretical improvement -- I think it would be good to adapt the wording in the paper and maybe also in th title to make this more explicit